# Location-dependent threat and associated neural abnormalities in clinical anxiety

Benjamin Suarez-Jimenez [1,2,3 ✉], Nicholas L. Balderston [3,4], James A. Bisby [5,6,7], Joseph Leshin [3,8], Abigail Hsiung[3,9], John A. King[10], Daniel S. Pine[3], Neil Burgess [5,6,11], Christian Grillon[3] & Monique Ernst[3]

Anxiety disorders are characterized by maladaptive defensive responses to distal or uncertain threats. Elucidating neural mechanisms of anxiety is essential to understand the development and maintenance of anxiety disorders. In fMRI, patients with pathological anxiety (ANX, $n = 23$) and healthy controls (HC, $n = 28$) completed a contextual threat learning paradigm in which they picked flowers in a virtual environment comprising a danger zone in which flowers were paired with shock and a safe zone (no shock). ANX compared with HC showed 1) decreased ventromedial prefrontal cortex and anterior hippocampus activation during the task, particularly in the safe zone, 2) increased insula and dorsomedial prefrontal cortex activation during the task, particularly in the danger zone, and 3) increased amygdala and midbrain/periaqueductal gray activation in the danger zone prior to potential shock delivery. Findings suggest that ANX engage brain areas differently to modulate context-appropriate emotional responses when learning to discriminate cues within an environment.

[1] Neuroscience Department, University of Rochester, Rochester, NY, USA. [2] Department of Psychiatry, Columbia University Medical Center, New York, NY, USA. [3] National Institute of Mental Health, Bethesda, MD, USA. [4] University of Pennsylvania School of Medicine, Philadelphia, PA, USA. [5] Institute of Cognitive Neuroscience, University College London, London, UK. [6] Queen Square Institute of Neurology, University College London, London, UK. [7] Division of Psychiatry, University College London, London, UK. [8] Department of Psychology and Neuroscience, The University of North Carolina, Chapel Hill, NC, USA. [9] Department of Psychology and Neuroscience, Duke University, Durham, NC, USA. [10] Department of Clinical Psychology, University College London, London, UK. [11] Wellcome Centre for Human Neuroimaging, University College London, London, UK. ✉email: Benjamin_Suarez-jimenez@URMC.Rochester.edu

When exploring our environment, we might encounter items that require us to learn about their threat value. Learning about potential threatening environments may induce anxiety, an anticipatory response to potential threats, and a fearful response evoked by an imminent acute threat[1–5]. Although anxious and fearful states are normal responses to threats, chronic manifestations of these states can be highly debilitating[6,7]. Research shows that patients with chronic anxiety lack the ability to integrate contextual cues to guide learning of threat and safety[8–10]. A previous investigation delineated the neural mechanisms underlying learning and discriminating threats within specific spatial locations in healthy adults[11]. However, very little is known about how patients with pathological anxiety learn about threat within an environment. Understanding how patients with pathological anxiety learn about threats within specific spatial locations in complex environments is essential to better understand the development and maintenance of the disorder.

Traditional context conditioning paradigms have shown that a defensive response can be triggered not only by an aversive stimulus but also by the context where the stimulus was encountered. Typically, healthy individuals can learn to distinguish between safe and dangerous contexts. That is, when healthy individuals associate a cue with an aversive stimulus (conditioned stimulus; CS) in context A (CS + A), they display a threat response that is dependent on the context in which the association was made. However, the same cue (CS) in a different context B (CS-B) elicits a weaker defensive response. Context conditioning can engage spatial processing strategies[8–10] (i.e., attend to the environment and surrounding landmarks to create a spatial representation of where a threat was encountered). These strategies have been mapped to neural systems that regulate emotion and memory, such as the hippocampus, amygdala, and prefrontal cortex (PFC)[8,10,12–20].

We developed a virtual-environment paradigm to probe how brain regions interact to shape behavior (i.e., threat learning and discrimination) over time[11] to quantify threat learning relevant to context-specific threat. The virtual environment depicts a circular grass field surrounded by mountains, divided equally into two zones, a safe and a danger zone. In both zones, flowers appear and need to be "picked" up. Picking flowers in the danger zone potentially causes an electric shock to the wrist (or "bee sting"), while flowers in the safe zone are never associated with shock. To learn threat contingencies, participants must rely on distal environmental cues (e.g., shape of the mountains and clouds, which differ in both zones, and beehives) to locate themselves and learn "where" they are in the environment and not on the physical properties of the stimuli (i.e., the flowers), which are all identical in both zones[11]. In other words, participants must learn to discriminate where in the environment is the threshold that divides the safe and danger zones (using the distal environmental cues) as there are no clear division in the circular grassy field, which is also identical throughout the circumference of the circle.

Previous findings with this paradigm in healthy volunteers informed three processes: threat learning, threat appraisal (anxiety-state), and threat anticipation (fear-state). (1) Threat learning: Healthy adults demonstrated behavioral/physiological learning: as the task progressed, shock expectancy ratings and skin conductance for flowers increased in the danger but not the safe zone. The neural substrates associated with learning about environmental threats during the flower approach, in either zone, engaged key nodes of the learning circuit, including the anterior hippocampus, amygdala, ventromedial prefrontal cortex (vmPFC), and vmPFC-hippocampal functional connectivity. (2) Threat appraisal: approach of the flower, in the danger zone compared to the safe zone, recruited sensory and control-related regions, i.e., the insula, dorsal anterior cingulate cortex (dACC), extending to the dorsomedial prefrontal cortex (dmPFC), and insula-hippocampal functional connectivity. (3) Threat anticipation: During the imminent threat of potential shock, upon picking a flower, a progressively increasing response of the periaqueductal gray matter (PAG) and posterior hippocampus and insula-dACC coupling were observed[11].

Research has shown that individuals with anxiety disorders display a higher defensive response to a safe cue (CS−), compared to healthy individuals, suggesting impaired ability to regulate their emotions or a generalization of the threat response to safe cues[6,7,12,14,21]. A discrimination of rings task has been used to systematically elucidate neural signatures of generalization/discrimination in patients with panic disorder[22], generalized anxiety disorder[23], and PTSD[24]. These studies show that patients, compared to healthy controls, exhibit an overgeneralization, or lack of discrimination, towards cues similar to the CS+. But it is not clear if this overgeneralization extends to context, particularly when there is no clear-cut boundary between safe and danger zones within the environment. Nevertheless, hippocampal dysfunction and decreased hippocampal volume have been associated with anxiety disorders[12,13,15]. For example, studies in both humans and rats suggest that impairment in hippocampal function leads to compensatory learning strategies that do not involve the hippocampus. These abnormal modulations of attention, linked to attention shifts to the cue and not the context, use compensatory neural mechanisms that lead to generalization of threat[6–10,12,14,16–20]. This study aims to test if patients with anxiety disorder can discriminate between environmental zones (safe; danger) and what are the neural mechanisms engaged during the learning process.

Using the virtual-environment paradigm described above, three main hypotheses are tested. We expect that individuals with an anxiety disorder (ANX; generalized anxiety disorder, social anxiety disorder) compared with a sample of healthy controls (HC) will show the following: (1) Threat learning: poor learning and discrimination associated with compensatory learning strategies that do not involve the hippocampus leading to a generalization of threat, reflected during, (2) Threat appraisal: stronger engagement of the dACC, dmPFC, amygdala, and insula in danger zones when approaching the flowers (higher anxiety-state), and (3) Threat anticipation: stronger engagement of the periaqueductal gray matter (PAG) activation in danger zones in anticipation of potential shock delivery (higher fear-state).

## Results

**Physiological and behavioral measures of threat learning.** Participants had full control of the virtual character (first-person perspective) and explored a virtual circular environment (Fig. 1a, b; see "Methods" section for details). The environment consisted of a mountain landscape and defined two half-zones recognizable by the unique shape of the mountains in the horizon. For each trial, participants freely explored the environment and were instructed to pick up flowers that appeared one at a time in random locations across the environment (approach period). When a participant picked a flower, their position was held stationary for a variable duration (2–8 s; stationary period), during which the participant rated the expectancy of receiving a shock (rating of 0–9). After the stationary period, a new flower would appear for the participant to find. There was only one flower in the environment at a time. Flowers located in one-half of the environment were paired with a shock delivered at the end of the stationary period on 50% of the trials within the danger zone. Flowers in the other half of the environment were never paired with a shock (safe zone). Since all flowers were identical,

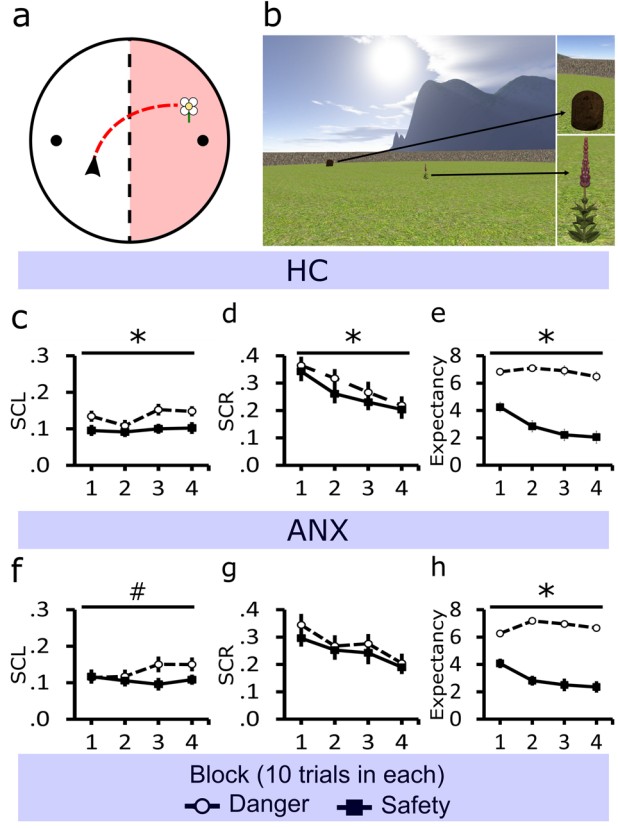

**Fig. 1 Task illustration and behavioral data across threat learning.**
**a** Helicopter view of the circular environment that participants (black arrow) explored (red trace) and how it was split into one-half associated with danger (red zone) and the other with safety. The environment included two beehives (black dots) located at opposite sides. Participants were required to collect flowers, which were generated within the environment. **b** Example of the participant's viewpoint, showing a beehive and flower in the environment. **c** HCs' and **f** ANXs' mean tonic skin conductance level (SCL) as flowers were approached. **d** HCs' and **g** ANXs' mean phasic skin conductance responses (SCR) during the stationary periods when flowers were picked. **e** HCs' and **h** ANXs' shock expectancy ratings at the onset of stationary periods when picking a flower. Error bars show standard error mean, *$p < 0.05$, #$p < 0.1$.

predictive value (danger or safety) could not be attributed to their physical characteristics. Participants had to build their own mental representation of the circular environment, which had no visible or distinctive boundaries between the zones, to define what they consider a safe and dangerous zone. For analysis, the data were divided by zones (safe, danger) and segregated into 4 learning blocks (10 trials in each).

Each approach period (approaching a flower) began at trial onset, when the flower appeared in the environment and ended when the flower was "collected." For analyses, we excluded the initial orienting period (looking for the flower) of the approach period, only including the last 75% of the approach (active navigation towards the flower). Each stationary period (after collecting the flower) began upon touching the flower, during this time the participant movement would be stopped for 2–8 s. Stationary periods were assessed for the entire 2–8 s duration.

*State-trait anxiety inventory (STAI).* Before and after the task participants completed the state anxiety section of the STAI. A $2 \times 2$ ANOVA (group by time) analysis of the state anxiety (pre-task, post-task) revealed a significant group effect ($F(1,$

22$) = 28.07$, $p = 3e{-}5$) showing an overall higher state anxiety in the ANX compared to the HC. No other significant main effects or interactions were observed ($F$'s $< 2$, $p$'s $> 0.05$). A post-hoc $t$-test revealed that ANX had higher state anxiety than HC only at post-task (pre-task, $t(49) = 2.56$, $p = 0.12$; post-task, $t(49) = 6.51$, $p = 0.014$).

*Skin conductance. Approach period*: Skin conductance level (SCL) was measured as participants navigated towards the flower. A $2 \times 2 \times 4$ ANOVA (zone by group by block) analysis of tonic changes in skin conductance showed a main effect of the zone with greater SCL when approaching flowers located in dangerous relative to safe zones ($F(1,49) = 16.24$, $p = 1e{-}4$). No other significant main effects or interactions were observed ($F$'s $< 2$, $p$'s $> 0.05$). Based on the first hypothesis, that ANX would evidence overgeneralization[12], within-group main effects of the zone were explored and indicated that only HCs had significantly higher SCL towards dangerous flowers as compared to safe flowers (HC, $F(27) = 15.55$, $p = 1e{-}3$; ANX, $F(22) = 3.97$, $p = 0.06$; Fig. 1c, f). However, a direct comparison between ANX and HC groups was not significant, and the ANX group showed similar effects at trend-wise levels.

*Stationary period*: Next, skin conductance responses (SCR) were analyzed during the stationary periods immediately after participants touched a flower, using a $2 \times 2 \times 4$ ANOVA (zone by group by block). This analysis revealed a main effect of the zone with greater SCRs to flowers located in dangerous compared to safe zones ($F(1,49) = 4.13$, $p = 0.04$). A main effect of the block was also observed with decreased SCRs over time ($F(3,47) = 17.62$, $p = 6e{-}6$). No other significant effects were found ($F$'s $< 2$, $p$'s $> 0.05$). Within-group main effects of zone were explored, showing that only HC's had significantly higher SCR towards dangerous flowers as compared to safe flowers (HC, $F(27) = 5.85$, $p = 0.02$; ANX, $F(22) = 0.97$, $p = 0.33$; Fig. 1d, g). However, a direct comparison between ANX and HC groups was not significant.

*Shock expectancy.* Assessing shock expectancy ratings, a $2 \times 2 \times 4$ ANOVA (zone by group by block) revealed a significant zone by block interaction ($F(3,47) = 31.07$, $p = 1e{-}15$). Also, significant main effects of zone ($F(1,49) = 121.08$, $p = 7e{-}15$) and block ($F(3,47) = 21.01$, $p = 2e{-}11$) were observed. No other significant effects were found ($F$'s $< 2$, $p$'s $> 0.05$). Further analysis of this interaction showed that within participants, danger zone shock expectancy ratings to flowers remained high from block 1 to block 4 ($t(50) = 0.06$, $p = 0.95$), while safe zone ratings significantly decreased (block 1 v block 4, $t(50) = 8.83$, $p = 8e{-}12$). Within-group main effects of zone were explored, showing that both groups showed higher expectancy ratings towards dangerous flowers as compared to safe flowers (HC, $F(1,27) = 59.27$, $p = 2e{-}8$; ANX, $F(1,22) = 67.73$, $p = 3e{-}8$; Fig. 1e, h).

*The spatial memory task.* Interleaved with these flower trials, participants performed a spatial memory task within the same environment (see "Methods" section for further details). Participants were required to learn the location of four objects, with two objects appearing on each side of the environment (i.e., the safe or danger zones, although objects were never paired with shock). Participants were required to replace objects where they had been found, and distance error from the correct location provided a measure of performance. This task served as a navigation control task to ensure that brain differences in the flower task were not confounded with the navigational effects of the virtual environment. Additionally, this task provided information that the participants were indeed learning to navigate their environment and were mapping the environment accurately.

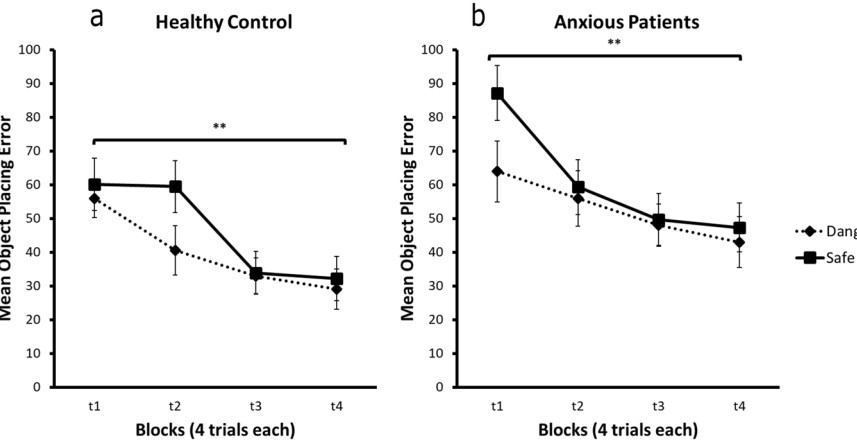

**Fig. 2 Mean object placement error. a** HC and **b** ANX mean object placement error, for the safe zone (Safe) and the dangerous zone (Dang) over the block. Error bars show standard error mean, **p < 0.01.

A $2 \times 2 \times 4$ ANOVA (zone by group by block) on mean distance error for the object placement showed no zone by group by block, zone by group, zone by block, or group by block interaction (all $F$'s < 2, $p$'s > 0.05). However, there was a significant effect of zone ($F(1,49) = 6.47$, $p = 0.01$) and block ($F(3,147) = 18.48$, $p = 3e{-}10$). A post-hoc comparison of performance, of both groups, across test blocks, showed that distance error decreased from block 1 to block 4 (danger, $t(50) = 4.67$; safe, $t(50) = 5.10$, $p$'s = 0.00) reflecting improved spatial memory performance irrespective of whether objects had been located in the danger or safe zones of the flower task. Main effects of zone analysis revealed no significant difference in the HC ($F(1,27) = 2.84$, $p = 0.10$) between the error of objects placed in the danger and safe zone. A trend-level greater distance error in the safe zone was found for the ANX ($F(1,22) = 3.69$, $p = 0.07$). This trend-level effect in learning between zones for ANX was due a significantly greater distance error in objects found in the safe compared to the danger zone in block 1 ($t(22) = -2.61$, $p = 0.02$) but not in block 4 ($t(22) = -0.74$, $p = 0.46$; Fig. 2).

### fMRI measures of threat learning

*Threat appraisal (anxiety-state): approaching flowers (approach period) in the danger vs. the safety zone.* We use the approach period involving active navigation towards the flower to compare brain activation between groups as participants were navigating towards flowers. We used this measure as a proxy for anxiety-related responding to distal cues.

We compared brain activation (i.e., presumed metabolic activity) between diagnostic groups (ANX vs. HC) as individuals approached flowers located in the danger/safe zones of the environment (see Supplementary Table 1 for full results from this analysis). To assess discrimination learning, trials were divided into two blocks comprising the first (early) and the second half (late) of the experiment. As opposed to the behavioral results which used four blocks to assess learning, here we divided the data into two blocks, where the data was divided in half. In other words, the fMRI data comprised blocks 1 and 2 (early block) and blocks 3 and 4 (late block). This was done to increase the signal-to-noise ratio of the fMRI data when looking at learning over time and for consistency with a previous study[11]. To directly examine group differences, the first-level analysis contrasted factors of zone (safe vs. danger) and block (early vs. late), whereas the second-level analysis directly compared groups (group: ANX vs HC). Significant peak activation was extracted and analyzed to disentangle the directionality of the results. All statistical values

reported are FWE whole-brain corrected ($p < 0.05$). However, given our a priori hypothesis, additional FWE small volume correction (SVC) was performed when areas of interest (hippocampus, amygdala, and mPFC) did not survive FWE whole-brain correction. One bilateral mask, which included the hippocampus, amygdala, and mPFC, was used for the SVC analysis.

A zone by block by group interaction of approach periods identified two opposing patterns of activation changes in a range of areas comprising posterior cingulate cortex (PCC; $p < 0.05$ FWE), vmPFC, orbitofrontal cortex (OFC)/subgenual anterior cingulate cortex (sACC), and bilateral anterior hippocampus ($p < 0.05$ FWE SVC, Fig. 3a). Parameter estimates show that the ANX group, compared to the HC group, demonstrated a greater increase in activation in these areas from early to late blocks of the safe zone (late > early; safe > danger) and a decrease in activation from early to late blocks of the danger zone (early > late; danger > safe). To understand these distinct patterns of activation, we performed direct group comparisons on separate components of the task.

To understand how brain activation differed when approaching flowers in each of the zones (main effect of zone), we examined the zones contrasts (danger, safe) between groups. This analysis takes the average of the early and late regressor and looks at the zone effects. When approaching flowers in the danger zone, the ANX group (ANX > HC) showed greater activation of the bilateral insula ($p < 0.05$ FWE) and dmPFC ($p < 0.05$ FWE SVC; Fig. 3b) compared to the HC group. That is, these areas were more responsive in the ANX group, compared to HC, when approaching flowers in the danger zone. No group differences were observed when looking for areas that showed greater activation when approaching the safe zone.

To further understand brain activation differences as participants learnt the contingencies of the task (main effect of block), we examined the block contrasts (early, late) between groups. This analysis takes the average of the safe and danger regressor and looks at the block effects. For the HC group, compared to ANX (HC > ANX), approaching flowers in the second half of the experiment, compared with the first half (late > early) showed that, regardless of zone, there was increased activation from early to late blocks in the PCC, vmPFC, OFC, and anterior hippocampus ($p < 0.05$ FWE SVC). No other significant results were found ($p > 0.001$).

Given that dmPFC and insula activation was consistently higher in the ANX compared to the HC across the approaching period, particularly in the danger zone, we were interested to see

## a  Approach periods interaction

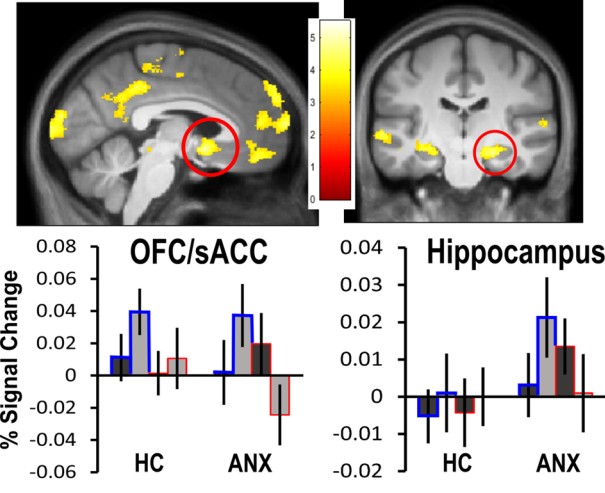

## b  Approaching danger periods

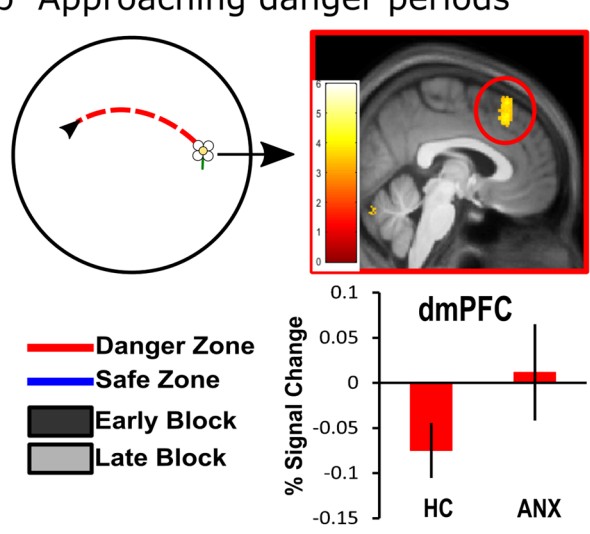

**Fig. 3 fMRI results of approaching flowers. a** Zone by block by group interaction shows two opposing patterns of activation between groups in the posterior cingulate cortex (PCC; *p* < 0.05 FWE), ventromedial prefrontal cortex (vmPFC), orbitofrontal cortex/subgenual anterior cingulate cortex (OFC/sACC; top left panel), and anterior hippocampus (*p* < 0.05 FWE SVC; top right panel). **b** Helicopter view of the circular environment that participants (black arrow) explored to approach the flower (red trace). For flowers in the danger zone, there was greater activation in the dorsomedial prefrontal cortex (dmPFC) across the whole test session in ANX compared to HC (*p* < 0.05 FWE SVC; lower right panel). All images are presented at *p* < 0.001 uncorrected for display purposes, not all clusters shown are significant at the whole-brain FWE-corrected level used outside of our ROIs. Percentage signal changes for danger and safety across early and late periods of learning extracted from **a** sACC (MNI coordinates: 5, 9, −11; left panel) and anterior hippocampus (MNI coordinates: −27, −17, −14; right panel); and **b** dmPFC (MNI coordinates: 9, 26, 45; right panel). Only a subset of percentage signal change graphs is shown to illustrate the pattern of activation, a similar pattern of activation was observed in the other relevant areas per contrast. Error bars show the standard error mean. *\*p* < 0.05 FWE SVC. For individual data points see Supplementary Fig. 1.

how the task and the physiological state it caused in the participants interacted with the brain activity to further understand brain–behavior associations of anxiety-states (threat appraisal). For this purpose, we used a psychophysiological

interaction (PPI) analysis for each participant group separately to identify dmPFC and insula (i.e., seed ROI) patterns in which connectivity changed during the danger vs. safe contrast. PPI examined the brain connectivity of the significant dmPFC (MNI coordinates: 9, 26, 45) and each insula side (MNI coordinates: (R) 41, 20, 3; (L) −44, 15, 0) from the approaching flower period. In the HC group, a positive association between the dmPFC and bilateral insula (*p* < 0.001 Bonferroni corrected) was found in the contrast (danger > safe). On the other hand, in ANX, a negative correlation between the dmPFC-OFC and dmPFC-vmPFC (*p* < 0.001 Bonferroni corrected) was found in the same contrast (danger > safe).

Overall, these results suggest that when approaching flowers in the dangerous zone, ANX showed reduced activation in the vmPFC, PCC, and anterior hippocampus while showing greater activation in the insula and dmPFC. These findings are further highlighted by negative correlation in activation between the dmPFC and vmPFC areas. On the other hand, HC displayed greater activation as a function of time in the vmPFC, OFC, PCC, and anterior hippocampus, regardless of the zone suggesting appropriate contextual learning.

*Threat anticipation (fear-state): held stationary in the danger vs. the safety zone.* We use this stationary period (entire 2–8 s) to understand brain activation differences between groups as participants were interacting with the flowers and anticipating the potential threat of shock delivery. We used this measure as a proxy for understanding fear associated with immediate cues.

We examined changes in brain activation when participants' positions were held stationary after picking flowers and anticipating a potential shock. We again examined the effects of the zone (danger vs. safe zone) and block (early vs. late blocks) in the first-level analysis. The second-level analysis consisted of the group comparison (ANX vs. HC; see Supplementary Table 2 for full results). Analyses followed the same model as for the approach period.

A zone by block by group interaction during stationary periods identified two opposing patterns of activation changes in a range of areas comprising PCC (*p* < 0.05 FWE), vmPFC, and OFC (*p* < 0.05 FWE SVC). Parameter estimates show that the ANX group, compared to the HC group, demonstrated a greater increase in activation in these areas from early to late blocks of the safe zone (late > early; safe > danger) and a decrease in activation from early to late blocks of the danger zone (early > late; danger > safe). To understand these distinct patterns of activation, we performed direct group comparisons on separate components of the task.

To understand how brain activation differed when interacting with flowers in each of the zones (main effect of zone), we examined the zones contrasts (danger, safe) between groups. This analysis takes the average of the early and late regressor and looks at the zone effects. When held stationary after picking a flower located in a zone of the environment associated with danger, the ANX group compared to HC (ANX > HC) showed greater activation in dmPFC, dACC, bilateral insula, caudate, thalamus, amygdala, and midbrain areas, including the periaqueductal gray (PAG; *p* < 0.05 FWE SVC; Fig. 4a). That is, these areas were more responsive in the ANX, compared to HC, when approaching flowers in the danger zone. For the HC group, compared to ANX (HC > ANX), flowers located in a zone of the environment associated with safety generated greater activation in the PCC (*p* < 0.05 FWE), vmPFC, OFC/sACC, and anterior hippocampus (*p* < 0.05 FWE SVC; Fig. 4b). That is, these areas were more responsive in the HC, compared to ANX, when approaching flowers in the safe zone.

To further understand brain activation differences as participants learnt the contingencies of the task (main effect of block),

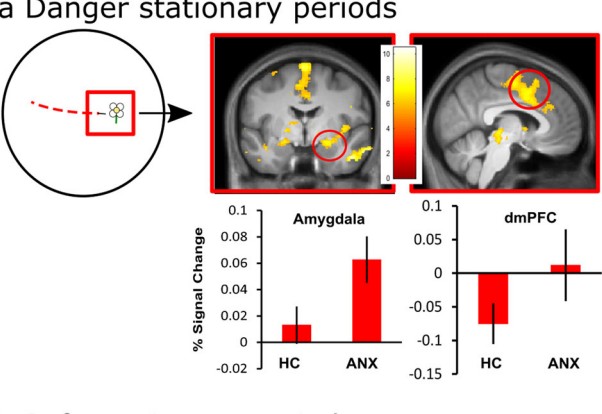

## a Danger stationary periods

## b Safe stationary periods

**Danger Zone**
**Safe Zone**
**Early Block**
**Late Block**

**Fig. 4 fMRI results of stationary periods.** Circular illustrations: Helicopter view of the circular environment that participants explored. The stationary period is represented for the dangerous flower as a red square and for the safe flower as a blue square. **a** The stationary period in the dangerous showed greater activation in the periaqueductal gray, dorsomedial prefrontal cortex (dmPFC), dorsal anterior cingulate cortex (dACC; middle panel), amygdala, and insula ($p < 0.05$ FWE; right panel) in ANX. **b** The stationary period in the safe zone showed greater activation in the posterior cingulate cortex (PCC; $p < 0.05$ FWE), ventromedial prefrontal cortex (vmPFC), orbitofrontal cortex/subgenual anterior cingulate cortex (OFC/sACC; lower middle panel), and anterior hippocampus ($p < 0.05$ FWE SVC; lower right panel) in HC. All images are presented at $p < 0.001$ uncorrected for display purposes, not all clusters shown are significant at the whole-brain FWE-corrected level used outside of our ROIs. Percentage signal changes during stationary periods for danger and safety across early and late parts of learning extracted from **a** amygdala (MNI coordinates: 26, −2, −15; middle panel) and dmPFC (MNI coordinates: 0, −8, 71; right panel) and **b** OFC (MNI coordinates: −3, 54, −17; middle panel) and anterior hippocampus (MNI coordinates: −32, −29, 12; right panel). Only a subset of percentage signal change graphs is shown to illustrate the pattern of activation, a similar pattern of activation was observed in the other relevant areas per contrast. Error bars show the standard error mean, **$p < 0.05$ FWE; *$p < 0.05$ FWE SVC. For individual data points see Supplementary Fig. 2.

we examined the block contrasts (early, late) between groups. This analysis takes the average of the safe and danger regressor and looks at the block effects. A group contrast (ANX > HC) of stationary periods (irrespective of danger or safety) showed increased dmPFC activation from early to late block (late > early) in ANX compared with HC ($p < 0.05$ FWE SVC). For HC, compared to ANX (HC > ANX), we found increased activation during the last half of learning (late > early) regardless of zone in vmPFC and OFC ($p < 0.05$ FWE SVC). No other significant results were found ($p > 0.001$).

Given that dmPFC, insula, and amygdala activation was consistently higher in the ANX compared to the HC across the stationary period, particularly in the danger zone, we were interested to see how the task and the physiological state it caused in the participants interacted with the brain activity to further understand brain–behavior associations of fear-states (threat anticipation). For this purpose, we used a PPI analysis for each participant group separately to identify dmPFC, insula, and amygdala patterns which connectivity changed during the danger vs. safe contrast. PPI examined the brain connectivity of the significant dmPFC, insula, and amygdala peak (i.e., seed ROI) from the stationary period. PPI analyses used dmPFC (MNI coordinates: 0, −8, 71), each insula side regions (MNI coordinates: (R) 41, −6, 0, (L) −39, 21, −5), and each amygdala side as seed regions (MNI coordinates: (R) 26, −2, −15, (L) −20, 2, −15). ANX showed increased functional connectivity between the dmPFC-bilateral insula, left amygdala-bilateral insula, and right amygdala-bilateral insula in danger compared to safe zones ($p < 0.001$ Bonferroni corrected). HC showed increased functional connectivity between the dmPFC-OFC, left amygdala-bilateral insula, right amygdala-bilateral insula, left amygdala-vmPFC, right amygdala-vmPFC, left amygdala-OFC, and right amygdala-OFC in danger compared to the safe zone during the stationary period ($p < 0.001$ Bonferroni corrected).

In summary, during the stationary period in the danger zone (after collecting a flower), ANX, compared to HC, demonstrated reduced activation in the vmPFC and PCC over time, with greater activation in the insula, amygdala, and PAG. This was further highlighted by increased connectivity among the dmPFC, amygdala, and insula while lacking any significant connectivity from vmPFC areas. On the other hand, HC recruitment of the vmPFC, OFC, and PCC was seen as a function of time, where those areas were recruited during the stationary periods in both the safe and dangerous zones. Furthermore, HC had significant connectivity of the vmPFC and OFC to areas such as the dmPFC and amygdala.

**Approach periods: differences between approaching flowers and object locations.** As a control for movement and to understand participant's approach to cues (emotional flowers vs. unemotional objects), brain activation changes were analyzed between groups when performing the two tasks. Approaching flowers during the threat learning task was collapsed across danger and safe conditions. Object-approach periods consisted of only trials when participants were instructed to collect the object and remember its spatial location (i.e., omitting object replacement trials). See Supplementary Table 3 for full results from this analysis. Each period began at trial onset, when the flower or object appeared in the environment, and ended when that flower or object was "collected."

The analysis was conducted after excluding the initial orienting period of the approach period focusing on the last 75% of the approach period (active navigation). To assess differences in learning between the flower and object task, trials were divided into early (first half of the experiment) and late blocks (second half of the experiment). The analysis was conducted on the full sample and in the same fashion as the approach and freezing period, which consisted of a first-level analysis ($2 \times 2$ ANOVA) with two within-subjects factors (task: object, flower; block: early, late). The second-level analysis consisted of a the between-subjects factor (group: ANX, HC). A task by block by group interaction of approaching cues identified the PCC ($p < 0.05$ FWE), OFC/sACC, and anterior hippocampus ($p < 0.05$ FWE SVC; see Fig. 5a). When approaching objects (objects > flowers; early > late) ANX showed increased PCC activation, while HC

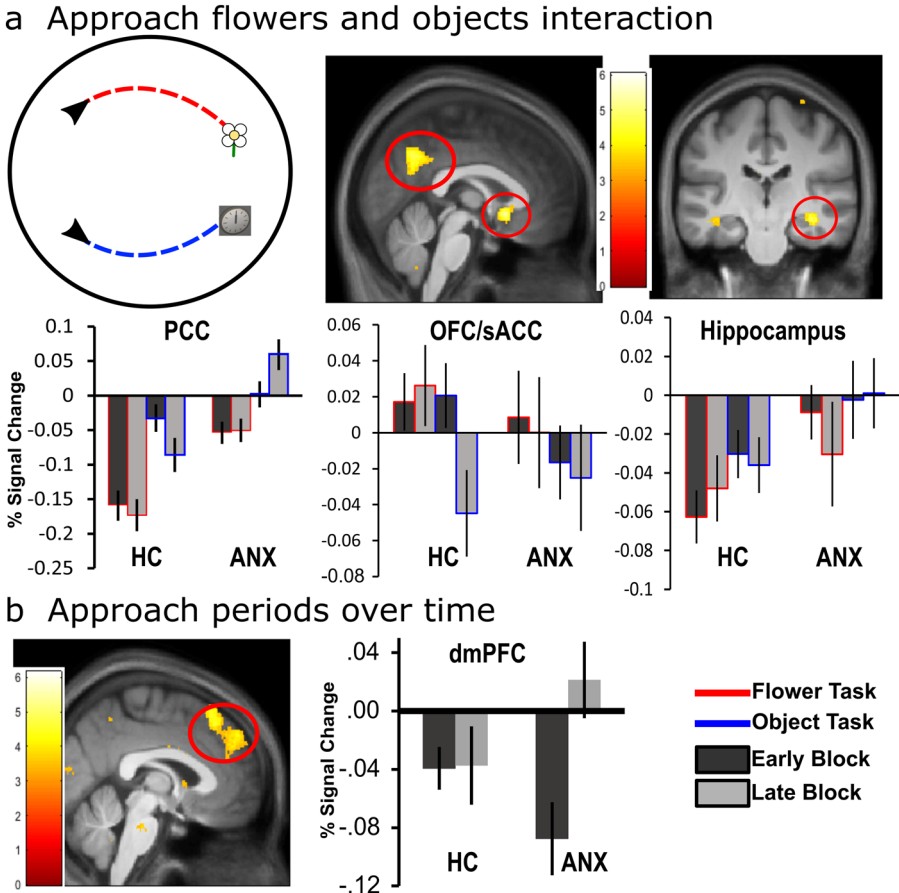

**Fig. 5 Activation differences between approaching flowers and objects during threat and spatial memory. a** Circular illustration (left panel): helicopter view of the circular environment that participants (black arrow) explored and approach to emotional flower (red trace) and approach to unemotional object (blue-trace). Percentage signal change of task by block by group interaction when approaching flowers compared to objects in a range of brain areas, including the posterior cingulate cortex (PCC; $p < 0.05$ FWE; middle panel), orbitofrontal cortex/subgenual anterior cingulate cortex (OFC/sACC; $p < 0.05$ FWE; middle panel), and hippocampus ($p < 0.05$ FWE SVC; right panel). **b** Activation change was greater from the first to the second half experiment (late > early) in the dorsomedial prefrontal cortex (dmPFC) in ANX ($p < 0.05$ FWE SVC; left panel). All images are presented at $p < 0.001$ uncorrected for display purposes, not all clusters shown are significant at the whole-brain FWE-corrected level used outside of our ROIs. Percentage signal changes for learning about threat and object locations across early and late periods of the task extracted from **a** PCC (MNI coordinates: −2, −63, 21; left panel), sACC (MNI coordinates: 0, 14, −8; middle panel), hippocampus (MNI coordinates: 23, −20, −14; right panel), and **b** dmPFC (MNI coordinates: 0, 29, 53; middle panel). Only a subset of percentage signal change graphs is shown to illustrate the pattern of activation, a similar pattern of activation was observed in the other relevant areas per contrast. Error bars show standard error mean, **$p < 0.05$ FWE, *$p < 0.05$ FWE SVC. For individual data points see Supplementary Fig. 3.

showed decreased activation, from early to late blocks. Additionally, while ANX showed a general OFC/sACC activation in both tasks, HC showed an increased activation for the flower task and decreased activation for the object task. On the other hand, when approaching flowers (flowers > objects; early > late), ANX showed decreased anterior hippocampus activation, while HC showed increased activation, from early to late blocks.

To further understand brain activation differences as participants learnt the contingencies of the tasks, we examined the block contrasts (early, late) between groups. This analysis takes the average of the flower and object tasks regressor and looks at the block effects. During the second half of the experiment (late > early), the ANX (ANX > HC) showed increased dmPFC activation from the early to the late blocks irrespective of the task ($p < 0.05$ FWE SVC; Fig. 5b). No other contrast revealed significant results.

### Discussion
The present study examined how patients with pathological anxiety learn to distinguish contextual features that inform the

dangerous vs. safe status of stimuli within a virtual environment. This virtual-environment task shows how brain regions interact to shape behavior over time. We hypothesized that patients with pathological anxiety would show (1) poor threat learning associated with lower recruitment of the anterior hippocampus; (2) biased threat appraisal with the stronger engagement of the dACC, dmPFC, amygdala, and insula (higher anxiety-state); and (3) higher threat anticipation with the stronger engagement of the PAG (higher fear-state).

Surprisingly, inconsistent with our hypotheses, patients with pathological anxiety showed no impairment in discriminating between threat and safety within the environment based on their skin conductance, subjective reports, and navigation time (Supplementary Information). However, consistent with our hypothesis, patients with pathological anxiety, compared to HC, displayed (1) weaker engagement of the anterior hippocampus and vmPFC, areas implicated in memory processing and emotional regulation, during both approach and stationary periods (Threat learning and discrimination); (2) higher engagement of the dACC, dmPFC,

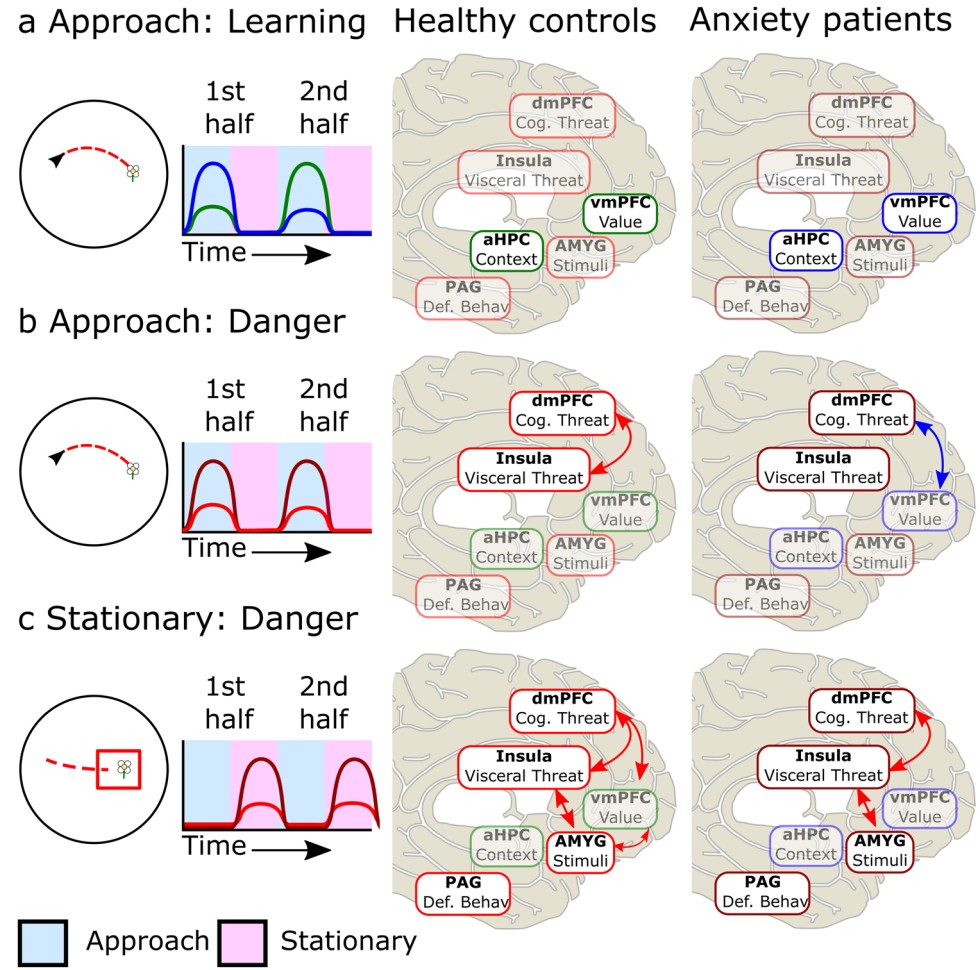

**Fig. 6 Illustration of sequential network activation in the flower task between groups.** Left panel: Helicopter view of the task phase and schematic of activations over blocks. Middle and right panels: Brain activation and functional connectivity of HC (middle) and ANX (right). Green lines/boxes represent activation (and red arrows represent functional connectivity) that increased from the first to second half of the experiment. Blue lines/boxes represent activation (and blue arrows represent functional connectivity) that decreased from the first to second half of the experiment. Red lines/boxes represent activation (and red arrows represent functional connectivity) that increased with danger, with darker red representing higher activation. **a** For HC, during the dangerous flower approach (left panel), activation in the anterior hippocampus (aHPC) and ventromedial prefrontal cortex (vmPFC) increased in the late phase compared to the early phase of learning (middle panel), while it decreased for ANX (right panel). **b** For HC, during the approach of flowers predicting danger (left panel), activation in the insula and dorsomedial prefrontal cortex (dmPFC) showed positive connectivity (middle panel). For ANX, compared to HC, higher activation was evident in the insula and dmPFC, and there was a negative connectivity of dmPFC-vmPFC (right panel). **c** For HC, when danger was imminent during the stationary period (left panel), the insula-amygdala (Amyg), insula-dmPFC, vmPFC-dmPFC, and vmPFC-Amyg were positively connected (middle panel). For ANX, compared to HC, higher activation was evident in the dmPFC, insula, amygdala, and periaqueductal gray (PAG), and there was positive connectivity in the insula-dmPFC and insula-Amyg. See Supplementary Tables 1–3 for a complete breakdown of regions across these analyses.

amygdala, and insula involved in negative valence, emotional expression, and conflict processing during both approach and stationary periods (Threat appraisal); and (3) higher engagement of midbrain areas, including the PAG, during anticipation of potential shock delivery (Threat anticipation; see Fig. 6). These results suggest that patients with pathological anxiety recruit compensatory learning strategies that do not involve the anterior hippocampus or vmPFC, which might explain the increased activation in the salience network (e.g., dmPFC, insula, PAG). In this task, the participants were forced to collect the flowers and therefore got the opportunity to know the contingencies of the zones. Real-life scenarios don't often offer that possibility, particularly if the patient prefers to avoids it, creating higher uncertainty and risk. We believe that reintegrating the anterior hippocampus and vmPFC into the circuitry could rescue appropriate learning strategies to discriminate between safety and danger, particularly when information and experience is limited.

Several points are noteworthy when considering the current findings in light of the previous study in healthy controls using the same task[11]. In both task, we have identified the same network of activation in HC's when approaching and interacting with flowers. We replicated the two neural networks to support the formation of cognitive maps and its relevant behavior: (1) The discrimination learning network (DLN; vmPFC, PCC, anterior hippocampus) activation increased when HC's learn to discriminate threatening and safe zones within the environment; (2) the salience network (SN; dmPFC/dACC, insula, PAG) activation increased when participants approached and interacted with flowers in the danger zone. There is one main difference between this and the past study in the SN, in the current study the peak activation was more in the dmPFC while in the previous study the peak was more in the dACC. However, the brain activation cluster of both studies often overlap between the two areas (dmPFC/dACC). Furthermore, while some brain activation

differences found in the current study are using a priori hypothesis SVC based on the past study, all the group brain activation differences (HC, ANX) of the current study are in the areas identified in the previous study (DLN, SN).

We observed changes in activation in several brain regions as the experiment progressed (first vs. second block) that differed between patients with pathological anxiety and healthy controls. Healthy controls showed increased activation over time for regions involved in emotion regulation (ventromedial prefrontal cortex; vmPFC), valuation (orbitofrontal cortex; OFC), spatial memory (hippocampus) and self-referential processing (posterior cingulate cortex; PCC) regardless of safe/danger zone. In patients with pathological anxiety, these brain areas (vmPFC, OFC, hippocampus, and PCC) differentiated between zones, with increased activation in the safe zone and decreased activation in the danger zone over time. The anterior hippocampus is essential for the integration of spatial information in mediating anxiety-like behaviors[25]. As seen in the current study with healthy controls, previous findings show that the hippocampus is involved in learning and discrimination of safe and danger zones, with increased activation with task experience[11]. Therefore, decreased engagement of the anterior hippocampus might reflect a compensatory mechanisms, whereby patients with pathological anxiety use alternative learning strategies that do not involve context-specific cues for learning the environment contingencies. These results are in line with research showing hippocampal and medial prefrontal cortex abnormalities in patients suffering from pathological anxiety[26,27] who report an exaggerated response to threats in contexts predicting safety. Reduced activation of the anterior hippocampus, particularly during potential or perceived threat, could impair emotional regulation abilities in novel contexts or when shifting contexts.

Patients with pathological anxiety displayed reduced activation in the vmPFC, OFC, and PCC throughout the flower task, particularly when interacting with flowers in the danger zone during the late blocks. These vmPFC/OFC areas are considered to have key roles in regulating contextually appropriate emotions[28–31]. Previous studies reported that in healthy individuals, vmPFC, PCC, and anterior hippocampus activation did not differentiate danger and safety during this flower task but rather increased with experience (from early to late block of the experiment), suggesting a role in learning and discrimination[11]. In the present study, patients with pathological anxiety, unlike healthy controls, demonstrated increased engagement for safe flowers and decreased engagement for dangerous flowers. Given the role of the anterior hippocampus and vmPFC/OFC, the hypoactivation observed in patients with pathological anxiety may suggest that they are unable to use contextual cues to regulate their emotional output, particularly in a potentially dangerous location.

As the vmPFC and PCC are considered part of the classic valuation network[32–34]—the brain network implicated in assessing the subjective valuation of stimuli—our findings could be explained also in terms of the safe zone being more valuable than the danger zone. However, based on a previous study in healthy adults[11] and this study's healthy control group, we see that activation in the vmPFC and PCC, along with the anterior hippocampus, increase over time as participants learn to discriminate between zones. That is, the highest activation in these areas is evident when they have learned the contingencies and rules of the environment, regardless of zone, as there is no zone difference in these areas. A zone difference surfaces only when we look at patients with anxiety disorder, who show an increase in activation over time in the vmPFC, PCC, and anterior hippocampus for flowers in the safe zone, while a decrease over time for flowers in the danger zone. This interaction in turn could be interpreted as suggesting that patients with anxiety disorders have

a higher confidence of value for the safe zone compared to the danger zone. Indeed, activation in the vmPFC and PCC areas has been associated with increased confidence of decision value[35], which could be associated with higher discrimination learning[36]. Additionally, we see that salient network areas activation (insula, dmPFC, and PAG) increase over time, particularly for the danger zone, suggesting an increase in negative valuation confidence. Therefore, we believe that this system takes over in the danger zone, overriding or dampening the activation of the vmPFC and PCC in patients with anxiety disorders. On the other hand, in healthy controls, this vmPFC and PCC network activity could be supporting valuation confidence as it increases equally in both zones as participants learn the environmental contingencies.

Compared to healthy controls, patients with pathological anxiety demonstrated greater activation in brain areas involved in threat appraisal (dmPFC, dACC, amygdala, insula) throughout the whole experiment, particularly in the danger zone. Greater engagement of these brain areas likely reflects higher arousal when interacting with the flowers, particularly in the danger zone. Activation in the dmPFC, along with the amygdala and its connectivity, has been associated with threat arousal[37–40] and may play a key role in integrating threat information to coordinate emotional responses with other brain regions. Increased dmPFC activation to threat stimuli has been documented in patients with pathological anxiety compared to healthy controls[28,29,38,39], suggesting an overactive appraisal of surroundings and cues. Accordingly, dACC and insula activations have been reported during the approach of threat in healthy adults using the flower task[11], likely reflecting the integration of visceral feelings and cognitive appraisals of threat to trigger fear expression[41]. However, in this study, patients with pathological anxiety showed greater dmPFC, dACC, and insula activation throughout the task, possibly reflecting an overactive threat appraisal and detection network.

During imminent threat, patients with pathological anxiety exhibited greater activation in brain areas related to error detection, conflict resolution, and emotional expression (dACC and amygdala) and regions related to salience and pain judgment (bilateral insula and PAG). Midbrain areas, including the PAG and amygdala, have been reported during imminent threat in healthy adults using the same task[11], likely reflecting the integration of cognitive appraisals to the emotional anticipation of threat to trigger appropriate fear expression[41]. However, in this study, patients with pathological anxiety, compared to healthy controls, showed greater PAG and amygdala activation during imminent threat, possibly reflecting an overactive threat anticipation. Particularly increased PAG activation during stationary periods in the danger zone might suggest greater anticipation of the potential shock just prior to its likely delivery.

Patients with pathological anxiety failed to exhibit task-related functional connectivity with higher-level cortical areas related to emotional regulation (vmPFC and OFC) throughout the task. During the flower approach, patients with pathological anxiety displayed negative functional connectivity between the dmPFC and both the vmPFC and OFC. During stationary periods in the danger zone, patients with pathological anxiety displayed positive dmPFC-insula and amygdala-insula functional connectivity. On the other hand, healthy controls, displayed functional connectivity of the dmPFC to the OFC and vmPFC. And while groups were not directly compared, these findings suggest that ANX and HC are engaging in different networks of brain activity when engaging with the task. Coupling between the dmPFC and amygdala has been previously observed with the induced threat of shock[28,42,43] and in patients with pathological anxiety[44] during an emotional identification task. In pathological anxiety, this circuit may become hyperactive and unable to "turn off," contributing to

an attentional bias towards threat. In this study, functional connectivity deficits with vmPFC/OFC areas in patients with pathological anxiety could reflect a faulty flexible learning spatial strategy. For example, patients with pathological anxiety may fail to integrate the learned contingencies within the environment and surrounding landmarks to create a spatial representation that can modulate emotional output to stimuli while navigating through the safe and danger zones. Therefore, an inflexible or exaggerated evaluation of the cue may hinder the use of spatial strategies to appropriately regulate emotional output. In other words, patients with pathological anxiety might have shock-related worry that gets progressively worse as they approach and pick up the flower and as the experiment progresses.

It is important to note that while ANX displayed a higher state anxiety that HC, particularly at the end of the task, it does not provide evidence that the task induced any changes in anxiety. This finding was further supported by the skin conductance results during the task (SCL; SCR), which were only significant when we explored the groups separately. Exploratory analysis did suggest that HC were showing skin conductance (SCL, SCR) differences between the zones, while in patients this was only evident at a trend-wise level for the SCL. However, as an exploratory analysis, this should be taken with caution as our main analysis revealed no interaction effects, and particularly as both patients and HC participants learned the task's contingencies (expectancy ratings). We can speculate that there are indeed skin conductance differences reflecting that patients learnt the contingencies appropriately through brain compensatory mechanisms. But, maybe these behavioral measures are not sufficient to capture the full spectrum of behavior between groups. While we did not find spatial trajectories or time spent in each zone differences between groups this could be due to several factors of the task, (1) participants were forced to collect one flower at a time; (2) the flowers appeared randomly throughout the environment, therefore, they could be close or far from each other; (3) the interaction of age, gender, and video game experience might have affected the task proficiency in general. And, we did not assess the video game experience of each participant to fully flush out these results. Future studies should consider using other sorts of behavioral measures (e.g., eye-tracking), increasing the stakes of the shock contingency (e.g., correct expectancy ratings diminish the risk of shock), gaining more insight into participants' video game experience, or increasing the sample size to increase power and therefore flush out more the exploratory findings. Another component that might have affected the behavioral results is the inclusion of patients with a social anxiety disorder (SAD), as the task did not include any social component. The study aimed to capture differences between HC and patients with any anxiety disorder. And while the study only recruited a minimal number of patients with SAD (7), their inclusion might have skewed the results more favorably to appropriate learning behavior. Future studies should focus on patients with a generalized anxiety disorder to test this.

In conclusion, patients with pathological anxiety show lower activation in the anterior hippocampus and vmPFC/OFC and lower connectivity to emotion expression brain areas (dmPFC, amygdala, and insula). In this task, participants were forced to collect a flower, there was no avoidance or multiple options available. Unlike real-world scenarios where we might opt out of experiencing something, here participants could not, forcing on them the opportunity to learn about the flowers. Therefore, while patients seemed to learn to discriminate between zones, lack of hippocampus and mPFC (particularly within the danger zone) could explain why patients with anxiety disorders often show heightened emotional responses when they are in a novel environment or situation and are not able to accurately identify a potential threat. This possibility is further supported by the observed higher dmPFC activation, and other emotion valuation and expression brain areas, during the task. The current results suggest that a disconnected circuitry in brain areas essential for memory and context processing and emotional regulation might lead to disrupted emotional output without extensive exposure. That is, while exploring unfamiliar environments and learning about cues within these environments, patients with an anxiety disorder might find it difficult to regulate their anxiety and opt out of the experience. Finding novel psychological interventions or training to reintegrate the vmPFC and hippocampus into the learning and discrimination circuitry could help patients with pathological anxiety regulate their emotional responses while navigating novel environments. This might allow patients more exposure time to learn whether they are safe or not.

## Methods

**Participants.** Sixty participants, aged 18–50 years, were recruited from the Washington D.C. and Maryland areas. Thirty participants were diagnosed with an anxiety disorder (ANX; generalized anxiety disorder ($n = 8$), social anxiety disorder ($n = 7$), and co-morbidity between the two ($n = 8$)) using the Structured Clinical Interview for Diagnostic and Statistical Manual of Mental Disorders 4th edition, while the other 30 volunteers were healthy controls (HC). Before taking part, all participants provided written informed consent and, after completion, were debriefed and reimbursed for their time. The study was approved by the NIH Institutional Review Board. All participants were right-handed, free from neurological impairment, or any psychological disorders (except for an anxiety disorder in the patient group). Four ANX were excluded from analyses because they were unable to explain the shock contingencies between the locations at the end of the task (see procedure below). Three ANX and two HC were omitted due to excessive head motion (>20%; see below). Therefore, the final sample included 23 ANX (mean age = 29.61; SD = 8.22) and 28 HC (mean age = 27.25; SD = 8.21). Participants were free of medications, and no significant differences ($p$'s > 0.05) in any demographic information between groups were found. See Supplementary Information for a demographic breakdown (Supplementary Table 4).

**Skin conductance.** Skin conductance was measured as an index of anxiety via 8 mm Ag/AgCl electrodes attached to the medial phalanges of the index and middle fingers of the participant's left hand. Data were acquired using a Biopac EDA100C MRI system (Biopac Systems, Inc., Goleta, CA, USA) at a sampling rate of 1000 Hz.

**Shocks.** Shocks (20 samples) were applied using a Digitimer DS7A electrical stimulator (Digitimer, Welwyn Garden City, UK) to the left hand with intensity up to 50 mA for 2 ms duration through a silver chloride electrode. Shock intensity was adjusted individually for each participant before starting the experiment. Individual adjustment procedures delivered a series of shocks to each participant, starting at 12 mA. Participants were asked to rate the level of pain with each shock on a 1–10 scale. Shock intensity was increased until the level was irritating but not painful.

**Task.** A description of the virtual-environment task, developed with Unity Software (Unity Technologies, USA), is available in a prior publication[11]. The virtual environment consisted of a circular grassland with a perimeter boundary wall surrounded by distal cues (mountains, sun, and clouds) for orienting, two landmarks (beehives) placed in the grassland, and flowers that appeared one at a time in random locations. When participants picked a flower, they rated on a scale from 0 to 9 the likelihood of receiving a shock from the flower picked. Then, the virtual character would enter a stationary period for 2000–8000 ms. After the stationary period, participants could move once again, and another flower would appear in the environment. The task included a total of 80 flowers—40 flowers were paired with a shock on 50% of the trials (danger) while encountering no shock with the remaining 40 (safe).

After every four flower trials, we included a spatial memory trial in which participants learned the location of one of four objects (wooden box, gas can, book, and clock); however, two objects appeared in each half of the environment. No shocks accompanied spatial memory stimuli. After the first four spatial memory trials, participants' memory for object locations was tested by asking participants to replace the objects where they originally found them.

At the end of the experiment, participants were asked to name the four objects and their locations, as well as explain the contingencies of danger and safety during threat learning, by answering if there was a pattern in the shocks. Participants who were unable to provide the objects' name and position or explain the contingencies were excluded from the final analysis to ensure that participants were paying attention to the task.

**fMRI acquisition**. Blood oxygen level-dependent T2*-weighted functional images were acquired on a 3T Skyra system (Siemens, Germany) using echo-planar imaging (EPI) with a 32-channel head coil. Images were acquired with a 45° oblique angle with the following parameters: 3300 ms TR; 30 ms TE; 1 mm inter-slice gap, 192 mm field of view, and 48 axial slices with 2 mm slice thickness resulting in 3 mm isotropic voxels. A single echo field map was recorded for distortion correction of the acquired EPI. After the functional scans, a T1-weighted 3-D structural image (1 mm³) was acquired to coregister and display the functional data.

**Statistics and reproducibility**
*Behavioral analysis*. Data processing and analysis of electrodermal activity (EDA) were performed using MATLAB. EDA data were down-sampled to 200 Hz and then synchronized to the task. EDA was assessed during two periods of the threat learning task. First, the mean skin conductance level during each approach quantified tonic skin conductance levels (SCL) as participants navigated towards the flower. SCL was quantified from the last three-quarters of the approach period from flower appearance until trial completion. The skin conductance level was calculated by measuring the mean skin conductance from the beginning of the active approach until right before the flower was picked for each trial. Second, skin conductance responses (SCR) were analyzed during the stationary period (entire 2–8 s) to examine phasic changes in anticipation of the shock outcome. SCRs were calculated for every trial by subtracting the minimum skin conductance during the stationary period (baseline) from the maximum response (peak) before the stimulus onset. Any response difference under 0.03 micro-Siemens was scored zero. SCRs were log-transformed (log [1 + SCR]) to normalize the distribution, and then range correction ([SCR-SCRmin]/[SCRmax-SCRmin]) was applied to control for individual variation in response[45]. The same correction was applied to the SCLs. For analyses, SCRs and SCLs were averaged into four equal blocks across the duration of the experiment, with each block including ten trials per condition (safe and danger).

Expectancy ratings taken at the beginning of each stationary period were analyzed similarly to skin conductance. Each rating (0–9) was averaged across trials to create four equal blocks separated by safe and danger conditions (10 trials in each block).

Finally, performance on the spatial memory task was analyzed by assessing distance error on each test trial. This distance error was calculated by taking the distance in virtual meters between the participant's response location when replacing the object and its correct location within the environment. Distance error was taken from each trial and averaged into four blocks (1 trial from each object in each block).

All results were analyzed using a general linear model (GLM) for repeated measures using $2 \times 2 \times 4$ ANOVAs to test differences between zone (safe, danger), group (HC, ANX), and block (1–4). Based on an a priori hypothesis of ANX overgeneralization[12], the simple effects within each group were examined using $2 \times 4$ ANOVAs to characterize differences between conditions (safe, danger) and block (1–4).

**fMRI analysis**. Data processing and analysis were performed using SPM12 (http://www.fil.ion.ucl.ac.uk/spm). EPI images were first preprocessed using a bias correction to control for within volume signal intensity difference, unwarping and realignment to correct for movement and slice-time correction. Images were then spatially normalized to the MNI template using parameter estimates from warping each participant's structural image to a T1-weighted average template image. All images were finally smoothed using an 8 mm FWHM Gaussian kernel.

The analysis model included 15 regressors of interest. Four separate regressors of interest were created for approach periods by zone (safe or danger) and block (first or second half of experiment); these consisted of boxcar functions from the end of the first quarter of each approach period to the point in which the flower was reached. Four regressors of interest (also by zone and block) were created for the stationary period of each trial, consisting of a boxcar function starting after the participant rated their shock expectancy and lasting for the duration of the stationary period. Three regressors using a stick/impulse function modeled the end of each trial in safe conditions and in danger conditions split by whether participants received a shock or not. Finally, four regressors were modeled when participants were replacing objects during the spatial memory task using a boxcar function covering when a response was made and the approach period to the location where the object was picked (first and second half of the experiment). Six regressors of no interest were also added to the model representing movement parameters estimated during realignment. Frames with more than 0.5 mm frame-wise head motion were detected as outliers and modeled using the Artifact detection tool (ART). Participants with outliers totaling 20% or more of their total scan were removed from the analysis.

Statistical analyses occurred in two stages, individual (general linear model analysis) and group level (two-sample *t*-test). At the individual level, an initial control analysis was conducted to examine emotional flower vs. unemotional object approaches to a cue. The neural responses to approach periods were compared between the threat learning task (approaching flowers) and the spatial memory task (approaching location to replace the object; see Supplementary Information). We created a model contrasting approach periods for threat learning (collapsing across safety and danger) with an approach during spatial memory across the first and second half of the experiment using a $2 \times 2$ ANOVA (task, block). Approach periods during threat learning were then

examined by contrasting approach to flowers associated with safety or danger and collected during the first or second half of the experiment using a $2 \times 2$ ANOVA (zone, block). Finally, stationary periods during threat learning, periods when the flower was picked, and participants were held stationary were analyzed in a similar first-level analysis using a $2 \times 2$ ANOVA (zone, block).

At the group level, the contrasts of interest, described above, were entered into two-sample *t*-tests to compare the ANX and HC. Within this model, *t*-contrasts were examined to investigate ANX > HC and HC > ANX differences. To aid and confirm the interpretation of significant group effects, parameter estimates were extracted from significant peak coordinates for the contrasts ANX > HC and HC > ANX using the MarsBaR region of interest toolbox. Post-hoc analysis of these contrasts was conducted using repeated measures $2 \times 2 \times 2$ ANOVA (zone or task, block, group) in SPSS.

Family-wise error (FWE; $p < 0.05$) corrected effects across the whole brain are reported for all analyses. Given the a priori hypotheses and previous findings[11], an ROI approach was employed based on the Automated Anatomical Labeling atlas (AAL)[46] and the WFU Pickatlas toolbox in SPM12[47]. One bilateral mask was defined which included the hippocampus, amygdala, and mPFC (orbitofrontal gyrus, ventromedial prefrontal cortex, dorsomedial prefrontal cortex, and anterior cingulate and medial cingulate cortex). Statistical threshold in this single mask was defined by small volume correction (SVC; $p < 0.05$ FWE).

For any significant interaction, the representative time-course was extracted through SPM12 MarsBaR (http://marsbar.sourceforge.net) toolbox, using a 6 mm sphere centered on the peak of the activation in the regions of interest, using the first eigenvariate calculated from singular value decomposition. The extracted values were used to illustrate and observe the directionality of the results.

"Activation" was used throughout the manuscript to indicate an increase in presumed metabolic activity.

**Functional connectivity analyses**. Functional connectivity was assessed at the individual level using psychophysiological interaction (PPI) analysis with SPM12. The PPI compares functional connectivity from a single seed region across multiple task conditions. For exploratory analyses, seed regions were selected based on group differences identified in the main analysis (dmPFC, insula, amygdala). Peak activation from these brain areas in the group level analysis, for approach and stationary periods, was used to create regions of interest (6 mm sphere centered on group level peak activation) for each participant. The seed time series activation for each participant was extracted at the center of the activation peak. The individual *t*-contrast images of the interaction from the PPI were examined using a group level one-sample *t*-test. As in a previous study[11], and to increase the strength and precision of the analysis, the PPIs were detected using a *t*-test with a threshold of $p < 0.001$ corrected for multiple comparisons.

**Reporting summary**. Further information on research design is available in the Nature Research Reporting Summary linked to this article.

## Data availability
The data sets generated during and/or analyzed during the current study are available in neurovault (https://identifiers.org/neurovault.collection:10908) and figshare (https://figshare.com/s/41ea63a25a2244084c9e).

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

## Acknowledgements

We thank Dr. Amit Lazarov for his valuable and constructive advice during the review process of the manuscript. His willingness to give his time so generously has been very much appreciated. This research was funded by the National Institute of Mental Health (NIMH) Intramural Research Program, Medical Research Council and the Wellcome Trust, United Kingdom. The authors are grateful to the NIMH Center for Multimodal Neuroimaging at Bethesda, Maryland and UCL Welcome Trust Centre for Neuroimaging in London, UK for providing facilities and services. We thank the NIMH-UCL Graduate Partnership Program, NIMH T32 grant MH015144, and NIMH K01 MH118428-01 for providing support to B.S.-J.

## Author contributions

B.S.-J. designed the experiment with contributions from J.A.K., D.S.P., C.G., N.B., and M.E. B.S.-J. programmed the VR task. B.S.-J. collected the data with contributions from N.L.B., J.C.L., and A.H.; B.S.-J. analyzed the data with contributions from N.L.B., J.A.B., N.B., and M.E.; B.S.-J., J.A.B., D.S.P., C.G., N.B., and M.E. interpreted the data. B.S.-J. wrote the manuscript with contributions from all authors.

## Competing interests

The authors declare no competing interests.
