## [Transparent Peer Review File · Communications Biology]

Reviewers' comments:

Reviewer #1 (Remarks to the Author):

In this fMRI study, Suarez-Jimenez and colleagues compare activity in approaching a threat vs. a neutral object between participants with generalised anxiety or social anxiety and control participants. The authors present a number of areas in which activity changes between participant groups, although these differences did not seem to translate into meaningful behavioral differences.

The paper is well written, but its structure could be clearer: what question the paper addresses, why this design, what is the reasoning behind this analysis, and so on. I have also a few concerns about the methodology that I develop below. I hope that the authors will find my comments helpful.

Important comments

The behavioral analyses were very succinct as compared to the richness to the experiment (only the ratings of shock expectancy are analysed).

- What about the participants' spatial trajectory in the environment? Could the authors analyse the mouse cursor trajectories for instance?
- Do participants spend more time in either zone? Do participants with anxiety spend more time in the safe zone as compared to the control group?
- Do participants with anxiety pick up fewer flowers overall? Or does the experiment stop until they have picked all flowers, and if so, did participants with anxiety take longer to complete the experiment?
- Could the authors find basic behavioral signatures of more avoidance in participants with anxiety, for instance exploring for longer before picking up any flower?

- I could not understand if analyses were ROI-based or whole-brain. We are told that all statistical values reported throughout the paper are FWE whole-brain corrected, but also that a ROI-based analysis was employed based on ROIs from the authors' previous study. I note that the z-score values reported in supplementary table makes it more likely that the latter is employed. It is important that the authors clarify what has been done.

- Why for Fig 2B dmPFC the split first half/second half was not used? I appreciate that the authors sought to decompose learning periods, but it would be great to better justify the time split, because the results section jumps back and forth from analysing all trials to analysing first half/second half to analysing block 1 to block 4. Similarly, the focus on first half vs. second half of the experiment seems post hoc? (as indicated by the authors, to "increase SNR"). Does the finding that "the ANX group demonstrated a greater increase in activity in these areas from early to late blocks (late > early) of the safe zone, and a decrease in activity from early to late blocks of the danger zone" (p. 8) average out when looking at the whole experiment?

- The authors dissociate stimuli (same in all zones) from context (different). But all these are visually displayed on the screen and no hierarchy exists between them, so could we reframe this as stimulus-response mappings learnt, not contextual information? Sure the flowers are the same, but the flowers embedded in context A or B could be re-labelled as a unique stimulus A or B. Relatedly, the authors say that people with anxiety generalise defensive response to CS-safe cues, which is different from a context. Do the authors base their claim that context conditioning requires the use of spatial processing strategies? (Not all contextual cues are spatial)

- Participants were a blend of Generalised Anxiety Disorder and Social Anxiety Disorder that are fairly different. Does the nature of the disorder influence the findings? Was there a particular motivation for recruiting social anxiety patients in a paradigm involving no social component? Could the authors provide a breakdown of how many with each disorder were recruited? Could they provide a breakdown

(at least) of the main effects?

- The spatial memory task null findings are reported in the Supplement would deserve to appear more prominently: it is important to know that all participants were able to learn the placement of objects, and that they similarly learnt in the safe and danger zones, and that control and anxiety participants also similarly learnt

- Could the findings be simply explained in terms of the safe zone being more valuable than the danger zone (that by design has more punishment), the classic valuation network being observed in this contrast (VMPFC/PCC)?

- Could the authors provide references supporting the first hypothesis about poor learning and discrimination in anxiety? Especially as the authors happen to find no evidence for this hypothesis in the end.

- No occipital activations are reported in any of the tables though some are visible e.g. Fig 2; could the authors clarify?

- A number of details were missing as compared to what is standard in the field. Could the authors report (i) how many participants were excluded for excessive motion; (ii) The authors excluded participants who in a debriefing were not able to report the task contingencies. How many? (iii) We are told that participant groups did not differ in terms of e.g. demographic variables, but statistical values should be reported to support these statements.

- We are told that the question is to "elucidate neural mechanisms of anxiety" which says very little about why the experimental design was set up the way it is. Could the authors clarify the specific question and specific motivation for their study?

- The conclusion indicates that participants with anxiety exhibit a heightened reactivity to threat; isn't it the very definition of anxiety? Could the authors make the specific contributions of their study more explicit?

- The neural differences observed did not seem to translate into behavioral differences. If anything, the differences between danger and safe zones are overall smaller in anxiety patients than controls (Fig S1). Could the authors discuss this?

- I would like to see more discussion of the differences between the current control group results and the authors' previous paper on the same task

Minor comments

- p 10 "negative connectivity between the dmPFC and vmPFC areas" is different from "in ANX, a negative correlation between the dmPFC-OFC and dmPFC-vmPFC ($p < 0.001$ Bonferroni corrected) was found". Please rephrase.

- Often the term discrimination is used whereas there was no discrimination to perform here, it is not a noisy or ambiguous perceptual decision-making task, only learning in the two zones

- A 2x4x2 ANOVA is then analysed in a 2x4 ANOVA only, that reports a trend-level effect at $p = .068$ I think this should be removed and I am not sure these two ANOVAs are best practice, could the authors explain why they did not conduct post hoc tests on their main ANOVA instead? (i.e. stats in Fig S2)

- Results sometimes read as a list "and next we did that", whereas it would help the future readers to motivate each analysis. Similarly, could the authors explain the motivation for performing PPI in the first place?
- In Fig 4, some labels have changed as compared to the authors' previous study, shouldn't they be consistent? Panel A and B show the same circle instead of illustrating a difference between Learning and Danger. Please indicate on the figure the meaning of colors, arrows, and boxes.
- p21 healthy comparisons do you mean healthy controls?
- I recommend removing from the abstract and introduction the parts about creating novel treatments which is too far fetched
- For the analysis examining differences between flowers (conditioned) and objects (not conditioned): it seems to me that it would have been more appropriate not to collapse over safe and danger zones?
- The bar plots in Fig 2 and Fig 3 need stars

Reviewer #2 (Remarks to the Author):

In this manuscript, Suarez-Jimenez et al. examine threat-related behaviours and their associated fMRI BOLD responses in two groups of participants (pathological anxiety ANX, n=23 and controls HC, n=28). Participants learn to associate the locations of flowers in a circular arena with a potential danger to receive an electric shock (half the arena) versus safety (no risk of shock, other half of the arena). While stress levels and performance do not differ between the groups, several differences emerge in the BOLD responses between anxious individuals and controls in a priori regions of interest evident when participants approach and anticipate potential shocks.

This study has the potential to be very interesting. The authors use a rich and ecological task paradigm which addresses important questions regarding learning about and anticipating potential threats. In particular, the authors are asking whether these behaviours and associated BOLD responses differ in anxious cohorts which is an important step towards a better understanding of anxiety disorders. In its current form, however, it is difficult to judge the validity of the statistical tests because the results are very incoherent and not consistent throughout. A second major problem is that markers of stress show no significant difference between the groups, which goes against much of the literature on anxiety disorders and disconfirms one of the authors' key hypothesis. While this might be interesting, it could also simply mean that the task did not measure stress-related processes actually affected in anxiety disorders and it makes it difficult to interpret any differences in BOLD responses reported during performance of the task.

Major:

1. There is no evidence that anxiety changes measures of stress in this task. The measures of stress presented (skin conductance and shock expectancy ratings) do not show a significant difference between groups (Fig S1; e.g. the lack of an effect ($p=0.06$) for ANX versus $p<0.01$ for HC does not provide evidence for a difference between groups and this is the case for both approach and stationary periods). The authors do not report data related to the duration of approach in safe vs danger zones (or the length of the path taken, though these are likely correlated) which could be interesting but from the data shown, there is no evidence that threat learning, appraisal or anticipation differs between ANX and HC in this task. This is not explicitly mentioned in the main text until the discussion, despite it being one of the three key hypotheses tested in this manuscript, and there is no figure showing these behavioural results in the main manuscript. Unfortunately, this lack of a behavioural effect makes subsequent differences in BOLD difficult to interpret. Currently, the authors offer little to

reconcile the absence of behavioural/skin conductance responses with the presence of many BOLD differences. In fact, in the discussion the authors state "In patients with pathological anxiety, reintegrating the anterior hippocampus and vmPFC into the circuitry could rescue appropriate learning strategies to discriminate between safety and danger." This statement does not seem to match up with the results of the manuscript where anxious participants had no problem discriminating between safety and danger.

2. At the moment, the results are very difficult to follow. The statistical methods and precise contrasts used are often unclear and described inconsistently (e.g. between figures and main text and methods and supplement) and as a consequence the meaning and validity of the results are difficult to judge.

I will give one example, but this applies throughout. In line 153, it sounds like the contrast tested is across both zones and task blocks and thus an overall difference between ANX and HC in the approach period of the task. This is what the abstract suggests as well (global decreased activation during the task), but I don't think this is what was done. Instead, I think the authors performed a full interaction between blocks, groups and zones (as suggested by Table S1). The next sentence (line 157) talks about comparisons between late>early blocks and safe vs danger zones making it sound different from the abstract. Is this a qualification of the previous sentence in line 153 or a new result? It is really important to be absolutely clear in the text and figures which contrasts were performed. This is also important for understanding from which contrast peak coordinates were extracted. Currently, it is hard to know whether the bar-plots presented throughout all figures are mere illustrations of the main contrasts shown above, or whether statistical post-hoc tests were performed. It is therefore not possible to be sure about any potential circularity that might be present. The authors talk about post-hoc tests but don't present any statistics on these. Line 157 is one example of such a statement which is statistically very vague, "Compared to the HC group, the ANX group demonstrated a greater increase in activity in these areas from early to late blocks of the safe zone and a decrease in activity from early to late blocks (note typo) of the danger zone." It seems that the authors have performed multiple tests on multiple brain areas here, but no statistical result is reported. And the Figure legend related to this result, starts "For ANX, activation..." making it sound like this was a contrast estimated for ANX only (rather than what I think it was namely an interaction between ANX>HC and block and zone). Please spell out much more clearly throughout what hypotheses are being tested, which precise tests were performed, what statistical tests including post-hoc tests were performed, and make sure conclusions stated in the abstract, discussion etc match up with the main text.

3. A similar point holds for the PPI analyses, it is unclear what exact hypotheses were tested (if any) and which precise analyses were conducted. It seems that the authors use any main activation they find to subsequently run a PPI on. This not only results in many tests but the results are hard to follow, and no figures are shown for any of the PPI results. The authors state, for example "PPI examined brain connectivity of each significant cluster (i.e., seed ROI) from the approaching flowers period." This is very vague, there are three contrasts reported in the supplement for the approach period, so which one are the authors talking about, and which/how many regions were taken as seeds? For the PPI, the authors also don't fully justify the choice of their statistical threshold ($p < 0.001$ Bonferroni) which differs from the rest of the manuscript. They refer to one of their own previous papers for justification which is not sufficient. Finally, PPI results cannot be interpreted as different between groups if a direct comparison between groups is not performed (as I understand it, PPI results were done on HC and ANX separately) e.g. see discussion: "Unlike healthy comparisons, patients with pathological anxiety failed to exhibit task-related functional connectivity with higher-level cortical areas related to emotional regulation (vmPFC and OFC) throughout the task."

4. It is also unclear if all or just a chosen subset of all possible contrasts were looked at (given the available GLM regressors). For instance, there are several that I think it would be interesting to look at, but I am unsure if they have been conducted (e.g. comparing ANX and HC averaged across zones and block, or just contrasting danger and safe zones independent of block between the groups).

5. The figures seem to show a somewhat incomplete selection of areas out of all those that are mentioned as significant in the text and it is unclear why peak activations are not shown for e.g. vmPFC and PCC in Figure 2A and insula in Figure 2B. Were the patterns the same as in the other regions shown or was there a reason for omitting them?

6. It seems like a large number of participants are excluded due to poor task performance (14 out of 70) but one of the criteria was their performance recalling object's names in a task not of interest to this study. How many participants would be excluded if only the flower contingencies relevant to the task at hand were used as a criterion and would that change any of the conclusions?

7. The BOLD results between approach and stationary periods seem quite similar, at least judging from the text (this does not match the figures for the stationary period). Please report the correlations between all regressors in the BOLD GLM to show that they were independently estimable (e.g. as a supplementary figure). While this should be done either way, I do wonder if this was a copy and paste error? Lines 157ff and 207ff are identical even though they describe results from two different phases, and they don't match up with the results that are shown for the stationary period in Figure 3.

Minor:

1. Please specify precise p-values for all statistical tests.
2. Please tone down direct link to treatments or explain how the results would be directly linked to treatments (e.g. abstract, introduction)
3. Were participants forced to pick all flowers even if they were sure that they were in the danger zone. Are they slower approaching/engaging/picking in later blocks when they know a flower is in the danger zone?
4. Please specify what time exactly contributes towards the stationary period e.g. for skin conductance analyses. Since the duration is variable (2-8s), are you just using the common 2s, or the entire duration?
5. Please be more precise about recruitment – line 393ff: it sounds a bit like 70 people were recruited in total and by chance 30 were ANX and 40 HC?
6. Anatomical labels: what is circled as OFC is on the medial surface and would usually be referred to as subgenual cingulate cortex or BA25.
7. Line 490: were there three or four regressors for modelling the end of each trial; three makes sense but that makes the overall number of regressors 15 rather than the 16 stated
8. There are grammar mistakes and typos throughout (e.g. line 87 "but this not clear if this extends to context"; "MNI coordinated", contrast "dange > safe", "blacks" instead of blocks)
9. "Healthy comparisons" is an unusual term to refer to healthy participants, usually, "healthy controls" would be used instead

Reviewer #3 (Remarks to the Author):

In this manuscript, Suarez-Jimenez et al. extend on their 2018 PNAS paper in which they assessed the neural mechanisms of contextual threat conditioning, threat approach and threat expectation in a virtual reality environment with fMRI. Here, they use a very similar design to look at patients with anxiety disorders (GAD, social anxiety disorder) vs. healthy controls.

Their main finding is that behaviourally, patients are unimpaired in learning, but they do differ in a number of fMRI responses to the different task phases. This is broadly interpreted as reduced threat

learning, increased threat appraisal, and higher threat anticipation. Nicely or oddly, the fMRI findings almost exactly match the hypotheses, even though they are based on the behavioural hypothesis which is not confirmed. Nevertheless, with some caveats, I believe this is a well-designed and important study that can make a contribution to the literature. I believe that several of the current weaknesses could be addressed by the authors.

First, the authors may want to avoid the impression of another just-so story in cognitive psychiatry – which it is not – by being more rigorous with the description of the analysis pipeline and the derivation of their hypotheses. This is a well-designed experiment, based on a previous study in healthy volunteers, and with almost the exact same design and even analysis pipeline. Still, there are some discrepancies with their earlier work (e.g. 16 instead of 15 regressors in the first-level design). To avoid any (certainly incorrect) impression that analysis was flexibly adjusted so that results "make sense", I would suggest a more systematic discussion of where the analyses deviates from the previous study, and why. Furthermore, it would be good to give a more nuanced derivation of the hypotheses, in particular of the regions of interest on which the authors focus. With respect to the behavioural hypothesis, the authors write that "Research has shown that individuals with anxiety disorders display a higher defensive response to a CS+, compared to healthy individuals, and they often generalize the threat response to safe cues (CS-) ". This is backed up with selected and slightly old primary and review papers, but there is a large literature on this topic and an exhaustive meta-analysis by Duits et al. 2015 with more than 2000 participants shows a more nuanced picture. This renders the behavioural hypothesis, which is based on this idea, somewhat brittle. The behavioural hypothesis is taken to support the first fMRI hypothesis (compensatory learning strategies not involving the hippocampus), which hence also appears dubious. Regarding the other two hypotheses, I was unsure where they come from, and I just note that they look a lot like the results.

Second, the hypotheses are rather unspecific and mention at least 6 regions/contrasts where the authors expect to find an effect to support their general theory. This multiplicity requires a strong strategy to guarantee that multiple hypothesis testing does not lead to inflated false positive rates. While I gather from the results that this is quite likely not the case and the conclusions are probably valid, this should be spelt out in more detail. Will the authors conclude in favour of their theory only if all these 6 tests are positive, or also if one or two are positive?

More technically but related to this point, it is not always clear a priori when the authors will do whole-brain correction and when they will do small volume correction, and how they will correct for the number of small regions that being corrected for.

Another technical point is their negative behavioural finding. Could it be that they do not find an impairment in contextual conditioning because they use a relatively non-standard and probably not the most sensitive scoring strategy? Sure the field has moved beyond peak-scoring in general, and study-specific peak scoring windows in particular?

Finally, there is a bit of vague speculation and reverse-inferencing in the discussion. For example, the conclusion that " Our findings suggests that patients' threat-related attention may cause deficits in emotion regulation. The current results suggest that disconnected circuitry in brain areas essential for emotional regulation might lead to disrupted emotional output while exploring environments and learning to discriminate cues within these environments." appears rather unsupported. The paper does not define what "emotion regulation" is on a descriptive level, let alone come forward with a mechanistic model of it. How can we conclude deficits in this unspecified process from the observation of more or less oxygen in some brain regions?

Reviewer #1

Comment 1

The behavioral analyses were very succinct as compared to the richness to the experiment (only the ratings of shock expectancy are analysed).

- a. What about the participants' spatial trajectory in the environment? Could the authors analyse the mouse cursor trajectories for instance?
- b. Do participants spend more time in either zone? Do participants with anxiety spend more time in the safe zone as compared to the control group?
- c. Do participants with anxiety pick up fewer flowers overall? Or does the experiment stop until they have picked all flowers, and if so, did participants with anxiety take longer to complete the experiment?
- d. Could the authors find basic behavioral signatures of more avoidance in participants with anxiety, for instance exploring for longer before picking up any flower?

Response 1

We thank the reviewer for this important comment and excellent suggested ideas for enriching the analyses of the task. Accordingly, we have made the following changes to the original manuscript:

- 1) We have added a new analysis of State anxiety before and after the task to make the results richer
- 2) We have addressed the issues of spatial trajectory and time spent in zones (also discussed in detailed below) in the Discussion, adding them as possible limitations of performed analysis
- 3) We have expanded on the importance of the brain analyses and its replication of previous findings with the same task.

1) State anxiety to increase behavioral analysis richness:

To enrich our behavioral results, we added an analysis of the state-trait anxiety inventory (STAI; Spielberger, Gorsuch, Lushene, Vagg, & Jacobs, 1983), focusing on the state anxiety scale, which we collected before and after the task. The state anxiety scale has 20 self-reported items assessing anxiety experienced at the present moment. Therefore, we were able to explore changes in state anxiety from before to after completion of the VR task. We found that ANX participants had higher state anxiety scores compared with HC participants at post-task, while at pre-task, both groups showed similar state-anxiety levels.

We have added this to the Results section as follows (line 138): “*State-Trait Anxiety Inventory (STAI). Before and after the task participants filled out the state anxiety section of the STAI. A 2x2 ANOVA (group x time) analysis of the state anxiety (pre-task, post-task) revealed a significant group effect ($F(1, 22)=28.07, p=0.000$). No other significant main effects or interactions were observed ($F's < 2, p's > 0.05$). A t-test revealed that ANX had higher state anxiety than HC only at post-task time point (pre-task, $t(49)=2.56, p=0.116$; post-task, $t(49)=6.51, p=0.014$).*”

- Spielberger, C. D., Gorsuch, R. L., Lushene, R., Vagg, P. R., & Jacobs, G. A. (1983). Manual for the State-Trait Anxiety Inventory. Palo Alto, CA: Consulting Psychologists Press.

2) Spatial trajectory and time spent in the zones limitations:

While we did collect data on participants' navigation and orientation, several factors have precluded our ability to adequately use it.

A. Participants' characteristics: While some people learned very fast how to use the controllers and navigated the environment, some learned more slowly but still learned it eventually (at different time points during the task), and others were very slow throughout the experiment. Hence, these differences also affected both the trajectory and the time spent in zones. However, these differences in learning rates could have been the result of several factors, not related to group differences. These include:

- i. Participants' experience with video games: We suspect that people with more experience playing video games, particularly those requiring navigation of some sort, would learn faster how to use the controllers and navigate the environment of the game, while people with no experience would take longer to learn the navigation, if at all. Unfortunately, we did not collect information about gaming experience or type of gaming experience. Most information was anecdotal gained through informal conversations with participants after the task, when they would mention never playing video games or saying how it reminded them of another game they had played.
- ii. Age: We noticed that in general, older people were more likely to be slower than younger people, although this was not always the case. Related to our previous point, older people are also less likely to have video game experience than younger people.
- iii. Gender: Females, again not all, tended to be slower than males. However, again, this could also be linked to gaming experience, as females are less likely to play video games than males (though this is changing fast in younger generations and with the accessibility of video games on cellphones).

B. Task characteristics: While flowers were counterbalanced between zones (20 in each zone), their location was randomized. Flowers could appear close to each other either at the boundary between zones or within the same zone. Or, they could appear far from each other, at different side of the environment, again, between different zones or within the same zone. Flowers were allowed to appear in the same zone, though we capped this at a maximum of 3 flowers in the same zone to increase movement between zones. Overall, these factors could affect time spent within or between zone, making it hard to determine whether participants spend more or less time in a zone due to conscious thought or due to the randomness of the experiment. These factors also affect spatial trajectory. Relatedly, sometimes flowers, particularly if they were very far away, might have been difficult for participants to see initially, increasing the time of orientation. And while we told participants at the beginning of the experiment "If you do not see the flower, try walking across the environment as it might be easier to find" participants used different strategies to find the flowers and not always used the walk across strategy to find them. Importantly, as the task was designed to measure spatial learning and discrimination of zones, not avoidance, approach, or conditioning, we did not account for the differences in spatial trajectory (navigation) when designing the experiment. So, while everyone learnt the contingencies differently we were more interested in the brain areas engaged during cognitive mapping of the environment to discriminate between zones. This is why we measure brain activity during approaching the flowers and after picking them (during restriction of movement). Participants indeed had to collect all the flowers to complete the study, everyone had to collect the 40 flowers in the danger zone and 40 flowers in the safe zone. And participants were forced to go towards the flower, there was not a selection of flowers for them to pick from. There was always one flower present in the environment, which they had to pick to make another one appears. Therefore, some people might be slow to pick the flower (as to

avoid potential shock) but other might want to get it over with and therefore rush towards them. But we did not ask about this strategy or thought process. We have clarified this in the results section as follows (line 118): *“For each trial, participants freely explored the environment and were instructed to pick up flowers that appeared one at a time in random locations across the environment (approach period). When a participant picked a flower, their position was held stationary for a variable duration (2-8 seconds; stationary period), during which the participant rated the expectancy of receiving a shock (rating of 0-9). After the stationary period a new flower would appear for the participant to find. There was only one flower in the environment at a time.”*

In sum, while we do agree with the reviewers’ ideas, due to the above-stated reasons, a clear and coherent conclusion based on this data might be difficult and possibly even misleading. Nonetheless, if the reviewer still wants us to run and add these analyses to the manuscript, we would be happy to oblige. However, for our future studies we are planning to look more closely at the approach/avoidance question through additional VR tasks we are developing, including eye-tracking to clearly assess gaze-navigation patterns. We will also collect more information on video game experience, including VR specifically, to correct these limitations. Yet, as we do concur with the reviewer, we now added some of the above-detailed information alongside an interpretation for the lack of behavioral findings to the discussion as follows (line 566): *“We could not add the spatial trajectories or time spent in each zone to the behavioral analysis due to several factors of the task, 1) participants were forced to collect one flower at a time; 2) the flowers appeared randomly throughout the environment therefore they could be close or far from each other; 3) age, gender, and video game experience might have affected the task proficiency in general. Future studies should consider using other sorts of behavioral measures (e.g. eye-tracking), increasing the stakes of the shock contingency (e.g. correct expectancy ratings diminish the risk of shock), or increasing the sample size to increase power and therefore flush out the exploratory findings.”*

3) **Brain analysis importance:**

Most importantly, while we don’t show a strong behavioral difference between groups, this study shows differences in brain activity which are not confounded by behavioral differences. And, in this time of replicability issues, we note that we were able to replicate our initial HC findings. Furthermore, the differences between groups are in the brain areas we identified in our initial study for learning to discriminate between zones and for tracking threat. This suggest that if there are indeed no behavioral differences between groups, the ANX group is engaging the same brain areas (albeit differently) as compensatory mechanisms to rescue the appropriate behavior. We have added more about the current HC results and the previous paper on the same task in the discussion as follows (line 444): *“Several points are noteworthy when considering the current findings in light of the previous study in healthy controls using the same task. In both task we have identified the same network of activation in HC’s when approaching and interacting with flowers. We replicated the two neural networks to support the formation of cognitive maps and its relevant behavior: 1) The discrimination learning network (DLN; vmPFC, PCC, anterior hippocampus) activation increased when HC’s learn to discriminate threatening and safe zones within the environment; 2) the salience network (SN; dmPFC/dACC, insula, PAG) activation increased when participants approached and interacted with flowers in the danger zone. There is one main difference between this and the past study in the SN, in the current study the peak activation was*

more in the dmPFC while in the previous study the peak was more in the dACC. However, the brain activation cluster of both studies often overlap between the two areas (dmPFC/dACC). Furthermore, while some brain activation differences found in the current study are using a priori hypothesis SVC based on the past study, all the group brain activation differences (HC, ANX) of the current study are in the areas identified in the previous study (DLN, SN).”

Comment 2

I could not understand if analyses were ROI-based or whole-brain. We are told that all statistical values reported throughout the paper are FWE whole-brain corrected, but also that a ROI-based analysis was employed based on ROIs from the authors' previous study. I note that the z-score values reported in supplementary table makes it more likely that the latter is employed. It is important that the authors clarify what has been done.

Response 2

We apologize for the lack of clarity when describing the analysis. All initial analyses were done using FWE whole-brain correction. But, given our a priori hypothesis and findings from the previous study (Suarez-Jimenez et al. 2018), we used SVC in areas of the medial prefrontal cortex (mPFC) and hippocampus when they did not survive this stringent FWE whole-brain correction. We opted for using the more lenient FWE ROI-based corrections using a single mask. We have revised throughout the manuscript to more accurately describe the brain analysis done.

We have added (line 228): *“All statistical values reported are FWE whole-brain corrected ($p < 0.05$). However, given our a priori hypothesis, additional FWE small volume correction (SVC) was performed when areas of interest (hippocampus, amygdala, and mPFC) did not survive FWE whole-brain correction. One bilateral mask, which included the hippocampus, amygdala, and mPFC, was used for the SVC analysis.”* to make it clearer before describing the results.

To clarify that we only used one mask for the SVC analysis we also edited the sentence in the Methods describing these analyses to (line 727): *“One bilateral mask was defined which included the hippocampus, amygdala, and mPFC (orbitofrontal gyrus, ventromedial prefrontal cortex, dorsomedial prefrontal cortex, and anterior cingulate and medial cingulate cortex). Statistical threshold in this single mask was defined by small volume correction (SVC; $p < 0.05$ FWE)”*. Therefore, no multiple comparison correction was used in the SVC analysis.

Comment 3

Why for Fig 2B dmPFC the split first half/second half was not used?

Response 3

All of our analysis used a group (HC, ANX) by zone (safe, danger) by block (early, late) interaction. For dmPFC during approach (as well as for the amygdala, OFC, and hippocampus during stationary periods) there was a significant group difference at the zone level. Taking for example the dmPFC, while there were no differences over time, overall ANX had a higher activity in this dmPFC region than HC. We sought to highlight this difference in the figure. Adding the time split, or even the safe zone bars, would make the difference between groups (of the danger zone) less clear, as the scale and information of the other activity (early, late, and safe zone) would have occluded it. For example, the bar graphs would have look as the figure below. We thought it would make a clearer and highlighted point to keep the figure simply to the zone difference between groups when appropriate to highlight the finding.

Figure legend: safe blocks (S) are surrounded by blue line; danger blocks (D) are surrounded by red line; dark grey bars represent early block (ER); light grey bars represent late block (LT); safe early block (S ER); safe late block (S LT); danger early block (D ER); danger late block (D LT); health control (HC) are represented in the left four bars; patients with anxiety (ANX) are represented in the right four bars.

Comment 4

I appreciate that the authors sought to decompose learning periods, but it would be great to better justify the time split, because the results section jumps back and forth from analyzing all trials to analyzing first half/second half to analyzing block 1 to block 4.

Response 4

We apologize for this confusion. To clarify, it is important to note that we only split into four blocks during the behavioral analysis because power allowed it to be decomposed in that manner. In our initial study in HC (Suarez-Jimenez B, et al. 2018) we were able to describe a clearer behavioral pattern by separating the behavioral results into four blocks. However, due to power in the brain data we found that dividing the blocks in half, provided a higher signal-to-noise ratio. We sought to keep in this study the same analysis and approach to the results as in our original study. Therefore, we use four blocks in the behavioral results (hoping to provide a more detailed window into behavior) and we divided the brain data in half to increase signal to noise ratio. We

have elaborated on this as follows (line 219): *“To assess discrimination learning, trials were divided into two blocks comprising the first (early) and second half (late) of the experiment. As opposed to the behavioral results which used four blocks to assess learning, here we divided the data into two blocks, where the data was divided in half. In other words, the fMRI data comprised of block 1 and 2 (early block) and block 3 and 4 (late block). This was done to increase the signal-to-noise ratio of the fMRI data when looking at learning over time and for consistency previous the previous study.”*

Comment 5

Similarly, the focus on first half vs. second half of the experiment seems post hoc? (as indicated by the authors, to “increase SNR”). Does the finding that “the ANX group demonstrated a greater increase in activity in these areas from early to late blocks (late > early) of the safe zone, and a decrease in activity from early to late blocks of the danger zone” (p. 8) average out when looking at the whole experiment?

Response 5

We are grateful to the reviewer for this comment that assisted us in clarifying this important issue. The early vs. late block is an important contrast in the study and not a post-hoc analysis. These analyses allow us to look at learning differences over time. In other words, how brain activity and behavior change over time, which was a key aspect of our previous findings and study (Suarez-Jimenez B, et al. 2018). We have rewritten the explanation to increase clarity in this section. As mentioned above it now reads as follows (line 219): *“To assess discrimination learning, trials were divided into two blocks comprising the first (early) and second half (late) of the experiment. As opposed to the behavioral results which used four blocks to assess learning, here we divided the data into two blocks, where the data was divided in half. In other words, the fMRI data comprised of block 1 and 2 (early block) and block 3 and 4 (late block). This was done to increase the signal-to-noise ratio of the fMRI data when looking at learning over time and for consistency previous the previous study.”*

Comment 6

The authors dissociate stimuli (same in all zones) from context (different). But all these are visually displayed on the screen and no hierarchy exists between them, so could we reframe this as stimulus-response mappings learnt, not contextual information? Sure the flowers are the same, but the flowers embedded in context A or B could be re-labelled as a unique stimulus A or B.

Response 6

While we agree that all stimuli and context is visually displayed on a screen, participants are freely moving (they have full control of the game avatar) within a 3D virtual environment. Therefore, participants must indeed learn about the contextual information of their surroundings to orient themselves, learn the flower contingencies, and the location of the objects they pick up and replace. Indeed, they are building a cognitive map of their environment, but this is different that stimulus-response mapping seen in traditional context conditioning studies where they see different pictures, shapes, colors, or where their movement are guided through a prerecorded path. Overall, there is no explicit context A or B, but rather a large circular environment the participant is exploring and learning about. At the end, the participant is the one defining what is “context A” and “context B” which would translate into the safe and dangerous zones. To better clarify this important point, we now further explain this in the Results section as follows (line 115): *“Participants had full control of the virtual character (first person perspective) and explored a virtual circular environment (Fig.*

1A and B; see Methods for details). The environment consisted of a mountain landscape and defined two half-zones recognizable by the unique shape of the mountains in the horizon.” and “Since all flowers were identical, predictive value (danger or safety) could not be attributed to their physical characteristics. Participants had to build their own mental representation of the circular environment, which had no visible or distinctive boundaries between the zones, to define what they consider a safe and dangerous zone.”

Comment 7

Relatedly, the authors say that people with anxiety generalize defensive response to CS-safe cues, which is different from a context. Do the authors base their claim that context conditioning requires the use of spatial processing strategies? (Not all contextual cues are spatial).

Response 7

We are interested in the process of discrimination learning, particularly within the environment – how do participants build cognitive maps that discriminate between danger and safe zones? We based our hypothesis on discrimination learning studies, which have mostly been conducted in stimulus-response studies. This is different from classical conditioning studies that aim to understand the difference between distinct stimuli or contexts. Here we aim to understand the process of discrimination when the environment and cues are similar (or identical in the case of the flower task). So, while we have based our hypothesis on discrimination learning studies, we are interested in exploring if this is also applicable to context conditioning and discrimination, which to our knowledge has not been previously done. We have added this information and references (see below) to the introduction as follows (line 90): *“A discrimination of rings task has been used to systematically elucidate neural signatures of generalization/discrimination in patients with panic disorder, generalized anxiety disorder, and PTSD. These studies show that patients, compared to healthy controls, exhibit an overgeneralization, or lack of discrimination, towards cues similar to the CS+. But it is not clear if this overgeneralization extends to context, particularly when there is no clear-cut boundary between safe and danger zones within the environment.”*

- Lissek S, Rabin S, Heller RE, et al. Overgeneralization of conditioned fear as a pathogenic marker of panic disorder. *Am J Psychiatry*. 2010;167(1):47-55.
- Lissek S, Kaczkurkin AN, Rabin S, Geraci M, Pine DS, Grillon C. Generalized anxiety disorder is associated with overgeneralization of classically conditioned fear. *Biol Psychiatry*. 2014;75(11):909-915.
- Kaczkurkin AN, Burton PC, Chazin SM, et al. Neural Substrates of Overgeneralized Conditioned Fear in PTSD. *Am J Psychiatry*. 2017;174(2):125-134.

Comment 8

Participants were a blend of Generalized Anxiety Disorder and Social Anxiety Disorder that are fairly different. Does the nature of the disorder influence the findings? Was there a particular motivation for recruiting social anxiety patients in a paradigm involving no social component? Could the authors provide a breakdown of how many with each disorder were recruited?

Response 8

We were interested in recruiting patient with anxiety disorders in general, however, we were only able to recruit patients with GAD (8), SAD (7), and GAD+SAD (8). Our aim was to assess if anxiety disorders in general affected the way people build cognitive maps of their environment. We have added this to the limitations of the study as follows (line 574): *“Another component that*

might have affected the behavioral results is the inclusion of patients with social anxiety disorder (SAD), as the task did not include any social component. The study aimed to capture differences between HC and patients with any anxiety disorder. And while the study only recruited a minimal number of patients with SAD (7), their inclusion might have skewed the results more favorably to appropriate learning behavior. Future studies should focus on patients with generalized anxiety disorder to test this.”

Comment 9

The spatial memory task null findings are reported in the Supplement would deserve to appear more prominently: it is important to know that all participants were able to learn the placement of objects, and that they similarly learnt in the safe and danger zones, and that control and anxiety participants also similarly learnt

Response 9

We agree with the reviewer that this results are important, therefore, as suggested, we have moved these results from the supplementary material to the main manuscript.

Comment 10

Could the findings be simply explained in terms of the safe zone being more valuable than the danger zone (that by design has more punishment), the classic valuation network being observed in this contrast (vmPFC/PCC)?

Response 10

This is a great point and indeed something to be considered, and we thank the reviewer for raising it. However, based on our original study in healthy adults (Suarez-Jimenez B, et al. 2018) and this replication in healthy controls we see that activity in the vmPFC/PCC along with the anterior hippocampus increase over time, as participants learn to discriminate between zones. That is, the highest activity in these areas is when they have learned the contingencies and rules of the environment, regardless of zone, as there is no zone difference in these areas. The zone difference surfaces when we look at patients with anxiety disorder, that show an increase in activity over time in the vmPFC/PCC/aHippocampus for flowers in the safe zone, while a decrease over time for flowers in the danger zone. This interaction could point to the possibility that patients with anxiety disorders have a higher valuation for the safe zone compared to the danger zone. However, we see that increases in salient network areas (insula/dmPFC/PAG) increase over time, particularly for the danger zone. Therefore, we believe that this system takes over in the danger zone, overriding or dampening the activity of the vmPFC/PCC. Additionally, if this vmPFC/PCC network was indeed active during the task for valuation we would expect that healthy participants would display this pattern of activity as well. One could reason that maybe this difference is due to patients with anxiety disorders being more “fearful” of the danger zone, hence giving a higher value to the safe zone. And compared to healthy participants who might not “fear” the danger zone as much, then it might make sense that they indeed don’t use this network for valuation or that they use it at a lesser extent. But still, we would expect a similar pattern even if at a lesser degree. And indeed, it is not the case as we see that healthy controls (in this study and the previous one) show an increase activity of the vmPFC/PCC/aHippocampus in both zones over time. As this paper is focusing in context discrimination and not in the cues themselves, we did not add this discussion on classic valuation of cue to the paper. However, we would be happy to add this point to the discussion if insisted upon by the reviewer.

Comment 11

Could the authors provide references supporting the first hypothesis about poor learning and discrimination in anxiety? Especially as the authors happen to find no evidence for this hypothesis in the end.

Response 11

This hypothesis is based on Lissek's studies of discrimination. Lissek et al. used a ring discrimination task to systematically elucidate neural signatures of generalization/discrimination. This task was tested in patients with panic disorder (Lissek et al. 2010), generalized anxiety disorder (Lissek et al. 2014), and PTSD (Lissek et al. 2017). These studies showed that psychiatric patients compared to normal controls exhibit stronger generalization, as a putative marker of disrupted threat discrimination. We have now added this information and references to the introduction as follows (line 90): "*A discrimination of rings task has been used to systematically elucidate neural signatures of generalization/discrimination in in patients with panic disorder, generalized anxiety disorder, and PTSD. These studies show that patients, compared to healthy controls, exhibit an overgeneralization, or lack of discrimination, towards cues similar to the CS+. But it is not clear if this overgeneralization extends to context, particularly when there is no clear-cut boundary between safe and danger zones within the environment.*"

- Lissek S, Rabin S, Heller RE, et al. Overgeneralization of conditioned fear as a pathogenic marker of panic disorder. *Am J Psychiatry*. 2010;167(1):47-55.
- Lissek S, Kaczkurkin AN, Rabin S, Geraci M, Pine DS, Grillon C. Generalized anxiety disorder is associated with overgeneralization of classically conditioned fear. *Biol Psychiatry*. 2014;75(11):909-915.
- Kaczkurkin AN, Burton PC, Chazin SM, et al. Neural Substrates of Overgeneralized Conditioned Fear in PTSD. *Am J Psychiatry*. 2017;174(2):125-134.

Comment 12

No occipital activations are reported in any of the tables though some are visible e.g. Fig 2; could the authors clarify?

Response 12

For display purposes all brain activity images were presented at $p < 0.001$ uncorrected to provide a more simplified image rather than multiple images across different brain regions. We have added this clarification in the figure caption: "*All images are presented at $p < 0.001$ uncorrected for display purposes, not all clusters observed represent actual significant clusters.*"

Comment 13

A number of details were missing as compared to what is standard in the field. Could the authors report (i) how many participants were excluded for excessive motion; (ii) The authors excluded participants who in a debriefing were not able to report the task contingencies. How many? (iii) We are told that participant groups did not differ in terms of e.g. demographic variables, but statistical values should be reported to support these statements.

Response 13

We apologize for this oversight. We have added this information to the methods section as follows:

- i. "*Three ANX and two HC were omitted due to excessive head motion (>20%; see below).*" (line 610)
- ii. "*Four ANX were excluded from analyses because they were unable to explain the shock contingencies between the locations at the end of the task*" (line 608)

- iii. *“Participants were free of medications, and no significant differences (p 's>0.05) in any demographic information between groups were found. See supplementary material for a demographic breakdown (Table S4).”* (line 612)

Comment 14

We are told that the question is to “elucidate neural mechanisms of anxiety” which says very little about why the experimental design was set up the way it is. Could the authors clarify the specific question and specific motivation for their study?

Response 14

We are grateful to the reviewer for this comment that helped us increase the clarity of the paper, and have taken several steps to expand on why was the experimental design was set up the way it is. Overall, we were interested in testing whether patients with anxiety disorders could learn to discriminate between zones within a single environment, and to explore which were the brain areas engaged during this process. Particularly, we wanted to test whether the brain of anxious participants behave similarly to that of healthy controls (as per our original PNAS study and the replication here) and if they engage compensatory mechanisms for learning (if indeed they were able to learn to discriminate, which now we know they can).

In the introduction we have revised the ending of the first paragraph as follows (line 43): *“Research shows that patients with chronic anxiety lack the ability to integrate contextual cues to guide learning of threat and safety. A previous investigation delineated the neural mechanisms underlying learning and discriminating threats within specific spatial locations in healthy adults. However, very little is known about how patients with pathological anxiety learn about threat within an environment. Understanding how patients with pathological anxiety learn about threats within specific spatial locations in complex environments is essential to better understand the development and maintenance of the disorder.”*

In line 88, we have rewritten a paragraph to further expand and clarify these points: *“Research has shown that individuals with anxiety disorders display a higher defensive response to a safe cue (CS-), compared to healthy individuals, suggesting impaired ability to regulate their emotions or a generalization of the threat response to safe cues. A discrimination of rings task has been used to systematically elucidate neural signatures of generalization/discrimination in in patients with panic disorder, generalized anxiety disorder, and PTSD. These studies show that patients, compared to healthy controls, exhibit an overgeneralization, or lack of discrimination, towards cues similar to the CS+. But it is not clear if this overgeneralization extends to context, particularly when there is no clear-cut boundary between safe and danger zones within the environment. Nevertheless, hippocampal dysfunction and decreased hippocampal volume have been associated with anxiety disorders. For example, studies in both humans and rats suggest that impairment in hippocampal function leads to compensatory learning strategies that do not involve the hippocampus. These abnormal modulations of attention, linked to attention shifts to the cue and not the context, use compensatory neural mechanisms that lead to generalization of threat. This study aims to test if patients with anxiety disorder can discriminate between environmental zones (safe; danger) and what are the neural mechanisms engaged during the learning process.”*

Comment 15

The conclusion indicates that participants with anxiety exhibit a heightened reactivity to threat; isn't it the very definition of anxiety? Could the authors make the specific contributions of their study more explicit?

Response 15

We have revised this statement in the abstract to read “*Our findings suggest that ANX engage brain areas differently to modulate context-appropriate emotional responses when learning to discriminate cues within an environment.*”

Comment 16

The neural differences observed did not seem to translate into behavioral differences. If anything, the differences between danger and safe zones are overall smaller in anxiety patients than controls (Fig S1). Could the authors discuss this?

Response 16

We thank the reviewer for this comment.

To better address this, we tried to flush out the behavioral findings a bit more by adding the STAI pre-post differences analysis where we found that state anxiety between groups differed only in the post-task assessment, which was driven by higher state anxiety in the ANX group. We have added the STAI pre-post difference to the results sections as follows (line 138): “*State-Trait Anxiety Inventory (STAI). Before and after the task participants filled out the state anxiety section of the STAI. A 2x2 ANOVA (group x time) analysis of the state anxiety (pre-task, post-task) revealed a significant group effect ($F(1, 22)=28.07, p=0.000$). No other significant main effects or interactions were observed (F 's < 2, p 's > 0.05). A t -test revealed that ANX had higher state anxiety than HC only at post-task time point (pre-task, $t(49)=2.56, p=0.116$; post-task, $t(49)=6.51, p=0.014$).”*

Additionally, we added the exploratory analysis of SCL, SCR, and expectancy ratings within each group. We now show that while both groups show similar expectancy ratings, only HC had significant differences between zones in the SCL and SCR. We have added exploratory analysis of the SCL and SCR of each group as follows (line 148): “*Based on the first hypothesis, that ANX would evidence overgeneralization, within-group main effects of zone were explored and indicated that only HCs had significantly higher SCL towards dangerous flowers as compared to safe flowers (HC, $t(27)=3.94, p<0.01$; ANX, $t(22)=1.99, p=0.06$; Figure 1 C and F).*” and “*Within-group main effects of zone were explored, showing that only HC's had significantly higher SCR towards dangerous flowers as compared to safe flowers (HC, $t(27)=2.42, p<0.05$; ANX, $t(22)=0.98, p>0.05$; Figure 1 D and G).*”

We believe that lack of behavioral differences could be explained in two ways: 1) due to compensatory mechanisms; or 2) maybe these are not the best behavioral measures for this task. Maybe the behavioral measures we selected doesn't give the breath of scope to measure the full behavior difference in anxiety disorders, maybe other measures such as eye-tracking would be better suited for this purpose. Increasing the task-related threat risk could also help flush out these differences in behavior, such as raising the stakes by adding a component where correct assessment of flowers reduces the chance of shock and vice-versa. Another possibility is that we don't have the statistical power to flush out the true behavioral differences using these measures. We now

further discuss the lack of clear behavioral difference findings in the discussion as follows (line 558): *“It is important to discuss that while ANX displayed a higher state anxiety at the end of the task both patients and HC learned the contingencies of the task (expectancy ratings). This finding was supported by the skin conductance results during the task (SCL; SCR), which were only significant at when we explored the groups separately. Exploratory analysis did suggest that only HC were showing skin conductance (SCL, SCR) differences between the zones while patients did not. However, as exploratory analysis this should be taken with caution. We can speculate that there are indeed skin conductance difference reflecting that patients learnt the contingencies appropriately through brain compensatory mechanisms. But, maybe these behavioral measures are not sufficient to capture the full spectrum of behavior between groups. We could not add the spatial trajectories or time spent in each zone to the behavioral analysis due to several factors of the task, 1) participants were forced to collect one flower at a time; 2) the flowers appeared randomly throughout the environment therefore they could be close or far from each other; 3) age, gender, and video game experience might have affected the task proficiency in general. Future studies should consider using other sorts of behavioral measures (e.g. eye-tracking), increasing the stakes of the shock contingency (e.g. correct expectancy ratings diminish the risk of shock), or increasing the sample size to increase power and therefore flush out the exploratory findings. Another component that might have affected the behavioral results is the inclusion of patients with social anxiety disorder (SAD), as the task did not include any social component. The study aimed to capture differences between HC and patients with any anxiety disorder. And while the study only recruited a minimal number of patients with SAD (7), their inclusion might have skewed the results more favorably to appropriate learning behavior. Future studies should focus on patients with generalized anxiety disorder to test this.”*

Comment 17

I would like to see more discussion of the differences between the current control group results and the authors' previous paper on the same task.

Response 17

We appreciate the reviewer's important point, particularly in this time where task-based fMRI studies are being questioned due to replicability. We have expanded our discussion on the similarities and differences between the two studies and added more on the importance of the current paper as follows (line 444): *“Several points are noteworthy when considering the current findings in light of the previous study in healthy controls using the same task. In both task we have identified the same network of activation in HC's when approaching and interacting with flowers. We replicated the two neural networks to support the formation of cognitive maps and its relevant behavior: 1) The discrimination learning network (DLN; vmPFC, PCC, anterior hippocampus) activation increased when HC's learn to discriminate threatening and safe zones within the environment; 2) the salience network (SN; dmPFC/dACC, insula, PAG) activation increased when participants approached and interacted with flowers in the danger zone. There is one main difference between this and the past study in the SN, in the current study the peak activation was more in the dmPFC while in the previous study the peak was more in the dACC. However, the brain activation cluster of both studies often overlap between the two areas (dmPFC/dACC). Furthermore, while some brain activation differences found in the current study are using a priori hypothesis SVC based on the past study, all the group brain activation differences (HC, ANX) of the current study are in the areas identified in the previous study (DLN, SN).”*

Comment 18

p 10 “negative connectivity between the dmPFC and vmPFC areas” is different from “in ANX, a negative correlation between the dmPFC-OFC and dmPFC-vmPFC ($p < 0.001$ Bonferroni corrected) was found”. Please rephrase.

Response 18

We have rephrased this to “*negative correlation in activity between the dmPFC and vmPFC areas*” to reflect the results findings.

Comment 19

Often the term discrimination is used whereas there was no discrimination to perform here, it is not a noisy or ambiguous perceptual decision-making task, only learning in the two zones

Response 19

We thank the reviewer for this important comment that helped us increase the clarity on the use of the term discrimination, which is essential to this paper. The VR task aims to have participants build a cognitive map of their surroundings using the two (identical) landmarks (beehives) and the distal environmental cues. Participants must use these cues to orient themselves in the environment and to learn to navigate it. The environment itself has the shape of a circle with no differences within it, all corners and grass look the same. Additionally, all the flowers look the same. Therefore, there are no clear boundaries to dissociate the safe from the danger zones. This is a threshold that participants must recognize and build by themselves. Participants must first learn that not all flowers give shock, and discriminate which ones do. Participants are hinted at the beginning to check if there is any association between the flowers, beehives, and beestings (shocks). But they learn by themselves that the location of the flower predicts the shock and they must discriminate which parts of the environment hold this shock component. They must set the gradient (or invisible line) that divides the zones to discriminate between the flowers. We now better clarify this in the introduction by editing the paragraph as follows: “*The virtual environment depicts a circular grass field surrounded by mountains, divided equally into two zones, a safe and a danger zone. In both zones, flowers appear and need to be “picked” up. Picking flowers in the danger zone potentially causes an electric shock to the wrist (or “bee sting”), while flowers in the safe zone are never associated with shock. To learn threat contingencies, participants must rely on distal environmental cues (e.g., shape of the mountains and clouds, which differ in both zones, and beehives) to locate themselves and learn ‘where’ they are in the environment and not on the physical properties of the stimuli (i.e., the flowers), which are all identical in both zones. In other words, participants must learn to discriminate where in the environment is the threshold that divides the safe and danger zones (using the distal environmental cues) as there are no clear division in the circular grassy field, which is also identical throughout the circumference of the circle.*”

Comment 20

A 2x4x2 ANOVA is then analysed in a 2x4 ANOVA only, that reports a trend-level effect at $p = .068$ I think this should be removed and I am not sure these two ANOVAs are best practice, could the authors explain why they did not conduct post hoc tests on their main ANOVA instead? (i.e. stats in Fig S2)

Response 20

We apologize for the confusion and the redundancy of the ANOVAS. Indeed, the post-hoc analysis were done on our main 2x4x2 ANOVA. Originally, we thought that by adding the 2x4 ANOVA

we could show again the effects were not due to group differences. We realize now this was redundant and we have removed it from the document and edited the paragraph for clarity as follows: “A 2x2x4 ANOVA (zone x group x block) on mean distance error for the object placement showed no zone x group x block, zone x group, zone x block, or group x block interaction (all F 's < 2, p 's > 0.05). However, there was a significant effect of zone ($F(1,49)=6.47$, $p<0.05$) and block ($F(3,147)=18.48$, $p<0.001$). A post-hoc comparison of performance, of both groups, across test blocks showed that distance error decreased from block 1 to block 4 (danger, $t(50)=4.67$; safe, $t(50)=5.10$, p 's < 0.01) reflecting improved spatial memory performance irrespective of whether objects had been located in the danger or safe zones of the flower task. Main effects of zone analysis revealed no significant difference in the HC ($F(1,27)=2.84$, $p=0.103$) between error of objects placed in the danger and safe zone. A trend-level greater distance error in the safe zone was found for the ANX ($F(1,22)=3.69$, $p=0.068$). This trend level effect in learning between zones for ANX was due a significantly greater distance error in objects found in the safe compared to the danger zone in block 1 ($t(22)=-2.61$, $p<0.05$) but not in block 4 ($t(22)=-0.74$, $p>0.05$).”

Comment 21

Results sometimes read as a list “and next we did that”, whereas it would help the future readers to motivate each analysis.

Response 21

We agree with the reviewer. To increase the clarity of the motivation of the conducted analyses we have added further explanation before each. For example, in the approach period analysis we have added (line 242): “To understand how brain activation differed when approaching flowers in each of the zones we examined the zones contrasts (danger, safe) between groups” At the beginning of the paragraph to illustrate that we were delving into the zone contrast. We took a similar approach for the block contrast as follows (line 251): “To further understand brain activation differences as participants learnt the contingencies of the task we examined the block contrasts (early, late) between groups.” We have applied the same approach to the stationary period and to the object placing period analyses.

Comment 22

Similarly, could the authors explain the motivation for performing PPI in the first place?

Response 22

We apologize for the lack of clarity of performing these analyses. Overall, we were interested in seeing if the task and the physiology changes the task created (particularly in anxiety patients) correlated with patterns of brain activity of the dmPFC in the approach and stationary periods. That is, we were interested in exploring if the dmPFC brain activation correlated with other brain areas to predict learning and discrimination.

During the approach period this was performed to understand threat-appraisal as participants approach the flowers (anxiety states). We have further explained this in the results section as follows (line 275): “Given that dmPFC activation was higher in the ANX compared to the HC across the approaching period, particularly in the danger zone, we were interested to see how the task and the physiological state it caused in the participants interacted with the brain activity to further understand brain-behavior associations of anxiety-states (threat appraisal). For this purpose, we used a psychophysiological interaction (PPI) analyses for each participant group separately to identify dmPFC patterns which connectivity changed during the danger vs. safe

contrast. PPI examined the brain connectivity of the significant dmPFC cluster (i.e., seed ROI) from the approaching flower period.”

During the stationary period this was performed to understand threat-anticipation as participants interacted with flowers (fear states). We have further explained this in the results section as follows (line 346): *“Given that dmPFC and amygdala activation was consistently higher in the ANX compared to the HC across the stationary period, particularly in the danger zone, we were interested to see how the task and the physiological state it caused in the participants interacted with the brain activity to further understand brain-behavior associations of fear-states (threat anticipation). For this purpose, we used a psychophysiological interaction (PPI) analyses for each participant group separately to identify dmPFC and amygdala patterns which connectivity changed during the danger vs. safe contrast. PPI examined the brain connectivity of the significant dmPFC and amygdala cluster (i.e., seed ROI) from the stationary period.”*

Comment 23

In Fig 4, some labels have changed as compared to the authors’ previous study, shouldn’t they be consistent? Panel A and B show the same circle instead of illustrating a difference between Learning and Danger. Please indicate on the figure the meaning of colors, arrows, and boxes.

Response 23

We agree with the reviewer and we aimed to keep Fig 4 consistent across both papers. The original reason for this figure (Fig 5 in the past paper) was to provide a visual recap of the findings. In the current paper we added Fig 4, with the same aim, to recap the findings, so we kept the colors and figure closely identical to the other paper. Several differences emerged when creating this figure. The primary difference is that our current finding peak activity was in the dmPFC (as opposed to the dACC, although the clusters overlap). The second main difference is that we found no posterior hippocampus difference between blocks (between groups) when interacting with the flowers. Third, we found greater amygdala activity in ANX than HC when interacting with flowers in the danger zone. Finally, we added a dark red color to reflect areas that activity was higher in ANX and blue to reflect areas that activity decreased over time. The meaning of the colors, arrows, and boxes, is captured in the figure legend in more detail as to not overcrowd the figure. Nonetheless, we aimed to also capture the meaning of the colors (which represents the boxes and arrows) in the middle graph panels. We discussed the peak activity difference between papers in the discussion as follows (line 451): *“There is one main difference between this and the past study in the SN, in the current study the peak activation was more in the dmPFC while in the previous study the peak was more in the dACC. However, the brain activation cluster of both studies often overlap between the two areas (dmPFC/dACC).”*

Comment 24

p21 healthy comparisons do you mean healthy controls?

Response 24

We have replaced healthy comparisons for healthy controls.

Comment 25

I recommend removing from the abstract and introduction the parts about creating novel treatments which is too far fetched

Response 25

We agree with the reviewer and have removed this from both introduction and abstract. We have changed the sentences to the following, in the abstract: *“Elucidating neural mechanisms of anxiety is essential to understand the development and maintenance of anxiety disorders.”*; Introduction (line 48): *“Understanding how patients with pathological anxiety learn about threats within specific spatial locations in complex environments is essential to better understand the development and maintenance of the disorder.”*

Comment 26

For the analysis examining differences between flowers (conditioned) and objects (not conditioned): it seems to me that it would have been more appropriate not to collapse over safe and danger zones?

Response 26

We understand the Reviewer comment. Yet, we opted to collapse across zones for several reasons: 1) The main reason is that we added this control task within the main task to ensure our main findings were indeed related to learning and discrimination and not to movement and navigation confounds. Therefore, by having objects that were never paired with a shock and extremely different from the flower, we could test the difference between approach and interaction with emotional vs non-emotional cues; 2) Additionally, we used this control task to ensure participants were paying attention to the game. At the end of the task, we would ask them to name the objects they were collecting and their location. These would allow us to assess whether they were paying attention to the game. We also asked them to tell us the flower contingencies and rules of the environment as the end of the game. Combining this information allowed us to assess whether they were truly learners vs non-learners, or was it just that they were not paying attention and going through the game just to finish and get compensated; 3) Finally, there was no significant learning difference between objects in the safe and danger zone between groups. Since there were no shocks associated with these objects, and were completely neutral and different from the flower, we saw that both healthy controls and patients with anxiety disorders were able to learn the locations of the objects similarly at the same rate and to the same accuracy. Because of these reasons, we kept these analyses as pure as possible, by only looking at navigational differences when participants were approaching an object vs. a flower.

Comment 27

The bar plots in Fig 2 and Fig 3 need stars

Response 27

We have now added the stars to indicate significance to the figures and captions as follows
*“** $p < 0.05$ FWE; * $p < 0.05$ FWE SVC”*

Reviewer #2

Comment 1

There is no evidence that anxiety changes measures of stress in this task. The measures of stress presented (skin conductance and shock expectancy ratings) do not show a significant difference between groups (Fig S1; e.g. the lack of an effect ($p=0.06$) for ANX versus $p<0.01$ for HC does not provide evidence for a difference between groups and this is the case for both approach and stationary periods).

Response 1

We agree with the reviewer that we need better ways to evidence anxiety changes due to the task. To enrich our behavioral results, we added an analysis of the state-trait anxiety inventory (STAI; Spielberger, Gorsuch, Lushene, Vagg, & Jacobs, 1983), focusing on the state anxiety scale, which we collected before and after the task. The state anxiety scale has 20 self-reported items assessing anxiety experienced at the present moment. Therefore, we were able to explore changes in state anxiety from before to after completion of the VR task. We found that ANX participants had higher state anxiety scores compared with HC participants at post-task, while at pre-task, both groups showed similar state-anxiety levels.

We have added this to the Results section as follows (line 138): “*State-Trait Anxiety Inventory (STAI). Before and after the task participants filled out the state anxiety section of the STAI. A 2x2 ANOVA (group x time) analysis of the state anxiety (pre-task, post-task) revealed a significant group effect ($F(1, 22)=28.07, p=0.000$). No other significant main effects or interactions were observed (F 's $<2, p$'s >0.05). A t -test revealed that ANX had higher state anxiety than HC only at post-task time point (pre-task, $t(49)=2.56, p=0.116$; post-task, $t(49)=6.51, p=0.014$).”*

- Spielberger, C. D., Gorsuch, R. L., Lushene, R., Vagg, P. R., & Jacobs, G. A. (1983). Manual for the State-Trait Anxiety Inventory. Palo Alto, CA: Consulting Psychologists Press.

Additionally, we added the exploratory analysis of SCL, SCR, and expectancy ratings within each group. We now show that while both groups show similar expectancy ratings, only HC had significant differences between zones in the SCL and SCR. We have added exploratory analysis of the SCL and SCR of each group as follows (line 148): “*Based on the first hypothesis, that ANX would evidence overgeneralization, within-group main effects of zone were explored and indicated that only HCs had significantly higher SCL towards dangerous flowers as compared to safe flowers (HC, $t(27)=3.94, p<0.01$; ANX, $t(22)=1.99, p=0.06$; Figure 1 C and F).*” and “*Within-group main effects of zone were explored, showing that only HC's had significantly higher SCR towards dangerous flowers as compared to safe flowers (HC, $t(27)=2.42, p<0.05$; ANX, $t(22)=0.98, p>0.05$; Figure 1 D and G).*”

We believe that lack of behavioral differences could be explained in two ways: 1) due to compensatory mechanisms; or 2) maybe these are not the best behavioral measures for this task. Maybe the behavioral measures we selected doesn't give the breath of scope to measure the full behavior difference in anxiety disorders, maybe other measures such as eye-tracking would be better suited for this purpose. Increasing the task-related threat risk could also help flush out these differences in behavior, such as raising the stakes by adding a component where correct assessment of flowers reduces the chance of shock and vice-versa. Another possibility is that we don't have the statistical power to flush out the true behavioral differences using these measures. We now

further discuss the lack of clear behavioral difference findings in the discussion as follows (line 558): *“It is important to discuss that while ANX displayed a higher state anxiety at the end of the task both patients and HC learned the contingencies of the task (expectancy ratings). This finding was supported by the skin conductance results during the task (SCL; SCR), which were only significant at when we explored the groups separately. Exploratory analysis did suggest that only HC were showing skin conductance (SCL, SCR) differences between the zones while patients did not. However, as exploratory analysis this should be taken with caution. We can speculate that there are indeed skin conductance difference reflecting that patients learnt the contingencies appropriately through brain compensatory mechanisms. But, maybe these behavioral measures are not sufficient to capture the full spectrum of behavior between groups. We could not add the spatial trajectories or time spent in each zone to the behavioral analysis due to several factors of the task, 1) participants were forced to collect one flower at a time; 2) the flowers appeared randomly throughout the environment therefore they could be close or far from each other; 3) age, gender, and video game experience might have affected the task proficiency in general. Future studies should consider using other sorts of behavioral measures (e.g. eye-tracking), increasing the stakes of the shock contingency (e.g. correct expectancy ratings diminish the risk of shock), or increasing the sample size to increase power and therefore flush out the exploratory findings. Another component that might have affected the behavioral results is the inclusion of patients with social anxiety disorder (SAD), as the task did not include any social component. The study aimed to capture differences between HC and patients with any anxiety disorder. And while the study only recruited a minimal number of patients with SAD (7), their inclusion might have skewed the results more favorably to appropriate learning behavior. Future studies should focus on patients with generalized anxiety disorder to test this.”*

Comment 2

The authors do not report data related to the duration of approach in safe vs danger zones (or the length of the path taken, though these are likely correlated) which could be interesting but from the data shown, there is no evidence that threat learning, appraisal or anticipation differs between ANX and HC in this task. This is not explicitly mentioned in the main text until the discussion, despite it being one of the three key hypotheses tested in this manuscript.

Response 2

This is an excellent idea which we aimed to address and add to the behavioral findings originally. Accordingly, we have made the following changes to the original manuscript:

- 1) We have addressed the issues of spatial trajectory and time spent in zones (also discussed in detailed below) in the Discussion, adding them as possible limitations of performed analysis
- 2) We have expanded on the importance of the brain analyses and its replication of previous findings with the same task.

1) Spatial trajectory and time spent in the zones limitations:

While we did collect data on participants' navigation and orientation, several factors have precluded our ability to adequately use it.

A. Participants' characteristics: While some people learned very fast how to use the controllers and navigated the environment, some learned more slowly but still learned it eventually (at different time points during the task), and others were very slow throughout the experiment. Hence, these differences also affected both the trajectory and the time spent in zones. However,

these differences in learning rates could have been the result of several factors, not related to group differences. These include:

- iv. Participants' experience with video games: We suspect that people with more experience playing video games, particularly those requiring navigation of some sort, would learn faster how to use the controllers and navigate the environment of the game, while people with no experience would take longer to learn the navigation, if at all. Unfortunately, we did not collect information about gaming experience or type of gaming experience. Most information was anecdotal gained through informal conversations with participants after the task, when they would mention never playing video games or saying how it reminded them of another game they had played.
- v. Age: We noticed that in general, older people were more likely to be slower than younger people, although this was not always the case. Related to our previous point, older people are also less likely to have video game experience than younger people.
- vi. Gender: Females, again not all, tended to be slower than males. However, again, this could also be linked to gaming experience, as females are less likely to play video games than males (though this is changing fast in younger generations and with the accessibility of video games on cellphones).

B. Task characteristics: While flowers were counterbalanced between zones (20 in each zone), their location was randomized. Flowers could appear close to each other either at the boundary between zones or within the same zone. Or, they could appear far from each other, at different side of the environment, again, between different zones or within the same zone. Flowers were allowed to appear in the same zone, though we capped this at a maximum of 3 flowers in the same zone to increase movement between zones. Overall, these factors could affect time spent within or between zone, making it hard to determine whether participants spend more or less time in a zone due to conscious thought or due to the randomness of the experiment. These factors also affect spatial trajectory. Relatedly, sometimes flowers, particularly if they were very far away, might have been difficult for participants to see initially, increasing the time of orientation. And while we told participants at the beginning of the experiment "If you do not see the flower, try walking across the environment as it might be easier to find" participants used different strategies to find the flowers and not always used the walk across strategy to find them. Importantly, as the task was designed to measure spatial learning and discrimination of zones, not avoidance, approach, or conditioning, we did not account for the differences in spatial trajectory (navigation) when designing the experiment. So, while everyone learnt the contingencies differently we were more interested in the brain areas engaged during cognitive mapping of the environment to discriminate between zones. This is why we measure brain activity during approaching the flowers and after picking them (during restriction of movement). Participants indeed had to collect all the flowers to complete the study, everyone had to collect the 40 flowers in the danger zone and 40 flowers in the safe zone. And participants were forced to go towards the flower, there was not a selection of flowers for them to pick from. There was always one flower present in the environment, which they had to pick to make another one appears. Therefore, some people might be slow to pick the flower (as to avoid potential shock) but other might want to get it over with and therefore rush towards them. But we did not ask about this strategy or thought process. We have clarified this in the results section as follows (line 118): "*For each trial, participants freely explored the environment and were instructed to pick up flowers that appeared one at a time in random locations across the*

environment (approach period). When a participant picked a flower, their position was held stationary for a variable duration (2-8 seconds; stationary period), during which the participant rated the expectancy of receiving a shock (rating of 0-9). After the stationary period a new flower would appear for the participant to find. There was only one flower in the environment at a time.”

In sum, while we do agree with the reviewers’ ideas, due to the above-stated reasons, a clear and coherent conclusion based on this data might be difficult and possibly even misleading. Nonetheless, if the reviewer still wants us to run and add these analyses to the manuscript, we would be happy to oblige. However, for our future studies we are planning to look more closely at the approach/avoidance question through additional VR tasks we are developing, including eye-tracking to clearly assess gaze-navigation patterns. We will also collect more information on video game experience, including VR specifically, to correct these limitations. Yet, as we do concur with the reviewer, we now added some of the above-detailed information alongside an interpretation for the lack of behavioral findings to the discussion as follows (line 566): *“We could not add the spatial trajectories or time spent in each zone to the behavioral analysis due to several factors of the task, 1) participants were forced to collect one flower at a time; 2) the flowers appeared randomly throughout the environment therefore they could be close or far from each other; 3) age, gender, and video game experience might have affected the task proficiency in general. Future studies should consider using other sorts of behavioral measures (e.g. eye-tracking), increasing the stakes of the shock contingency (e.g. correct expectancy ratings diminish the risk of shock), or increasing the sample size to increase power and therefore flush out the exploratory findings.”*

2) Brain analysis importance:

Most importantly, while we don’t show a strong behavioral difference between groups, this study shows differences in brain activity which are not confounded by behavioral differences. And, in this time of replicability issues, we note that we were able to replicate our initial HC findings. Furthermore, the differences between groups are in the brain areas we identified in our initial study for learning to discriminate between zones and for tracking threat. This suggest that if there are indeed no behavioral differences between groups, the ANX group is engaging the same brain areas (albeit differently) as compensatory mechanisms to rescue the appropriate behavior. We have added more about the current HC results and the previous paper on the same task in the discussion as follows (line 444): *“Several points are noteworthy when considering the current findings in light of the previous study in healthy controls using the same task. In both task we have identified the same network of activation in HC’s when approaching and interacting with flowers. We replicated the two neural networks to support the formation of cognitive maps and its relevant behavior: 1) The discrimination learning network (DLN; vmPFC, PCC, anterior hippocampus) activation increased when HC’s learn to discriminate threatening and safe zones within the environment; 2) the salience network (SN; dmPFC/dACC, insula, PAG) activation increased when participants approached and interacted with flowers in the danger zone. There is one main difference between this and the past study in the SN, in the current study the peak activation was more in the dmPFC while in the previous study the peak was more in the dACC. However, the brain activation cluster of both studies often overlap between the two areas (dmPFC/dACC). Furthermore, while some brain activation differences found in the current study are using a priori hypothesis SVC based*

on the past study, all the group brain activation differences (HC, ANX) of the current study are in the areas identified in the previous study (DLN, SN).”

Comment 3

there is no figure showing these behavioural results in the main manuscript.

Response 3

As recommended, we have added figures for the expectancy ratings, SCR, SCL (Fig 1), and object placing error task (Fig 2).

Comment 4

Unfortunately, this lack of a behavioural effect makes subsequent differences in BOLD difficult to interpret. Currently, the authors offer little to reconcile the absence of behavioural/skin conductance responses with the presence of many BOLD differences. In fact, in the discussion the authors state “In patients with pathological anxiety, reintegrating the anterior hippocampus and vmPFC into the circuitry could rescue appropriate learning strategies to discriminate between safety and danger.” This statement does not seem to match up with the results of the manuscript where anxious participants had no problem discriminating between safety and danger.

Response 4

We thank the reviewer for this comment which we agree with. To enhance the interpretation of the results and to reconcile the absence of behavioral responses to the BOLD we have added a paragraph to the discussion as follows (line 558): *“It is important to discuss that while ANX displayed a higher state anxiety at the end of the task both patients and HC learned the contingencies of the task (expectancy ratings). This finding was supported by the skin conductance results during the task (SCL; SCR), which were only significant at when we explored the groups separately. Exploratory analysis did suggest that only HC were showing skin conductance (SCL, SCR) differences between the zones while patients did not. However, as exploratory analysis this should be taken with caution. We can speculate that there are indeed skin conductance difference reflecting that patients learnt the contingencies appropriately through brain compensatory mechanisms. But, maybe these behavioral measures are not sufficient to capture the full spectrum of behavior between groups.*

Most importantly, while we don't show a strong behavioral difference between groups, this study has strong brain activity implication. Mainly, in this time of replicability issues, we were able to replicate our initial HC findings. Furthermore, the differences between groups are in the brain areas we identified in our initial study for learning to discriminate between zones and for tracking threat. This suggest that if there are indeed no behavioral differences between groups, the ANX group is engaging the same brain areas, albeit with different activity patterns, as compensatory mechanisms to rescue the appropriate behavior. We have added more about the current HC results and the previous paper on the same task in the discussion as follows (line 444): *“Several points are noteworthy when considering the current findings in light of the previous study in healthy controls using the same task. In both task we have identified the same network of activation in HC's when approaching and interacting with flowers. We replicated the two neural networks to support the formation of cognitive maps and its relevant behavior: 1) The discrimination learning network (DLN; vmPFC, PCC, anterior hippocampus) activation increased when HC's learn to discriminate threatening and safe zones within the environment; 2) the salience network (SN; dmPFC/dACC, insula, PAG) activation increased when participants approached and interacted*

with flowers in the danger zone. There is one main difference between this and the past study in the SN, in the current study the peak activation was more in the dmPFC while in the previous study the peak was more in the dACC. However, the brain activation cluster of both studies often overlap between the two areas (dmPFC/dACC). Furthermore, while some brain activation differences found in the current study are using a priori hypothesis SVC based on the past study, all the group brain activation differences (HC, ANX) of the current study are in the areas identified in the previous study (DLN, SN).”

Finally, we have revised the statement “In patients with pathological anxiety, reintegrating the anterior hippocampus and vmPFC into the circuitry could rescue appropriate learning strategies to discriminate between safety and danger” to clarify our meaning and interpretation of the results based on the behavioral findings as follows (line 435): *“These results suggest that patients with pathological anxiety recruit compensatory learning strategies that do not involve the anterior hippocampus or vmPFC, which might explain the increased activation in the salience network (e.g., dmPFC, insula, PAG). However, in this task the participants were forced to collect the flowers and therefore got the opportunity to know the contingencies of the zones. Real life scenarios don’t often offer that possibility, particularly if the patient prefers to avoid it, creating higher uncertainty and risk. We believe that reintegrating the anterior hippocampus and vmPFC into the circuitry could rescue appropriate learning strategies to discriminate between safety and danger, particularly when information and experience is limited.”*

Comment 5

At the moment, the results are very difficult to follow. The statistical methods and precise contrasts used are often unclear and described inconsistently (e.g. between figures and main text and methods and supplement) and as a consequence the meaning and validity of the results are difficult to judge.

I will give one example, but this applies throughout. In line 153, it sounds like the contrast tested is across both zones and task blocks and thus an overall difference between ANX and HC in the approach period of the task. This is what the abstract suggests as well (global decreased activation during the task), but I don’t think this is what was done. Instead, I think the authors performed a full interaction between blocks, groups and zones (as suggested by Table S1). The next sentence (line 157) talks about comparisons between late>early blocks and safe vs danger zones making it sound different from the abstract. Is this a qualification of the previous sentence in line 153 or a new result? It is really important to be absolutely clear in the text and figures which contrasts were performed. This is also important for understanding from which contrast peak coordinates were extracted. Currently, it is hard to know whether the bar-plots presented throughout all figures are mere illustrations of the main contrasts shown above, or whether statistical post-hoc tests were performed. It is therefore not possible to be sure about any potential circularity that might be present. The authors talk about post-hoc tests but don’t present any statistics on these. Line 157 is one an example of such a statement which is statistically very vague, “Compared to the HC group, the ANX group demonstrated a greater increase in activity in these areas from early to late blocks of the safe zone and a decrease in activity from early to late blacks (note typo) of the danger zone.” It seems that the authors have performed multiple tests on multiple brain areas here, but no statistical result is reported. And the Figure legend related to this result, starts “For ANX, activation... “ making it sound like this was a contrast estimated for ANX only (rather than what

I think it was namely an interaction between ANX>HC and block and zone). Please spell out much more clearly throughout what hypotheses are being tested, which precise tests were performed, what statistical tests including post-hoc tests were performed, and make sure conclusions stated in the abstract, discussion etc. match up with the main text.

Response 5

We deeply apologize for the lack of clarity and inconsistencies throughout the manuscript. We have extensively revised all of these and reworked several sections of the manuscript to increase consistency and clarity. Throughout the Results section we aimed to be more explicit about the contrasts or interactions being done. We truly hope the reviewer finds the revised manuscript sufficiently clear and coherent.

As per the example given by the reviewer, it is correct, we performed an interaction, then extracted the time series to observe the directionality. This interaction was followed by main effect contrasts. We have revised this throughout the results and methods.

For example, we rewrote the previous line 153 as follows (line 233): “*A zone by block by group interaction of approach periods identified two opposing patterns of activation changes in a range of areas comprising posterior cingulate cortex (PCC; $p < 0.05$ FWE), vmPFC, orbitofrontal cortex (OFC)/subgenual anterior cingulate cortex (sACC), and bilateral anterior hippocampus ($p < 0.05$ FWE SVC, Fig. 3A).*” We revised the previous line 157 as follows (line 236): “*When extracting the time series, we observed that the ANX group, compared to the HC group, demonstrated a greater increase in activation in these areas from early to late blocks of the safe zone (late > early; safe > danger) and a decrease in activation from early to late blocks of the danger zone (early > late; danger > safe). To understand these distinct patterns of activation, we performed direct group comparisons on separate components of the task.*” We also rewrote the figure caption to (line 259): “*A Zone by block by group interaction shows two opposing patterns of activation between groups in the posterior cingulate cortex (PCC; $p < 0.05$ FWE), ventromedial prefrontal cortex (vmPFC), orbitofrontal cortex/subgenual anterior cingulate cortex (OFC/sACC; top left panel), and anterior hippocampus ($p < 0.05$ FWE SVC; top right panel).*” Finally, we rewrote the methods to reflect this as follows: “*For any significant interaction, the representative time-course was extracted through SPM12 MarsBaR (<http://marsbar.sourceforge.net>) toolbox, using a 6mm sphere centered on the peak of the activation in the regions of interest, using the first eigenvariate calculated from singular value decomposition. The extracted values were used to illustrate and observe the directionality of the results.*”

Comment 6

A similar point holds for the PPI analyses, it is unclear what exact hypotheses were tested (if any) and which precise analyses were conducted. It seems that the authors use any main activation they find to subsequently run a PPI on. This not only results in many tests but the results are hard to follow, and no figures are shown for any of the PPI results. The authors state, for example “PPI examined brain connectivity of each significant cluster (i.e., seed ROI) from the approaching flowers period.” This is very vague, there are three contrasts reported in the supplement for the approach period, so which one are the authors talking about, and which/how many regions were taken as seeds?

Response 6

We apologize for the lack of clarity of performing these analyses. Overall, we were interested in seeing if the task and the physiology changes the task created (particularly in anxiety patients) correlated with patterns of brain activity of the dmPFC in the approach and stationary periods. That is, we were interested in exploring if the dmPFC brain activation correlated with other brain areas to predict learning and discrimination.

During the approach period this was performed to understand threat-appraisal as participants approach the flowers (anxiety states). We have further explained this in the results section as follows (line 275): *“Given that dmPFC activation was higher in the ANX compared to the HC across the approaching period, particularly in the danger zone, we were interested to see how the task and the physiological state it caused in the participants interacted with the brain activity to further understand brain-behavior associations of anxiety-states (threat appraisal). For this purpose, we used a psychophysiological interaction (PPI) analyses for each participant group separately to identify dmPFC patterns which connectivity changed during the danger vs. safe contrast. PPI examined the brain connectivity of the significant dmPFC cluster (i.e., seed ROI) from the approaching flower period.”*

During the stationary period this was performed to understand threat-anticipation as participants interacted with flowers (fear states). We have further explained this in the results section as follows (line 346): *“Given that dmPFC and amygdala activation was consistently higher in the ANX compared to the HC across the stationary period, particularly in the danger zone, we were interested to see how the task and the physiological state it caused in the participants interacted with the brain activity to further understand brain-behavior associations of fear-states (threat anticipation). For this purpose, we used a psychophysiological interaction (PPI) analyses for each participant group separately to identify dmPFC and amygdala patterns which connectivity changed during the danger vs. safe contrast. PPI examined the brain connectivity of the significant dmPFC and amygdala cluster (i.e., seed ROI) from the stationary period.”*

Comment 7

For the PPI, the authors also don't fully justify the choice of their statistical threshold ($p < 0.001$ Bonferroni) which differs from the rest of the manuscript. They refer to one of their own previous papers for justification which is not sufficient.

Response 7

We have now added more justification for the $p < 0.001$ Bonferroni correction. In sum, while PPI analysis are known for having increased false negatives often due to lack of power (O'Reilly et al 2012; see below), we wanted to make sure that the results we were getting were strong and significant. Hence we selected a more stringent approach. The sentence in the methods now reads as follows (line 746): *“As in a previous study, and to increase the strength and precision of the analysis, the PPIs were detected using a t-test with a threshold of $p < 0.001$ corrected for multiple comparisons.”*

- O'Reilly, J. X., Woolrich, M. W., Behrens, T. E., Smith, S. M., & Johansen-Berg, H. (2012). Tools of the trade: psychophysiological interactions and functional connectivity. *Social cognitive and affective neuroscience*, 7(5), 604–609. <https://doi.org/10.1093/scan/nss055>

Comment 8

Finally, PPI results cannot be interpreted as different between groups if direct comparisons between groups are not performed (as I understand it, PPI results were done on HC and ANX separately) e.g. see discussion: “Unlike healthy comparisons, patients with pathological anxiety failed to exhibit task-related functional connectivity with higher-level cortical areas related to emotional regulation (vmPFC and OFC) throughout the task.”

Response 8

As noted by the reviewer the PPI results were done on HC and ANX separately. A limitation in PPI analysis is that results between groups are often driven by group differences on the main analysis. That is, since we found overall higher dmPFC in ANX > HC, it is very likely that we will find all PPI analysis related to the dmPFC higher in ANX than HC, and would occlude the PPI results of the HC. Because of this we opted for doing the analysis separately, conducting direct group comparisons that allowed us to understand the correlation between areas in both groups. In addition, this allowed us to replicate the original findings of the HC in the previous study.

We agree that we used a poor choice of words when using “Unlike HC”, as we meant to say that HCs show a correlation in activity between these areas, while ANX do not. We did not mean to imply that a direct comparison was made. To increase the clarity and transparency on this point we have rewritten that section and now reads as follows (line 538): *“Patients with pathological anxiety failed to exhibit task-related functional connectivity with higher-level cortical areas related to emotional regulation (vmPFC and OFC) throughout the task. During the flower approach, patients with pathological anxiety displayed negative functional connectivity between the dmPFC and both the vmPFC and OFC. During stationary periods in the danger zone, patients with pathological anxiety displayed positive dmPFC-insula and amygdala-insula functional connectivity. On the other hand, healthy controls, displayed functional connectivity of the dmPFC to the OFC and vmPFC. And while groups were not directly compared, these findings suggest that ANX and HC are engaging in different networks of brain activity when engaging with the task. Coupling between the dmPFC and amygdala has been previously observed with induced threat of shock^{25,34,35} and in patients with pathological anxiety³⁶ during an emotional identification task...”*

Comment 9

It is also unclear if all or just a chosen subset of all possible contrasts were looked at (given the available GLM regressors). For instance, there are several that I think it would be interesting to look at, but I am unsure if they have been conducted (e.g. comparing ANX and HC averaged across zones and block, or just contrasting danger and safe zones independent of block between the groups).

Response 9

We apologize for the lack of clarity in which contrasts were used.

Indeed, we started with a zone by block by group interaction, and then proceeded to look at the main effects of zone and block by group. These main effects are the averages across zone and block. For example, the main effect of block takes the average of safe and danger zone and just looks at early vs late. In the case of the main effect of zone, it takes the average of early and late block and just looks at safe vs danger. We have now made notes of the contrasts by explicitly

stating them in parenthesis, for example “(safe > danger)” to indicate that this was a safe vs. danger contrast where we looked at brain activity which were higher in safe compared to danger zones.

To further increase the clarity of the motivation of the analysis we have also added further explanation before each of the analysis. For example, in the approach period analysis we have added (line 242): “*To understand how brain activation differed when approaching flowers in each of the zones we examined the zones contrasts (danger, safe) between groups. This analysis takes the average of the early and late regressor and looks at the zone effects.*” At the beginning of the paragraph to illustrate that we were delving into the zone contrast. We did a similar approach for the block contrast as follows (line 251): “*To further understand brain activation differences as participants learnt the contingencies of the task we examined the block contrasts (early, late) between groups. This analysis takes the average of the safe and danger regressor and looks at the block effects.*” We have applied the same approach to the stationary period and to the object placing period analyses.

Comment 10

The figures seem to show a somewhat incomplete selection of areas out of all those that are mentioned as significant in the text and it is unclear why peak activations are not shown for e.g. vmPFC and PCC In Figure 2A and insula in Figure 2B. Were the patterns the same as in the other regions shown or was there a reason for omitting them?

Response 10

The reviewer is right, we selected a couple of areas to illustrate the patterns of activation as to not overcrowd the figures with all the significant regions, especially since the activity pattern was the same. To be completely transparent and to improve clarity of our methods we added the following statement to the figure caption: “*Only a subset of percentage signal change graphs are shown to illustrate the pattern of activation, similar pattern of activation were observed in the other relevant areas per contrast.*”

Comment 11

It seems like a large number of participants are excluded due to poor task performance (14 out of 70) but one of the criteria was their performance recalling object’s names in a task not of interest to this study. How many participants would be excluded if only the flower contingencies relevant to the task at hand were used as a criterion and would that change any of the conclusions?

Response 11

We apologize for the confusion and we realized there was an error on our part. We recruited for this protocol 30 participants in each group. We had by mistake counted 10 HC participants, which were not used in the analysis, from when we were testing and piloting the task in the new location (NIMH) from the previous study site (UCL). These 10 subjects were removed not only because they were from a pilot study to set-up the task and train the staff in the new location, but also because of the nature of setting-up the new protocol we had many technical issues as the staff was learning to do this very complex protocol.

But to answer the question at hand, while we did use the object names and location as a measure for attention to the task (since the objects were common items like a book or a clock, which should be easy to remember), all the participants who could not remember the location or names of these objects also failed to explicitly state the contingencies of the flower task. We had no subject who

could state the contingencies of the flowers but not the names or location of the object task. Therefore, all subjects excluded were not able to state the flower contingencies relevant to the task, which would keep the results and conclusions the same. We have revised this in the methods as follows (line 608): “*Four ANX were excluded from analyses because they were unable to explain the shock contingencies between the locations at the end of the task (see procedure below).*”

Comment 12

The BOLD results between approach and stationary periods seem quite similar, at least judging from the text (this does not match the figures for the stationary period). Please report the correlations between all regressors in the BOLD GLM to show that they were independently estimable (e.g. as a supplementary figure). While this should be done either way, I do wonder if this was a copy and paste error? Lines 157ff and 207ff are identical even though they describe results from two different phases, and they don’t match up with the results that are shown for the stationary period in Figure 3.

Response 12

Again, we apologize for the lack of clarity. We hope with the changes we have done (see above) we have been able to increase clarity throughout the paper for the analysis done and illustrated. In this case we found a zone by block by group interaction in the approach (for PCC, vmPFC, OFC, aHPC) and stationary (for PCC, vmPFC, OFC) periods. Both of these interactions behave similarly with increase over time for the safe zone and decrease over time for the danger zone in ANX. Since they behave similarly, and to avoid repetition in the figures and overcrowding, we decided to illustrate this interaction in Figure 2 only for approach. Figure 3 illustrates the average zone effects between groups during the stationary periods. We aimed to illustrate, for example, that even though ANX had increase activity in OFC in the safe zone, HC still had higher overall activity than ANX.

Comment 13

Please specify precise p-values for all statistical tests.

Response 13

As requested by the reviewer, we have added precise p-values for all the statistical tests where possible, mainly this covers all the behavioral data. Z-scores have been provided for all the fMRI data peak activity in the supplementary tables.

Comment 14

Please tone down direct link to treatments or explain how the results would be directly linked to treatments (e.g. abstract, introduction)

Response 14

As recommended, we have removed this from both the introduction and abstract. We have changed the sentences to the following, in the abstract: “*Elucidating neural mechanisms of anxiety is essential to understand the development and maintenance of anxiety disorders.*”; Introduction (line 48): “*Understanding how patients with pathological anxiety learn about threats within specific spatial locations in complex environments is essential to better understand the development and maintenance of the disorder.*”

Comment 15

Were participants forced to pick all flowers even if they were sure that they were in the danger zone. Are they slower approaching/engaging/picking in later blocks when they know a flower is in the danger zone?

Response 15

We appreciate this comment that helped us clarify the task used in the study. To answer the reviewer's question - Yes, participants were forced to go towards the flower, there was not a selection of flowers for them to pick from. There was always one flower present in the environment, which they had to pick to make another one appear. And while we were interested in evaluating navigational path and time to approach, there were many factors (as mentioned above; see Comment 2) that impeded us from doing so and extracting meaningful interpretation of this data. For example, some people might be slow to pick the flower (as to avoid potential shock) but other might want to get it over with and therefore rush towards them. We agree that these sorts of analysis would have been a nice addition to explore behavioral differences between groups, however, the purpose of this study was not to address approach/avoidance differences but rather to understand spatial mapping and discrimination. We are presently creating additional VR tasks to look at these questions of approach/avoidance. To increase the clarity of this point in the paper, we added more on this to the results section as follows (line 118): *“For each trial, participants freely explored the environment and were instructed to pick up flowers that appeared one at a time in random locations across the environment (approach period). When a participant picked a flower, their position was held stationary for a variable duration (2-8 seconds; stationary period), during which the participant rated the expectancy of receiving a shock (rating of 0-9). After the stationary period a new flower would appear for the participant to find. There was only one flower in the environment at a time.”*

Comment 16

Please specify what time exactly contributes towards the stationary period e.g. for skin conductance analyses. Since the duration is variable (2-8s), are you just using the common 2s, or the entire duration?

Response 16

We used the entire duration to ensure capturing the whole window of behavior during that time. To increase clarity, we added more on this to the results as follows (line 135): *“Each stationary period (after collecting the flower) began upon touching the flower, during this time the participant movement would be stopped for 2-8 secs. Stationary periods were assessed for the entire 2-8 secs duration.”* And in (line 294): *“We use this stationary period (entire 2-8 secs) to understand brain activation differences between groups as participants were interacting with the flowers and anticipating potential threat of shock delivery.”* And in the methods as follows: *“Second, skin conductance responses (SCR) were analyzed during the stationary period (entire 2-8 secs) to examine phasic changes in anticipation of the shock outcome.”*

Comment 17

Please be more precise about recruitment – line 393ff: it sounds a bit like 70 people were recruited in total and by chance 30 were ANX and 40 HC?

Response 17

We apologize for the error, as mentioned above (see Comment 11), we recruited for this protocol 30 in each group. We had by mistake counted 10 HC participants, which were not used in the

analysis, from when we were testing and piloting the task in the new location (NIMH) from the previous study site (UCL). This is why we had a large group of participants that were “dropped” from the analysis.

We have revised and corrected this in the methods as follows (line 600): *“Sixty participants, aged 18-50 years, were recruited from the Washington D.C. and Maryland areas. Thirty participants were diagnosed with an anxiety disorder (ANX; generalized anxiety disorder (n=8), social anxiety disorder (n=7), and co-morbidity between the two (n=8)) using the Structured Clinical Interview for Diagnostic and Statistical Manual of Mental Disorders 4th edition, while the other 30 volunteers were healthy controls (HC)... Four ANX were excluded from analyses because they were unable to explain the shock contingencies between the locations at the end of the task (see procedure below). Three ANX and two HC were omitted due to excessive head motion (>20%; see below). Therefore, the final sample included 23 ANX (mean age=29.61; SD=8.22) and 28 HC (mean age=27.25; SD=8.21).”*

We have also added a supplementary table 4 to breakdown the demographics of the population.

Comment 18

Anatomical labels: what is circled as OFC is on the medial surface and would usually be referred to as subgenual cingulate cortex or BA25.

Response 18

We agree that part of the activation (particularly as illustrated) is in the subgenual anterior cingulate cortex (sACC). However, peak activities for those clusters are often in the OFC as well. To address this point, we have changed and revised to sACC, and OFC/sACC where appropriate in the manuscript, figures, figure captions, and supplementary material.

Comment 19

Line 490: were there three or four regressors for modelling the end of each trial; three makes sense but that makes the overall number of regressors 15 rather than the 16 stated

Response 19

We thank the reviewer for catching this discrepancy. Indeed, we used three regressors for modelling the end of each trial. We have now fixed the stated 16 to reflect the accurate 15 regressors.

Comment 20

There are grammar mistakes and typos throughout (e.g. line 87 “but this not clear if this extends to context”; “MNI coordinated”, contrast “dange > safe”, “blacks” instead of blocks)

Response 20

We apologize for the typos and appreciate the reviewer for pointing them out. We have revised the manuscript throughout for further grammar mistakes.

Comment 21

“Healthy comparisons” is an unusual term to refer to healthy participants, usually, “healthy controls” would be used instead

Response 21

We have changed the term to healthy controls throughout the paper.

Reviewer #3

Comment 1

First, the authors may want to avoid the impression of another just-so story in cognitive psychiatry – which it is not – by being more rigorous with the description of the analysis pipeline and the derivation of their hypotheses. This is a well-designed experiment, based on a previous study in healthy volunteers, and with almost the exact same design and even analysis pipeline. Still, there are some discrepancies with their earlier work (e.g. 16 instead of 15 regressors in the first-level design). To avoid any (certainly incorrect) impression that analysis was flexibly adjusted so that results "make sense", I would suggest a more systematic discussion of where the analyses deviates from the previous study, and why.

Response 1

We truly appreciate and thank the reviewer's positive impressions of the paper. And we apologize for the confusion about differences between this work and the previous one. There was a mistake in the count of regressors which should have been 15 instead of 16. The whole analysis section was purposely designed to be the same as the previous work, with the exception of comparing between patients and controls. We have revised this mistake in the methods section. Nonetheless, we have expanded our discussion on the similarities and differences between the two studies and added more in the importance of this paper in the groups differences as follows (line 444): *“Several points are noteworthy when considering the current findings in light of the previous study in healthy controls using the same task. In both task we have identified the same network of activation in HC's when approaching and interacting with flowers. We replicated the two neural networks to support the formation of cognitive maps and its relevant behavior: 1) The discrimination learning network (DLN; vmPFC, PCC, anterior hippocampus) activation increased when HC's learn to discriminate threatening and safe zones within the environment; 2) the salience network (SN; dmPFC/dACC, insula, PAG) activation increased when participants approached and interacted with flowers in the danger zone. There is one main difference between this and the past study in the SN, in the current study the peak activation was more in the dmPFC while in the previous study the peak was more in the dACC. However, the brain activation cluster of both studies often overlap between the two areas (dmPFC/dACC). Furthermore, while some brain activation differences found in the current study are using a priori hypothesis SVC based on the past study, all the group brain activation differences (HC, ANX) of the current study are in the areas identified in the previous study (DLN, SN).”*

Comment 2

Furthermore, it would be good to give a more nuanced derivation of the hypotheses, in particular of the regions of interest on which the authors focus. With respect to the behavioural hypothesis, the authors write that "Research has shown that individuals with anxiety disorders display a higher defensive response to a CS+, compared to healthy individuals, and they often generalize the threat response to safe cues (CS-) ". This is backed up with selected and slightly old primary and review papers, but there is a large literature on this topic and an exhaustive meta-analysis by Duits et al. 2015 with more than 2000 participants shows a more nuanced picture. This renders the behavioural hypothesis, which is based on this idea, somewhat brittle. The behavioural hypothesis is taken to support the first fMRI hypothesis (compensatory learning strategies not involving the hippocampus), which hence also appears dubious. Regarding the other two hypotheses, I was unsure where they come from, and I just note that they look a lot like the results.

Response 2

We thank the reviewer for the recommendation that assisted us in improving our manuscript.

We have rewritten that sentence and added the suggested citation as follows: “*Research has shown that individuals with anxiety disorders display a higher defensive response to a safe cue (CS-), compared to healthy individuals, suggesting impaired ability to regulate their emotions or a generalization of the threat response to safe cues*” to further drive the point of overgeneralization we have added three studies from the Lissek group as follows (line 90): “*A discrimination of rings task has been used to systematically elucidate neural signatures of generalization/discrimination in patients with panic disorder, generalized anxiety disorder, and PTSD. These studies show that patients, compared to healthy controls, exhibit an overgeneralization, or lack of discrimination, towards cues similar to the CS+. But it is not clear if this overgeneralization extends to context, particularly when there is no clear-cut boundary between safe and danger zones within the environment.*”

- Lissek S, Rabin S, Heller RE, et al. Overgeneralization of conditioned fear as a pathogenic marker of panic disorder. *Am J Psychiatry*. 2010;167(1):47-55.
- Lissek S, Kaczkurkin AN, Rabin S, Geraci M, Pine DS, Grillon C. Generalized anxiety disorder is associated with overgeneralization of classically conditioned fear. *Biol Psychiatry*. 2014;75(11):909-915.
- Kaczkurkin AN, Burton PC, Chazin SM, et al. Neural Substrates of Overgeneralized Conditioned Fear in PTSD. *Am J Psychiatry*. 2017;174(2):125-134.

The next two hypothesis are based on this notion of higher arousal to CS- suggesting overgeneralization and from the previous study in healthy control. In other words, overgeneralization, due to lack of engagement of the hippocampus and vmPFC (modulatory areas), would lead to dysregulated threat detection and valuation areas (e.g. dmPFC). We have tried to better reflect this as follows (line 104): “*Using the virtual environment paradigm described above, three main hypotheses are tested. We expect that individuals with an anxiety disorder (ANX; generalized anxiety disorder, social anxiety disorder) compared with a sample of healthy controls (HC) will show the following: (1) Threat learning: poor learning and discrimination associated with compensatory learning strategies that do not involve the hippocampus leading to generalization of threat, reflected during, (2) Threat appraisal: stronger engagement of the dACC, dmPFC, amygdala, and insula in danger zones when approaching the flowers (higher anxiety-state), and (3) Threat anticipation: stronger engagement of the periaqueductal gray matter (PAG) activation in danger zones in anticipation of potential shock delivery (higher fear-state).*”

Comment 3

Second, the hypotheses are rather unspecific and mention at least 6 regions/contrasts where the authors expect to find an effect to support their general theory. This multiplicity requires a strong strategy to guarantee that multiple hypothesis testing does not lead to inflated false positive rates. While I gather from the results that this is quite likely not the case and the conclusions are probably valid, this should be spelt out in more detail. Will the authors conclude in favour of their theory only if all these 6 tests are positive, or also if one or two are positive?

Response 3

We understand the reviewers concern and we concur that it is not the case. While we are interested in 6 regions/contrasts we performed whole brain analysis, and we had only 1 small volume

correction map which included the 6 ROI of interest. Outside of this, we used whole-brain family-wise error correction, which is one of the most stringent correction used in fMRI analysis. This ensures that we don't inflate false positives, in any case, we might be inflating false negatives. But we wanted to ensure the accuracy and strength of the results (particularly as we used a SVC mask in the ROIs). Nonetheless, we were not as stringent as to expect all 6 test to be positive, we were interested in how these ROIs (and whole brain) behaved during the task when comparing between groups. We were pleased to see that we were able to replicate our initial results from the previous study in HC and that these same areas were differently implicated in the ANX group.

Comment 4

More technically but related to this point, it is not always clear a priori when the authors will do whole-brain correction and when they will do small volume correction, and how they will correct for the number of small regions that being corrected for.

Response 4

We deeply apologize for the lack of clarity when describing the analysis. All initial analyses were done using FWE whole-brain correction. But, since some areas of the medial prefrontal cortex (mPFC) and hippocampus did not survive this stringent correction, we opted for using FWE ROI-based corrections using a single mask. The vmPFC and hippocampus are important areas implicated in our previous study and central to the hypothesis of the current paper. We have revised the language used throughout the manuscript to describe the brain analysis done.

We have added (line 228): *“All statistical values reported are FWE whole-brain corrected ($p < 0.05$). However, given our a priori hypothesis, additional FWE small volume correction (SVC) was performed when areas of interest (hippocampus, amygdala, and mPFC) did not survive FWE whole-brain correction. One bilateral mask, which included the hippocampus, amygdala, and mPFC, was used for the SVC analysis.”* to make it clearer before describing the results.

To clarify that we only used one mask for the SVC analysis we also edited the sentence in the methods describing these analyses to (line 727): *“One bilateral mask was defined which included the hippocampus, amygdala, and mPFC (orbitofrontal gyrus, ventromedial prefrontal cortex, dorsomedial prefrontal cortex, and anterior cingulate and medial cingulate cortex). Statistical threshold in this single mask was defined by small volume correction (SVC; $p < 0.05$ FWE).”* Therefore, no multiple comparison correction was used in the SVC analysis.

Comment 5

Another technical point is their negative behavioural finding. Could it be that they do not find an impairment in contextual conditioning because they use a relatively non-standard and probably not the most sensitive scoring strategy? Sure the field has moved beyond peak-scoring in general, and study-specific peak scoring windows in particular?

Response 5

We initially thought the same way, and we did try using several scoring strategies for the skin conductance data such as peak-scoring in general, average scoring from the whole period, average scoring from the windowed period, using the max or the min from each participant to normalize the data, log equations, etc. Surprisingly, all came to the same result, of no group difference. The method we employed here was the same one we used in our previous study, which coincidentally was the one which showed the clearest pattern of behavior across groups. Still, maybe skin

conductance in this task is not the most sensitive to pick group differences, particularly since ANX were able to learn the environment contingencies. For future studies we are planning to use eye-tracking in this VR tasks and to collect more information on video game experience and the experience during the VR task to look at navigational patters which we were not able to analyze in this study. We have added some of this information alongside an interpretation for the lack of behavioral findings to the discussion as follows (line 558): *“It is important to discuss that while ANX displayed a higher state anxiety at the end of the task both patients and HC learned the contingencies of the task (expectancy ratings). This finding was supported by the skin conductance results during the task (SCL; SCR), which were only significant at when we explored the groups separately. Exploratory analysis did suggest that only HC were showing skin conductance (SCL, SCR) differences between the zones while patients did not. However, as exploratory analysis this should be taken with caution. We can speculate that there are indeed skin conductance difference reflecting that patients learnt the contingencies appropriately through brain compensatory mechanisms. But, maybe these behavioral measures are not sufficient to capture the full spectrum of behavior between groups. We could not add the spatial trajectories or time spent in each zone to the behavioral analysis due to several factors of the task, 1) participants were forced to collect one flower at a time; 2) the flowers appeared randomly throughout the environment therefore they could be close or far from each other; 3) age, gender, and video game experience might have affected the task proficiency in general. Future studies should consider using other sorts of behavioral measures (e.g. eye-tracking), increasing the stakes of the shock contingency (e.g. correct expectancy ratings diminish the risk of shock), or increasing the sample size to increase power and therefore flush out the exploratory findings. Another component that might have affected the behavioral results is the inclusion of patients with social anxiety disorder (SAD), as the task did not include any social component. The study aimed to capture differences between HC and patients with any anxiety disorder. And while the study only recruited a minimal number of patients with SAD (7), their inclusion might have skewed the results more favorably to appropriate learning behavior. Future studies should focus on patients with generalized anxiety disorder to test this.”*. Nonetheless, if the reviewer still wants us to run and add these analyses to the manuscript, we would be happy to oblige.

To enrich our behavioral results, we added an analysis of the state-trait anxiety inventory (STAI; Spielberger, Gorsuch, Lushene, Vagg, & Jacobs, 1983), focusing on the state anxiety scale, which we collected before and after the task. The state anxiety scale has 20 self-reported items assessing anxiety experienced at the present moment. Therefore, we were able to explore changes in state anxiety from before to after completion of the VR task. We found that ANX participants had higher state anxiety scores compared with HC participants at post-task, while at pre-task, both groups showed similar state-anxiety levels.

We have added this to the Results section as follows (line 138): *“State-Trait Anxiety Inventory (STAI). Before and after the task participants filled out the state anxiety section of the STAI. A 2x2 ANOVA (group x time) analysis of the state anxiety (pre-task, post-task) revealed a significant group effect ($F(1, 22)=28.07, p=0.000$). No other significant main effects or interactions were observed (F 's<2, p 's>0.05). A t -test revealed that ANX had higher state anxiety than HC only at post-task time point (pre-task, $t(49)=2.56, p=0.116$; post-task, $t(49)=6.51, p=0.014$).”*

- Spielberger, C. D., Gorsuch, R. L., Lushene, R., Vagg, P. R., & Jacobs, G. A. (1983). Manual for the State-Trait Anxiety Inventory. Palo Alto, CA: Consulting Psychologists Press.

Most importantly, while we don't show a strong behavioral difference between groups, this study shows differences in brain activity which are not confounded by behavioral differences. And, in this time of replicability issues, we note that we were able to replicate our initial HC findings. Furthermore, the differences between groups are in the brain areas we identified in our initial study for learning to discriminate between zones and for tracking threat. This suggest that if there are indeed no behavioral differences between groups, the ANX group is engaging the same brain areas (albeit differently) as compensatory mechanisms to rescue the appropriate behavior. We have added more about the current HC results and the previous paper on the same task in the discussion as follows (line 444): *“Several points are noteworthy when considering the current findings in light of the previous study in healthy controls using the same task. In both task we have identified the same network of activation in HC’s when approaching and interacting with flowers. We replicated the two neural networks to support the formation of cognitive maps and its relevant behavior: 1) The discrimination learning network (DLN; vmPFC, PCC, anterior hippocampus) activation increased when HC’s learn to discriminate threatening and safe zones within the environment; 2) the salience network (SN; dmPFC/dACC, insula, PAG) activation increased when participants approached and interacted with flowers in the danger zone. There is one main difference between this and the past study in the SN, in the current study the peak activation was more in the dmPFC while in the previous study the peak was more in the dACC. However, the brain activation cluster of both studies often overlap between the two areas (dmPFC/dACC). Furthermore, while some brain activation differences found in the current study are using a priori hypothesis SVC based on the past study, all the group brain activation differences (HC, ANX) of the current study are in the areas identified in the previous study (DLN, SN).”*

Comment 6

Finally, there is a bit of vague speculation and reverse-inferencing in the discussion. For example, the conclusion that " Our findings suggests that patients’ threat-related attention may cause deficits in emotion regulation. The current results suggest that disconnected circuitry in brain areas essential for emotional regulation might lead to disrupted emotional output while exploring environments and learning to discriminate cues within these environments." appears rather unsupported. The paper does not define what "emotion regulation" is on a descriptive level, let alone come forward with a mechanistic model of it. How can we conclude deficits in this unspecified process from the observation of more or less oxygen in some brain regions?

Response 6

We have rewritten several parts of the discussion and the conclusions to improve the clarity of our thoughts. While we find increased anxiety (STAI) pre-post task, we do not find significant group interaction in the skin conductance. This might reflect that patients are learning the contingencies of the task because we are forcing them to experience the task. Particularly, since they cannot avoid the flower nor they have multiple choices. This supports studies that show that patients with anxiety don't necessarily have learning difficulties, but rather use more inflexible learning strategies (outside of the hippocampus) which requires more time to learn and it is prone to overgeneralization, particularly in novel situations. We have reworked several sections of the manuscript to better reflect these thoughts. For example:

We have added the following to the second paragraph of the discussion (line 435): *“These results suggest that patients with pathological anxiety recruit compensatory learning strategies that do*

not involve the anterior hippocampus or vmPFC, which might explain the increased activation in the salience network (e.g., dmPFC, insula, PAG). In this task the participants were forced to collect the flowers and therefore got the opportunity to know the contingencies of the zones. Real life scenarios don't often offer that possibility, particularly if the patient prefers to avoid it, creating higher uncertainty and risk. We believe that reintegrating the anterior hippocampus and vmPFC into the circuitry could rescue appropriate learning strategies to discriminate between safety and danger, particularly when information and experience is limited."

We have added the following to the Threat Learning and Discrimination section of the discussion (line 488): *"The anterior hippocampus is essential for the integration of spatial information in mediating anxiety-like behaviors. As seen in the current study with healthy controls, previous findings show that the hippocampus is involved in learning and discrimination of safe and danger zones, with increased activation with task experience. Therefore, decreased engagement of the anterior hippocampus might reflect a compensatory mechanism, whereby patients with pathological anxiety use alternative learning strategies that do not involve context-specific cues for learning the environment contingencies. These results are in line with research showing hippocampal and medial prefrontal cortex abnormalities in patients suffering from pathological anxiety who report an exaggerated response to threats in contexts predicting safety. Reduced activation of the anterior hippocampus, particularly during potential or perceived threat, could impair emotional regulation abilities in novel contexts or when shifting contexts."*

Finally, we have rewritten the last (conclusion) paragraph of the discussion to the following (line 508): *"In conclusion, patients with pathological anxiety show lower activation in the anterior hippocampus and vmPFC/OFC and lower connectivity to emotion expression brain areas (dmPFC, amygdala, and insula). In this task, participants were forced to collect a flower, there was no avoidance or multiple options available. Unlike real-world scenarios where we might opt out of experiencing something, here participants could not, forcing on them the opportunity to learn about the flowers. Therefore, while patients seemed to learn to discriminate between zones, lack of hippocampus and mPFC (particularly within the danger zone) could explain why patients with anxiety disorders often show heightened emotional responses when they are in a novel environment or situation and are not able to accurately identify potential threat. This possibility is further supported by the observed higher dmPFC activation, and other emotion valuation and expression brain areas, during the task. The current results suggest that a disconnected circuitry in brain areas essential for memory and context processing and emotional regulation might lead to disrupted emotional output without extensive exposure. That is, while exploring unfamiliar environments and learning about cues within these environments, patients with anxiety disorder might find it difficult to regulate their anxiety and opt out of the experience. Finding novel psychological interventions or training to reintegrate the vmPFC and hippocampus into the learning and discrimination circuitry could help patients with pathological anxiety regulate their emotional responses while navigating novel environments. This might allow patients more exposure time to learn whether they are safe or not."*

Reviewers' comments:

Reviewer #1 (Remarks to the Author):

I have carefully considered the authors' responses to my comments. A number of aspects about the analysis pipeline have now been clarified. I could not find a manuscript with tracked changes as standard, which I would have appreciated.

Initial comment 1

My main comment was about qualitative behavioral signatures, since brain differences between groups are not translating into behavioral differences between groups, a point that I note has also been raised by Reviewer 2.

- The authors' response about the STAI before and after the task is clear, but it does not address task behaviour per se.
- Similarly, the brain importance paragraph (and the fact that there are brain differences) is interesting in and of itself, but it does not address the behavioural signatures.
- Then the last explanation of the authors is that comparing behaviour between groups would not be meaningful or even misleading because there may be differences in age, gender and video game experience between groups, which would render comparison of response times meaningless. However, the authors report that they found no difference in demographics between groups, including no differences in age and gender between groups. Similarly, the argument that flowers appeared at random locations should average out since the randomization was done properly, and/or the analyses could control for distance between flower(t) and flower(t+1).
For these reasons, unless I am mistaken, it sounds like at least some (not all) of the behavioural analyses I suggested could be easily performed and provide meaningful information (namely, Do participants spend more time in either zone? Do participants with anxiety spend more time in the safe zone as compared to the control group? Could the authors find basic behavioral signatures of more avoidance in participants with anxiety, for instance exploring for longer (ie. spending more time) before picking up a flower?)

Initial comment 10

Regarding brain activity and the arguments for ruling out an interpretation in terms of a valuation network, I think the material brought up by the authors is interesting and would enrich the discussion, particularly as valuation in this network has been the object of a vast literature - but I leave it up to the authors to include it in their paper.

I also note that besides valuation, that an increase in vmPFC/PCC/aHippocampus over time while participants learn to discriminate between zones could be associated with increased confidence about mastering the task/environment (e.g. De Martino et al., 2013 Nat neuro).

Reviewer #2 (Remarks to the Author):

The authors have revised several important aspects of the manuscript which has improved its readability and addressed many of my concerns. However, the reporting is still not always clear, consistent and correct. Therefore, I have several outstanding points that would need to be addressed:

Major

(1) To provide evidence for anxiety changes due to the task, the manuscript now reports that ANX

participants had a higher STAI score compared to HC after task completion. However, the Anova with the pre-task and post-task time point does not show a timepoint x group interaction. In other words, this effect is not an anxiety effect induced by the task, but one driven by both time points (even if pre-task it is not quite significant, there is no significant difference between the groups pre- vs post-task). That just means the ANX participants had a larger STAI score in general, it does not provide evidence that the task induced any changes in anxiety. This is not a problem per se but please ensure that this is how this result is presented and interpreted.

(2) Similarly, the statistics on Figure 1C/F showing SCL changes were only present in HC is not based on a direct test between groups nor on a Bayesian test that provides evidence for the null in ANX (in fact ANX are at $p=0.06$ and HC at $p<0.01$ and the full zone x block x group interaction shows they are not significantly different from each other). Similarly, for the SCR, there are no precise stats given except that HC is $p<0.05$ and ANX $p>0.05$, but again the groups are not different from each other given the lack of an interaction effect. Please state this more explicitly to avoid misinterpretation. In general, as per my comment in the previous round of revisions, please report precise p-values throughout – this has still not been done.

(3) I have several problems with the presentation of the neural results in Figures 3-5, in particular the related bargraphs:

- a. The bargraphs in Figures 3-5 do not report independent tests but serve to illustrate the effects shown in the corresponding brain maps. They should therefore not have any stars to indicate significance as any further test would be circular.
- b. The bargraphs do not always match the summary given in the text. For example, with respect to Fig 3A the authors state “[...] we observed that the ANX group, compared to the HC group, demonstrated a greater increase in activation in these areas from early to late blocks of the safe zone (late > early; safe > danger)” but this is only true for Hippocampus, not for OFC/sACC.
- c. Relatedly, bargraphs are not reported consistently across brain regions. For example, in Figure 3B, the contrast is danger>safe in ANX vs HC, but the bargraph is only showing danger, not matching with the text which states: “That is, these areas were more responsive in the ANX group when approaching flowers in the danger zone than in the safe zone.” This statement would require the blue bars for the safe zone to be shown as well. Note that the insula activation is entirely missing from both the whole-brain and bar graph figure. It is therefore not clear why the PPI is performed based on the dmPFC and not the insula given both are identified within the same contrast.
- d. The bargraphs in Figure 4 and 5 are similarly inconsistent. The contrast of interest in Figure 4 is danger>safe in ANX>HC but the bar graphs for one region (dmPFC) are shown for early and late blocks but only danger, for the other region (amygdala), only the danger across blocks is shown. To match the contrast conducted and illustrate what was driving the effects in the whole-brain maps, activations in both regions should show the same bars below, and those should be safe and danger for ANX and HC (but not split by early/late). The same applies to Figure 4B; and again, in Figure 5A, PCC is shown for all 8 bars, but OFC and hippocampus just show four bars – please correct these inconsistencies.
- e. Please also add individual data points to bar graphs throughout the manuscript.

Minor

(4) The authors state “When extracting the time series”, but the figures show parameter estimates, not time series – is that what the authors mean? Please correct throughout.

(5) The authors explain how video game experience, age, gender and other factors such as the random location of the flower could influence the trajectories to flowers. However, assuming these effects are similar in both groups and average out, these are not valid reasons for not conducting these analyses. I would suggest stating that such analyses could be conducted in future work, rather than giving reasons for not doing them which do not seem valid.

(6) There are still many typos. (e.g. “This trend level effect in learning between zones for ANX was

due a significantly" or "neural signatures of generalisation/discrimination in in patients" for "for consistency previous the previous study", "consited" should be consisted, FEW should be FWE and others)

(7) In the next round and in general in the future, please submit a manuscript with changes highlighted. This greatly reduces the workload of reviewers - thank you!

Reviewer #3 (Remarks to the Author):

The authors have addressed my concerns.

Reviewer #1

We are grateful to Reviewer #1 for all the valuable feedback and suggestions. It seems our last revision document with track changes was not successfully uploaded, we apologize for the inconvenience and have ensured that it is included this time.

Comment 1 to Initial comment 1

My main comment was about qualitative behavioral signatures, since brain differences between groups are not translating into behavioral differences between groups, a point that I note has also been raised by Reviewer 2.

- The authors' response about the STAI before and after the task is clear, but it does not address task behaviour per se.
- Similarly, the brain importance paragraph (and the fact that there are brain differences) is interesting in and of itself, but it does not address the behavioural signatures.
- Then the last explanation of the authors is that comparing behaviour between groups would not be meaningful or even misleading because there may be differences in age, gender and video game experience between groups, which would render comparison of response times meaningless. However, the authors report that they found no difference in demographics between groups, including no differences in age and gender between groups. Similarly, the argument that flowers appeared at random locations should average out since the randomization was done properly, and/or the analyses could control for distance between flower(t) and flower(t+1).

For these reasons, unless I am mistaken, it sounds like at least some (not all) of the behavioural analyses I suggested could be easily performed and provide meaningful information (namely, Do participants spend more time in either zone? Do participants with anxiety spend more time in the safe zone as compared to the control group? Could the authors find basic behavioral signatures of more avoidance in participants with anxiety, for instance exploring for longer (ie. spending more time) before picking up a flower?)

Response 1

We agree with the reviewer. Accordingly, as recommended by the reviewer, we have conducted the above-mentioned analyses to explore avoidance and anxiety through navigation differences. Specifically, we now analyze: 1) the overall time spent performing the task; 2) navigation time (averaged across both zones) over time; and 3) navigation time by zone over time. Results of all analyses only show a significant effect of time, where over time participants get better at navigating the environment regardless of zone or group. These results were performed twice, once for all navigation time and once only for active navigation towards the flowers. Results of these analyses appear in full in the Supplementary Material of the manuscript. Yet, if deemed necessary we are more than willing to move them to the main manuscript.

1. We have added to the supplementary materials these results which reads as follows: *“Environmental navigation. Overall approach navigation: Overall approach navigation was measured as participants navigated the environment before collecting a flower. A 2x2x4 ANOVA*

(zone by group by block) showed a main effect of time with navigation time decreasing from block 1 to block 4 ($F(3,47)=23.06, p=0.00$). No other significant main effects or interactions were observed ($F's < 2, p's > 0.05$).

Active approach navigation: Active approach navigation was measured as participants actively navigated towards a flower. A $2 \times 2 \times 4$ ANOVA (zone by group by block) analysis showed a main effect of time with navigation time decreasing from block 1 to block 4 ($F(3,47)=13.66, p=0.00$). No other significant main effects or interactions were observed ($F's < 2, p's > 0.05$).

Overall task duration: Overall task duration was measured as the total time participants took to complete the task. An independent samples t -test showed no group differences on completing the task ($t(49)=0.42, p=0.51$).

2. We now also address these results in the Discussion of the revised manuscript as follows (line 425): *“Surprisingly, inconsistent with our hypotheses, patients with pathological anxiety showed no impairment in discriminating between threat and safety within the environment based on their skin conductance, subjective reports, and navigation time (see supplementary material).”*

3. We agree with the reviewer that while there are no demographic differences between groups that would affect the analysis (and that the randomization of flowers was performed in a way to control for game experience differences). However, as we did not assess gaming experience across participants, we cannot ensure that this variable did not affect the results (or interact differently with other demographics). Therefore, we still believe that there might have been confounds in the results due to gaming experience (and its interaction with gender and age), which could affect the way participants learn and engage with the task. To make this issue clearer in the revised manuscript, we have edited the corresponding limitation in the Discussion section of the manuscript as follows (line 588): *“While we did not find spatial trajectories or time spent in each zone differences between groups this could be due to several factors of the task, 1) participants were forced to collect one flower at a time; 2) the flowers appeared randomly throughout the environment therefore they could be close or far from each other; 3) the interaction of age, gender, and video game experience might have affected the task proficiency in general. And, we did not assess video game experience of each participant to fully flush out these results. Future studies should consider using other sorts of behavioral measures (e.g. eye-tracking), increasing the stakes of the shock contingency (e.g. correct expectancy ratings diminish the risk of shock), gaining more insight into participants' video game experience, or increasing the sample size to increase power and therefore flush out more the exploratory findings.”*

Comment 2 to Initial comment 10

Regarding brain activity and the arguments for ruling out an interpretation in terms of a valuation network, I think the material brought up by the authors is interesting and would enrich the discussion, particularly as valuation in this network has been the object of a vast literature - but I leave it up to the authors to include it in their paper.

I also note that besides valuation, that an increase in vmPFC/PCC/aHippocampus over time while participants learn to discriminate between zones could be associated with increased confidence about mastering the task/environment (e.g. De Martino et al., 2013 Nat neuro).

Response 2

We appreciate the Reviewer's comments about valuation and task confidence. As the reviewer found this to be of interest possibly enriching our Discussion, we have added these points as follows (line 510): *“As the vmPFC and PCC are considered part of the classic valuation network (FitzGerald et al., 2009; Kable and Glimcher 2007; Rangel & Hare 2010) - the brain network implicated in assessing the subjective valuation of stimuli - our findings could be explained also in terms of the safe zone being more valuable than the danger zone. However, based on a previous study in healthy adults and this study's healthy control group we see that activation in the vmPFC and PCC, along with the anterior hippocampus, increase over time as participants learn to discriminate between zones. That is, the highest activation in these areas is evident when they have learned the contingencies and rules of the environment, regardless of zone, as there is no zone difference in these areas. A zone difference surfaces only when we look at patients with anxiety disorder, who show an increase in activation over time in the vmPFC, PCC, and anterior hippocampus for flowers in the safe zone, while a decrease over time for flowers in the danger zone. This interaction in turn could be interpreted as suggesting that patients with anxiety disorders have a higher confidence of value for the safe zone compared to the danger zone. Indeed, activation in the vmPFC and PCC areas has been associated with increased confidence of decision value (De Martino et al., 2013), which could be associated with higher discrimination learning (Rolls et al., 2010). However, we do see that salient network areas activation (insula, dmPFC, and PAG) increase over time, particularly for the danger zone. Therefore, we believe that this system takes over in the danger zone, overriding or dampening the activation of the vmPFC and PCC in patients with anxiety disorders. If this vmPFC and PCC network was indeed active during the task for valuation confidence we would expect that healthy participants would display this pattern of activation as well, with higher activation in the safe compared to the danger zone. And indeed, we show that it is not the case in healthy controls.”*

FitzGerald THB, Seymour B, Dolan RJ. The Role of Human Orbitofrontal Cortex in Value Comparison for Incommensurable Objects. *Journal of Neuroscience*. 2009;29:8388–8395.

Kable JW, Glimcher PW. The neural correlates of subjective value during intertemporal choice. *Nature Neuroscience*. 2007;10:1625–1633.

Rangel A, Hare T. Neural computations associated with goal-directed choice. *Current Opinion in Neurobiology*. 2010;20:262–270.

De Martino B, Fleming, SM, Garret N, Dolan R. Confidence in value-based choice. *Nat Neurosci*. 2013;16(1):105-110.

Rolls ET, Grabenhorst F, Deco G. Choice, difficulty, and confidence in the brain. *Neuroimage*. 2010;53:694–706.

Reviewer #2

Major

Comment 1

To provide evidence for anxiety changes due to the task, the manuscript now reports that ANX participants had a higher STAI score compared to HC after task completion. However, the Anova with the pre-task and post-task time point does not show a timepoint x group interaction. In other words, this effect is not an anxiety effect induced by the task, but one driven by both time points (even if pre-task it is not quite significant, there is no significant difference between the groups pre- vs post-task). That just means the ANX participants had a larger STAI score in general, it does not provide evidence that the task induced any changes in anxiety. This is not a problem per se but please ensure that this is how this result is presented and interpreted.

Response 1

We agree with the reviewer.

1. To ensure clarity we have edited the results as follows (line 139): “A 2x2 ANOVA (group by time) analysis of the state anxiety (pre-task, post-task) revealed a significant group effect ($F(1, 22)=28.07, p=0.00$) showing an overall higher state anxiety in the ANX compared to the HC. No other significant main effects or interactions were observed ($F's < 2, p's > 0.05$). A post-hoc t-test revealed that ANX had higher state anxiety than HC only at post-task (pre-task, $t(49)=2.56, p=0.12$; post-task, $t(49)=6.51, p=0.014$).”

2. In the discussion we have further clarified this point as follows (line 578): “It is important to note that while ANX displayed a higher state anxiety than HC, particularly at the end of the task, it does not provide evidence that the task induced any changes in anxiety.”

Comment 2

Similarly, the statistics on Figure 1C/F showing SCL changes were only present in HC is not based on a direct test between groups nor on a Bayesian test that provides evidence for the null in ANX (in fact ANX are at $p=0.06$ and HC at $p<0.01$ and the full zone x block x group interaction shows they are not significantly different from each other). Similarly, for the SCR, there are no precise stats given except that HC is $p<0.05$ and ANX $p>0.05$, but again the groups are not different from each other given the lack of an interaction effect. Please state this more explicitly to avoid misinterpretation. In general, as per my comment in the previous round of revisions, please report precise p-values throughout – this has still not been done.

Response 2

1. We have edited the results to include precise p-values for all analyses. We also better clarify that these analyses were done at an exploratory level.

- For SCL it reads as follows (line 148): “Based on the first hypothesis, that ANX would evidence overgeneralization, 12 within-group main effects of zone were explored and indicated that only HCs had significantly higher SCL towards dangerous flowers as compared to safe flowers (HC, $t(27)=15.55, p=0.00$; ANX, $t(22)=3.97, p=0.06$; Fig. 1 C and F).”

- For SCR it reads as follows (line 158): “*Within-group main effects of zone were explored, showing that only HC’s had significantly higher SCR towards dangerous flowers as compared to safe flowers (HC, $t(27)=5.85, p=0.02$; ANX, $t(22)=0.97, p=0.33$; Fig. 1 D and G).*”

2. Furthermore, to avoid misinterpretation of these exploratory analysis we now better clarify this point in the Discussion as follows (line 425): “*Surprisingly, inconsistent with our hypotheses, patients with pathological anxiety showed no impairment in discriminating between threat and safety within the environment based on their skin conductance and subjective reports.*”, and (line 581) “*Exploratory analysis did suggest that only HC were showing skin conductance (SCL, SCR) differences between the zones while patients did not. However, as an exploratory analysis this should be taken with caution as our main analysis revealed no interaction effects, and particularly as both patients and HC participants learned the task’s contingencies (expectancy ratings).*”

Comment 3

I have several problems with the presentation of the neural results in Figures 3-5, in particular the related bargraphs:

- a. The bargraphs in Figures 3-5 do not report independent tests but serve to illustrate the effects shown in the corresponding brain maps. They should therefore not have any stars to indicate significance as any further test would be circular.
- b. The bargraphs do not always match the summary given in the text. For example, with respect to Fig 3A the authors state “[...] we observed that the ANX group, compared to the HC group, demonstrated a greater increase in activation in these areas from early to late blocks of the safe zone (late > early; safe > danger)” but this is only true for Hippocampus, not for OFC/sACC.
- c. Relatedly, bargraphs are not reported consistently across brain regions. For example, in Figure 3B, the contrast is danger>safe in ANX vs HC, but the bargraph is only showing danger, not matching with the text which states: “That is, these areas were more responsive in the ANX group when approaching flowers in the danger zone than in the safe zone.” This statement would require the blue bars for the safe zone to be shown as well. Note that the insula activation is entirely missing from both the whole-brain and bar graph figure. It is therefore not clear why the PPI is performed based on the dmPFC and not the insula given both are identified within the same contrast.
- d. The bargraphs in Figure 4 and 5 are similarly inconsistent. The contrast of interest in Figure 4 is danger>safe in ANX>HC but the bar graphs for one region (dmPFC) are shown for early and late blocks but only danger, for the other region (amygdala), only the danger across blocks is shown. To match the contrast conducted and illustrate what was driving the effects in the whole-brain maps, activations in both regions should show the same bars below, and those should be safe and danger for ANX and HC (but not split by early/late). The same applies to Figure 4B; and again, in Figure 5A, PCC is shown for all 8 bars, but OFC and hippocampus just show four bars – please correct these inconsistencies.
- e. Please also add individual data points to bar graphs throughout the manuscript.

Response 3

We apologize for these inconsistencies and are grateful for the reviewers' comments regarding the figures. We believe that addressing these comments improved the Figures' quality and enhanced their understanding.

- a) We agree with the reviewer and have removed the stars from the bar graphs.
- b) We have reviewed the text and the figures and clarified some of the findings.
 1. For example, Fig 3B text now reads as follows (line 244): *“When approaching flowers in the danger zone, the ANX group (ANX > HC) showed greater activation of the bilateral insula ($p < 0.05$ FWE) and dmPFC ($p < 0.05$ FWE SVC; Fig. 3B) compared to the HC group. That is, these areas were more responsive in the ANX group, compared to HC, when approaching flowers in the danger zone.”* However, we did not change the specific example of the reviewer as we find that it does reflect the finding. That is, Fig 3A shows in both Hippocampus and OFC/sACC that activity in the ANX increases in the safe zone (blue bars) but decreases in the danger zone (red bars) from early (dark gray bars) to late (light gray bars) block (for the HC both blue and red bars increase with block).
 2. Fig 4 text now reads as follows (line 310): *“When held stationary after picking a flower located in a zone of the environment associated with danger, the ANX group compared to HC (ANX > HC) showed greater activation in dmPFC, dACC, bilateral insula, caudate, thalamus, amygdala, and midbrain areas, including the periaqueductal gray (PAG; $p < 0.05$ FWE SVC; Fig. 4A). That is, these areas were more responsive in the ANX, compared to HC, when approaching flowers in the danger zone. For the HC group, compared to ANX (HC > ANX), flowers located in a zone of the environment associated with safety generated greater activation in the PCC ($p < 0.05$ FWE), vmPFC, OFC/sACC, and anterior hippocampus ($p < 0.05$ FWE SVC; Fig. 4B). That is, these areas were more responsive in the HC, compared to ANX, when approaching flowers in the safe zone.”* and (line 321): *“A group contrast (ANX > HC) of stationary periods (irrespective of danger or safety) showed increased dmPFC activation from early to late block (late > early) in ANX compared with HC ($p < 0.05$ FWE SVC; Fig 4A).”*
 3. Fig 5 text now reads as follows (line 384): *“A task by block by group interaction of approaching cues identified the PCC ($p < 0.05$ FWE), OFC/sACC, and anterior hippocampus ($p < 0.05$ FWE SVC; see Fig. 5A). When approaching objects (objects > flowers; early > late) ANX showed increased PCC activation, while HC showed decreased activation, from early to late blocks. On the other hand, when approaching flowers (flowers > objects; early > late), ANX showed decreased PFC/sACC and anterior hippocampus, while HC showed increased activation, from early to late blocks.*

To further understand brain activation differences as participants learnt the contingencies of the tasks we examined the block contrasts (early, late) between groups. This analysis takes the average of the flower and object tasks regressor and looks at the block effects. During the second half of the experiment (late > early),

the ANX (ANX > HC) showed increased dmPFC activation from the early to the late blocks irrespective of the task ($p < 0.05$ FWE SVC; Fig. 5B)."

- c) We agree with the reviewer that bar graphs are not reported consistently across brain regions. As pointed by the reviewer these figures and graphs are intended to be more of the illustrative kind. We wanted to showcase an array of brain areas and felt that it would become redundant to showcase the same areas of the brain which showed similar patterns of activity. Also showcasing all the brain areas and bar graphs would overwhelm the figures with all the brain areas and bar graphs findings. That is why we decided to spread the findings (diversify the findings) across the figures. As we clarified the findings are specific to zones (see point b above), we hope it clarifies the choice of bars. For example, in the dmPFC during approach (as well as for the amygdala, OFC, and hippocampus during stationary periods) there was a significant group difference at the zone level. Taking for example the dmPFC, while there were no differences over time, overall ANX had a higher activity in this dmPFC region than HC. We sought to highlight this difference in the figure. Adding the time split, or even the safe zone bars, would make the difference between groups (of the danger zone) less clear, as the scale and information of the other activity (early, late, and safe zone) would have occluded it. For example, the bar graphs would have looked as the figure below. We thought it would make a clearer and highlighted point to keep the figure simply to the zone difference between groups when appropriate to highlight the finding.

Figure legend: safe blocks (S) are surrounded by blue line; danger blocks (D) are surrounded by red line; dark grey bars represent early block (ER); light grey bars represent late block (LT); safe early block (S ER); safe late block (S LT); danger early block (D ER); danger late block (D LT); health control (HC) are represented in the left four bars; patients with anxiety (ANX) are represented in the right four bars.

- d) For figure 4A, we have switched the order of the illustrations to represent the order in which they are described in the text. That is, since we describe the danger stationary period contrast of zone and then of block, we have switched the amygdala and dmPFC illustration to follow the text order. For Figure 5A, we have added the missing interaction bar graph. For Fig 5B, we realized that here there should not be any color in the bars since this Figure describes the average block effect of the two tasks.
- e) We have opted out of using individual data points over the bar graphs as the figures are already charged with information and it would be harder to maintain the quality and visibility (in size) of the figures.

Minor

Comment 4

The authors state “When extracting the time series”, but the figures show parameter estimates, no time series – is that what the authors mean? Please correct throughout.

Response 4

The reviewer is correct, and we have revised this throughout the manuscript.

Comment 5

The authors explain how video game experience, age, gender and other factors such as the random location of the flower could influence the trajectories to flowers. However, assuming these effects are similar in both groups and average out, these are not valid reasons for not conducting these analyses. I would suggest stating that such analyses could be conducted in future work, rather than giving reasons for not doing them which do not seem valid.

Response 5

We agree with the reviewer. Therefore, we have conducted the above-mentioned analyses to explore avoidance and anxiety through navigation differences. Specifically, we now analyze: 1) the overall time spent performing the task; 2) navigation time (averaged across both zones) over time; and 3) navigation time by zone over time. Results of all analyses only show a significant effect of time, where over time participants get better at navigating the environment regardless of zone or group. These results were performed twice, once for all navigation time and once only for active navigation towards the flowers. Results of these analyses appear in full in the Supplementary Material of the manuscript. Yet, if deemed necessary we are more than willing to move them to the main manuscript.

1. We have added to the supplementary materials these results which reads as follows: *“Environmental navigation. Overall approach navigation: Overall approach navigation was measured as participants navigated the environment before collecting a flower. A 2x2x4 ANOVA (zone by group by block) showed a main effect of time with navigation time decreasing from block 1 to block 4 ($F(3,47)=23.06, p=0.00$). No other significant main effects or interactions were observed ($F's < 2, p's > 0.05$).*

Active approach navigation: Active approach navigation was measured as participants actively navigated towards a flower. A 2x2x4 ANOVA (zone by group by block) analysis showed a main effect of time with navigation time decreasing from block 1 to block 4 ($F(3,47)=13.66, p=0.00$). No other significant main effects or interactions were observed ($F's < 2, p's > 0.05$).

Overall task duration: Overall task duration was measured as the total time participants took to complete the task. An independent samples t-test showed no group differences on completing the task ($t(49)=0.42, p=0.51$).”

2. We now also address these results in the Discussion of the revised manuscript as follows (line 425): *“Surprisingly, inconsistent with our hypotheses, patients with pathological anxiety showed*

no impairment in discriminating between threat and safety within the environment based on their skin conductance, subjective reports, and navigation time (see supplementary material)."

3. We agree with the reviewer that while there are no demographic differences between groups that would affect the analysis (and that the randomization of flowers was performed in a way to control for game experience differences). However, as we did not assess gaming experience across participants, we cannot ensure that this variable did not affect the results (or interact differently with other demographics). Therefore, we still believe that there might have been confounds in the results due to gaming experience (and its interaction with gender and age), which could affect the way participants learn and engage with the task. To make this issue clearer in the revised manuscript, we have edited the corresponding limitation in the Discussion section of the manuscript as follows (line 588): *"While we did not find spatial trajectories or time spent in each zone differences between groups this could be due to several factors of the task, 1) participants were forced to collect one flower at a time; 2) the flowers appeared randomly throughout the environment therefore they could be close or far from each other; 3) the interaction of age, gender, and video game experience might have affected the task proficiency in general. And, we did not assess video game experience of each participant to fully flush out these results. Future studies should consider using other sorts of behavioral measures (e.g. eye-tracking), increasing the stakes of the shock contingency (e.g. correct expectancy ratings diminish the risk of shock), gaining more insight into participants' video game experience, or increasing the sample size to increase power and therefore flush out more the exploratory findings."*

Comment 6

There are still many typos. (e.g. "This trend level effect in learning between zones for ANX was due a significantly" or "neural signatures of generalisation/discrimination in in patients" for "for consistency previous the previous study", "consited" should be consisted, FEW should be FWE and others)

Response 6

We thank the Reviewer for noticing these oversights on our part, which we apologize for. We have corrected these and others throughout the manuscript.

Comment 7

In the next round and in general in the future, please submit a manuscript with changes highlighted. This greatly reduces the workload of reviewers - thank you!

Response 7

It seems our last revision document with track changes was not successfully uploaded, we apologize for the inconvenience and have ensured that it is included this time.

Reviewers' comments:

Reviewer #1 (Remarks to the Author):

I have now reviewed the authors' navigation time analyzes, and found it useful in complementing the brain analyzes.

Regarding the interpretation of an increase vmPFC/PCC/aHippocampus activity over time, the authors argue in favour of a valuation system rather than an increase in confidence. It seems to me that the authors' arguments brought forward support a confidence interpretation instead, namely:

"If this vmPFC and PCC network was indeed active during the task for valuation confidence we would expect that healthy participants would display this pattern of activation as well, with higher activation in the safe compared to the danger zone. And indeed, we show that it is not the case in healthy controls."

Participants could be equally confident in both zones as they have figured out and identified the zones (confidence interpretation). This is also compatible with: "the highest activation in these areas is evident when they have learned the contingencies and rules of the environment, regardless of zone, as there is no zone difference in these areas.". Instead, if the authors had observed higher activity in the safe compared to the danger zone in control participants, that would be an argument in favour of a valuation interpretation.

The authors take an increased activity in dmPFC and insula as part a "salient network", although these areas could equally be described as a "negative valuation" and "negative confidence" networks.

Because this is a discussion point, I leave it to the authors how committed they are to either interpretation.

Reviewer #2 (Remarks to the Author):

Thanks for your responses to my previous comments. Some points have still not been fully addressed.

Previous comment 2: I stated this in the two previous rounds already but reporting precise p-values is essential and still not done. A $p=0.00$ is not a precise value and appears eleven times in the manuscript. Even though the authors state "We have edited the results to include precise p-values for all analyses.", this is unfortunately not the case. Most softwares allow changing their default settings to see p-values at a greater decimal precision, so that precise values can be reported (e.g. $p=5e-10$ for 0.0000000005).

I also asked the authors to explicitly mention the lack of a group difference in the results when stating that the SCR and SCL results were 'only significant in HC' (lines 148/158). Please include such a statement e.g. "However, a direct comparison between ANX and HC groups was not significant, and the ANX group showed similar effects at trend-wise levels." Statistical reporting needs to be transparent and accurate.

The following sentence in the discussion is still incorrect: "Exploratory analysis did suggest that only HC were showing skin conductance (SCL, SCR) differences between the zones while patients did not." The part that states 'patients did not' is not consistent with the results ($p=0.06$ for ANX for SCL). If the authors want to argue that there is no effect for ANX for SCL, this requires a Bayesian Anova/t-test (e.g. performed in JASP). You can either tone down this difference between groups or perform

correct testing in support of the null hypothesis in ANX using Bayesian statistics.

Previous comment 3:

The reviewers decided not to show individual data points (or at least the data distribution as whisker or similar plots). I think this is a shame because it has now become common practice in the field, but of course I leave this up to them and the journal to discuss/decide.

Figure 4A: I don't think swapping the left and right figures for amygdala and dmPFC in Fig4A has made any difference. It is still unclear why the amygdala is shown with two bars and dmPFC with four (i.e., split by early/late). If this is because later in line 328 the authors refer to block effects in dmPFC, this is even more confusing because the dmPFC result is under the figure title "Danger stationary periods" but that paragraph from line 323 explicitly mentions "This analysis takes the average of the safe and danger regressors", so should be under a different figure sub-heading, while dmPFC in Figure 4A should be shown in the same way as the amygdala if that is the point the authors are trying to make in Figure 4A. At the very least, I think the dmPFC bars should not have red colour around them if they depict effects across danger and safe zones. Or, if they are indeed just for the danger zone, then they are inconsistent with the paragraph in line 328, and should show the safe zone in blue bars in addition to the red ones already shown.

Figure 5: "On the other hand, when approaching flowers (flowers > objects; early > late), ANX showed decreased OFC/sACC and anterior hippocampus, while HC showed increased activation, from early to late blocks." This text is not consistent with what's seen in the figure for OFC/sACC (it does capture the hippocampus accurately though), so a separate sentence will be needed. The effect in OFC/sACC is not in the flower task and not in ANX, but rather a change from early to late for the object task in HC.

My previous comment on the PPI has not been addressed: "It is therefore not clear why the PPI is performed based on the dmPFC and not the insula given both are identified within the same contrast."

Please re-check spellings and grammar (e.g. line 592 that -> than, line 593 it -> this)

Editorial policy checklist

- the name and date are missing at the top.
- ticked individual data points are shown when possible – but this is not actually done

Reporting summary

- Ticked that precise p-values are given, but this is not actually the case
- Statistic type for inference is missing the cluster-defining threshold which is critical

Reviewer #1

Comment 1

Regarding the interpretation of an increase vmPFC/PCC/aHippocampus activity over time, the authors argue in favour of a valuation system rather than an increase in confidence. It seems to me that the authors' arguments brought forward support a confidence interpretation instead, namely:

“If this vmPFC and PCC network was indeed active during the task for valuation confidence we would expect that healthy participants would display this pattern of activation as well, with higher activation in the safe compared to the danger zone. And indeed, we show that it is not the case in healthy controls.”

Participants could be equally confident in both zones as they have figured out and identified the zones (confidence interpretation). This is also compatible with: “the highest activation in these areas is evident when they have learned the contingencies and rules of the environment, regardless of zone, as there is no zone difference in these areas.”. Instead, if the authors had observed higher activity in the safe compared to the danger zone in control participants, that would be an argument in favour of a valuation interpretation.

The authors take an increased activity in dmPFC and insula as part a “salient network”, although these areas could equally be described as a “negative valuation” and “negative confidence” networks.

Because this is a discussion point, I leave it to the authors how committed they are to either interpretation.

Response 1

We appreciate and agree with the reviewer's input, this is a point of view we had not considered. We have revised the Discussion as follows (line 528): *“Indeed, activation in the vmPFC and PCC areas has been associated with increased confidence of decision value, which could be associated with higher discrimination learning. Additionally, we see that salient network areas activation (insula, dmPFC, and PAG) increase over time, particularly for the danger zone, suggesting an increase in negative valuation confidence. Therefore, we believe that this system takes over in the danger zone, overriding or dampening the activation of the vmPFC and PCC in patients with anxiety disorders. On the other hand, in healthy controls this vmPFC and PCC network activity could be supporting valuation confidence as it increases equally in both zones as participants learn the environment contingencies.”*

Reviewer #2

Comment 1 to previous comment 2

Previous comment 2: I stated this in the two previous rounds already but reporting precise p-values is essential and still not done. A $p=0.00$ is not a precise value and appears eleven times in the manuscript. Even though the authors state “We have edited the results to include precise p-values for all analyses.”, this is unfortunately not the case. Most softwares allow changing their default settings to see p-values at a greater decimal precision, so that precise values can be reported (e.g. $p=5e-10$ for 0.0000000005).

Response 1

We apologize for the misunderstanding on our part. We have now corrected the p values to reflect the precise values using the suggested format $p=Xe-Y$. For example:

- For state-trait anxiety inventory (line 140): “...*significant group effect* ($F(1, 22)=28.07, p=3e-5$)...”
- For skin conductance (line 146): “...*main effect of zone with greater SCL when approaching flowers located in dangerous relative to safe zones* ($F(1,49)=16.24, p=1e-4$).”
- For shock expectancy (line 163): “...*a 2x2x4 ANOVA (zone by group by block) revealed a significant zone by block interaction* ($F(3,47)=31.07, p=1e-15$).”

Comment 2

I also asked the authors to explicitly mention the lack of a group difference in the results when stating that the SCR and SCL results were ‘only significant in HC’ (lines 148/158). Please include such a statement e.g. “However, a direct comparison between ANX and HC groups was not significant, and the ANX group showed similar effects at trend-wise levels.” Statistical reporting needs to be transparent and accurate.

Response 2

As suggested by the reviewer, we have added the following statements to the Results section to increase transparency and clarity (line 152): “*However, a direct comparison between ANX and HC groups was not significant, and the ANX group showed similar effects at trend-wise levels.*” And (line 161): “*However, a direct comparison between ANX and HC groups was not significant.*”

Comment 3

The following sentence in the discussion is still incorrect: “Exploratory analysis did suggest that only HC were showing skin conductance (SCL, SCR) differences between the zones while patients did not.” The part that states ‘patients did not’ is not consistent with the results ($p=0.06$ for ANX for SCL). If the authors want to argue that there is no effect for ANX for SCL, this requires a Bayesian Anova/t-test (e.g. performed in JASP). You can either tone down this difference between groups or perform correct testing in support of the null hypothesis in ANX using Bayesian statistics.

Response 3

The reviewer is correct that the sentence was inaccurate and misleading, and we so apologize for this. We have corrected this sentence as follows (line 587): “*Exploratory analysis did suggest that HC were showing skin conductance (SCL, SCR) differences between the zones, while in patients this was only evident at a trend-wise level for the SCL.*”

Comment 5 to previous comment 3

The reviewers decided not to show individual data points (or at least the data distribution as whisker or similar plots). I think this is a shame because it has now become common practice in the field, but of course I leave this up to them and the journal to discuss/decide.

Response 5

As suggested by the reviewer we have added the individual data points for Fig 3-5. As adding individual data points make the figures clutter and hard to see the main bar graphs, these has been added to the supplementary material to keep the main figures of the manuscript clear and easy to see.

Comment 6

Figure 4A: I don't think swapping the left and right figures for amygdala and dmPFC in Fig4A has made any difference. It is still unclear why the amygdala is shown with two bars and dmPFC with four (i.e., split by early/late). If this is because later in line 328 the authors refer to block effects in dmPFC, this is even more confusing because the dmPFC result is under the figure title “Danger stationary periods” but that paragraph from line 323 explicitly mentions “This analysis takes the average of the safe and danger regressors”, so should be under a different figure sub-heading, while dmPFC in Figure 4A should be shown in the same way as the amygdala if that is the point the authors are trying to make in Figure 4A. At the very least, I think the dmPFC bars should not have red colour around them if they depict effects across danger and safe zones. Or, if they are indeed just for the danger zone, then they are inconsistent with the paragraph in line 328, and should show the safe zone in blue bars in addition to the red ones already shown.

Response 6

We are grateful for the reviewer's careful observation and detailed comment. The reviewer is right that the dmPFC bar graph should be reflective of Danger differences in stationary periods for HC vs ANX, similar to the amygdala graph. We have now fixed the bar graph, and removed the incorrect additional citation of the figure from line 328.

Comment 7

Figure 5: “On the other hand, when approaching flowers (flowers > objects; early > late), ANX showed decreased OFC/sACC and anterior hippocampus, while HC showed increased activation, from early to late blocks.” This text is not consistent with what's seen in the figure for OFC/sACC (it does capture the hippocampus accurately though), so a separate sentence will be needed. The effect in OFC/sACC is not in the flower task and not in ANX, but rather a change from early to late for the object task in HC.

Response 7

To improve clarity, we have separated the effects into two sentences, as suggested by the reviewer. We have rewritten the results as follows (line 392): *“Additionally, while ANX showed a general OFC/sACC activation in both tasks, HC showed an increase activation for the flower task and decreased activation for the object task. On the other hand, when approaching flowers (flowers > objects; early > late), ANX showed decreased anterior hippocampus activation, while HC showed increased activation, from early to late blocks.”*

Comment 8

My previous comment on the PPI has not been addressed: “It is therefore not clear why the PPI is performed based on the dmPFC and not the insula given both are identified within the same contrast.”

Response 8

We apologize for the lack of clarity. We performed PPI analysis in the bilateral insula but it did not yield any additional results from those already presented. To better clarify this point we have added the insula as seed ROIs in the results.

- In the approach period we added (line 276): *“Given that dmPFC and insula activation was consistently higher in the ANX compared to the HC across the approaching period, particularly in the danger zone, we were interested to see how the task and the physiological state it caused in the participants interacted with the brain activity to further understand brain-behavior associations of anxiety-states (threat appraisal). For this purpose, we used a psychophysiological interaction (PPI) analyses for each participant group separately to identify dmPFC and insula (i.e., seed ROI) patterns which connectivity changed during the danger vs. safe contrast. PPI examined the brain connectivity of the significant dmPFC (MNI coordinates: 9, 26, 45) and each insula side (MNI coordinates: (R) 41, 20, 3; (L) -44, 15, 0) from the approaching flower period”*
- In the stationary period we added (line 349): *“Given that dmPFC, insula, and amygdala activation was consistently higher in the ANX compared to the HC across the stationary period, particularly in the danger zone, we were interested to see how the task and the physiological state it caused in the participants interacted with the brain activity to further understand brain-behavior associations of fear-states (threat anticipation). For this purpose, we used a PPI analyses for each participant group separately to identify dmPFC, insula, and amygdala patterns which connectivity changed during the danger vs. safe contrast. PPI examined the brain connectivity of the significant dmPFC, insula, and amygdala peak (i.e., seed ROI) from the stationary period. PPI analyses used dmPFC (MNI coordinates: 0, -8, 71), each insula side regions (MNI coordinates: (R) 41, -6, 0, (L) -39, 21, -5), and each amygdala side as seed regions (MNI coordinates: (R) 26, -2, -15, (L) -20, 2, -15).”*
- In the methods we have added (line 770): *“For exploratory analyses, seed regions were selected based on group differences identified in the main analysis (dmPFC, insula, amygdala).”*

Comment 9

Editorial policy checklist

- the name and date are missing at the top.
- ticked individual data points are shown when possible – but this is not actually done

Response 9

- We have now added the name and date at the top.
- We have added the individual data points as per the reviewer's suggestions (see Response 2).

Comment 10

Reporting summary

- Ticked that precise p-values are given, but this is not actually the case
- Statistic type for inference is missing the cluster-defining threshold which is critical

Response 10

- We have now included the precise p-values as per the reviewer's suggestion (see Response 1).
- We apologize for the mistake, this should have read "voxelwise-based family-wise error (FWE; $p < 0.05$)" as this is indeed invalid for clusterwise inference and not reflective of the methods in the paper. We have fixed this error.